# Proteomic characterization identifies clinically relevant subgroups of soft tissue sarcoma

Shaoshuai Tang[1,6], Yunzhi Wang [1,6], Rongkui Luo [2,6], Rundong Fang[1,6], Yufeng Liu[2,6], Hang Xiang[1], Peng Ran[1], Yexin Tong[1], Mingjun Sun[1], Subei Tan [1], Wen Huang[2], Jie Huang[2], Jiacheng Lv[1], Ning Xu[1], Zhenmei Yao[1], Qiao Zhang[1], Ziyan Xu[1], Xuetong Yue [1], Zixiang Yu [2], Sujie Akesu[2], Yuqin Ding[2,3], Chen Xu [2] ✉, Weiqi Lu[4] ✉, Yuhong Zhou[5] ✉, Yingyong Hou [2] ✉ & Chen Ding [1] ✉

Soft tissue sarcoma is a broad family of mesenchymal malignancies exhibiting remarkable histological diversity. We portray the proteomic landscape of 272 soft tissue sarcomas representing 12 major subtypes. Hierarchical classification finds the similarity of proteomic features between angiosarcoma and epithelial sarcoma, and elevated expression of SHC1 in AS and ES is correlated with poor prognosis. Moreover, proteomic clustering classifies patients of soft tissue sarcoma into 3 proteomic clusters with diverse driven pathways and clinical outcomes. In the proteomic cluster featured with the high cell proliferation rate, APEX1 and NPM1 are found to promote cell proliferation and drive the progression of cancer cells. The classification based on immune signatures defines three immune subtypes with distinctive tumor microenvironments. Further analysis illustrates the potential association between immune evasion markers (PD-L1 and CD80) and tumor metastasis in soft tissue sarcoma. Overall, this analysis uncovers sarcoma-type-specific changes in proteins, providing insights about relationships of soft tissue sarcoma.

Soft tissue sarcomas (STSs) are rare solid cancers arising from mesenchymal tissues, including muscle, adipose, bone, and fibrous tissues, comprising approximately 1% of adult malignancies[1]. According to the WHO classification, STS consists of more than 70 histological subtypes[2,3]. Different STS histological subtypes have diverse morbidities. The most common subtypes of STS contain liposarcoma (LPS), leiomyosarcoma (LMS), and undifferentiated pleomorphic sarcoma (UPS)[4]. Meanwhile, some histological subtypes, such as angiosarcoma (AS) and malignant peripheral nerve sheath tumor (MPNST) are

relatively rare. The 5-year overall survival (OS) of STS is approximately 50%[5], and differs among histological subtypes. For example, patients with LPS and SS had longer survival times than patients with other histological subtypes, such as UPS[6]. Moreover, different histological subtypes of STS also show diversity in lesion location, cancer cellular morphology, relapse/metastasis tendency, molecular aberrations, etc. Although World Health Organization (WHO) has depicted the taxonomy of STS, it is mainly based on lineages, prognosis, and driver alterations, sufficient global molecular profiling restricts its ability to

[1]State Key Laboratory of Genetic Engineering and Collaborative Innovation Center for Genetics and Development, School of Life Sciences, Institutes of Biomedical Sciences, Human Phenome Institute, Department of General Surgery, Zhongshan Hospital, Fudan University, Shanghai 200433, China. [2]Department of Pathology, Zhongshan Hospital, Fudan University, Shanghai, China. [3]Shanghai Institute of Medical Imaging, Shanghai, China. [4]Department of General Surgery, Zhongshan Hospital, Fudan University, Shanghai, China. [5]Department of Medical Oncology, Zhongshan Hospital, Fudan University, Shanghai, China. [6]These authors contributed equally: Shaoshuai Tang, Yunzhi Wang, Rongkui Luo, Rundong Fang, Yufeng Liu. ✉e-mail: xu.chen@zs-hospital.sh.cn; Lu.weiqi@zs-hospital.sh.cn; zhou.yuhong@zs-hospital.sh.cn; hou.yingyong@zs-hospital.sh.cn; chend@fudan.edu.cn

reveal the similarity of biological pathway changes among STS histological subtypes[7,8]. Further analysis based on global molecular profiling is required to distinguish the relationships of histological subtypes from molecular aspects and reflect their diverse tumor biology.

With the boost of the next-generation sequence, genome and transcriptome have accelerated revelations of molecular mechanisms in STS tumorigenesis and classification. As an example, the Cancer Genome Atlas (TCGA) studies have generated comprehensive molecular profiles including somatic mutations and RNA for STS from 6 histological subtypes[9]. This study identifies recurrent mutations of genes, including *TP53*, *ATRX*, and *RB1*. Moreover, integrated multiomics identify distinguished subgroups and cancer-related pathways. Despite the progression, many of these findings don't reach the expected efficacy in clinical experiments[10–12]. A potential explanation for this phenomenon is that previous researches focus on genomic or transcriptomic data, which could not panoramically reflect the molecular features of STS. Thus, the proteome, shaped by these genomic and transcriptomic alterations, representing tumor progression and infiltration of immune cells, has potential vulnerabilities that can be therapeutically exploited[13–15]. Some proteomic researches of STS have been published and provided valuable resource for understanding the molecular features of STS. Jessica Burns e.al. released a proteomic resource of STS to identify clinical subgroups related with clinical therapy[16]. However, considering the complexity of STS, more proteomic researches are required to facilitate the understanding of STS, especially for the heterogeneity of STS in different population.

Local relapse and metastasis are the primary threat to STS patients, accounting for 25–50% of patients based on initial stages and subtypes[17]. In the case of locally advanced and metastatic STS, the first-line chemotherapies are doxorubicin and/or ifosfamide[18]. The efficacy of this one-size-fits-all paradigm is limited since metastatic STS patients responded poorly. Resultingly, the OS of newly diagnosed metastatic STS is about 10–15 months[5,19]. So far, the metastatic mechanism of STS is not very clear. Some molecules have been reported to show elevated expressions in metastasis samples of STS, such as CD34, SOX10, CD117, and CTNNB1[20]. However, it is still functionally ambiguous how these metastasis-enriched proteins impact the progression of STS. Although different STS histological subtypes have diverse metastasis risks, the heterogeneity of some histological subtypes suggests the histological subtype couldn't be an independent risk factor to predict metastasis. More analyses focusing on molecular features from the pan-sarcoma aspect are required to explain the metastatic mechanism of STS.

The tumor microenvironment (TME) plays a significant role in clinical outcomes and response to therapy[21,22]. Some TME components have been proven to be associated with patient outcomes. For example, macrophage could enhance tumor progression and metastasis, which have been confirmed in breast, colon, and gastric cancers[23–25]. However, the relationship between most TME components and clinical outcomes in STS is unclear. Deciphering the tumor-immune microenvironment profile of cancer can improve the tailoring of targeted and immunotherapeutic strategies. Different STS histological subtypes have variable TME features. The histological subtypes with higher mutational burden, such as undifferentiated pleomorphic sarcoma (UPS) and dedifferentiated liposarcoma (DDLPS), generally contain dense infiltration by immune effector cells and respond better to immune therapy[26,27]. Even in these histological subtypes responding to immune therapy, the availability of immune therapy is still limited and only a small minority of patients might get meaningful clinical benefits, which reveals heterogeneity of TME in the histological subtype[19]. To enhance the efficiency of immunotherapy, it is important to characterize the diverse immune cell infiltration signatures of STS and to uncover the heterogeneity of TME in STS.

Here, we establish a 272 Chinese STS patients' cohort containing 12 sarcoma subtypes, including well-differentiated liposarcoma (WDLPS), myxoid liposarcoma (MLPS), dedifferentiated liposarcoma (DDLPS), angiosarcoma (AS), undifferentiated sarcoma (UPS), myxofibrosarcoma (MFS), other fibroblastic/myofibroblastic tumors (otherFS), leiomyosarcoma (LMS), rhabdomyosarcoma (RMS), malignant peripheral nerve sheath tumor (MPNST), synovial sarcoma (SS), and epithelioid sarcoma (ES). We integrate proteomic and phosphoproteomic data to uncover the similarity and differences of these STS histological subtypes, unravel the potential mechanism of STS metastasis, and understand their immune microenvironment features.

## Results

### Clinical and molecular features of the STS cohort

To systematically portray the proteomic landscape of STS, we collected formalin-fixed paraffin-embedded (FFPE) tissues from a cohort of 272 Chinese patients diagnosed with STS: 17 AS, 35 DDLPS, 5 ES, 52 LMS, 26 MFS, 11 MLPS, 6 MPNST, 8 otherFS, 15 RMS, 18 SS, 43 UPS, 36 WDLPS (Fig. 1A, B). Among the 272 tumor samples, 91 matched tumor-adjacent tissues (NATs) were collected ("Methods"). One 4 μm trick slide from each FFPE block was sectioned and stained by hematoxylin and eosin (H&E) for histological evaluation. Specifically, each tumor/tumor-adjacent sample was checked by three expert pathologists to confirm the sample quality according to the following criteria: For tumor samples: (1) pathologists evaluated and defined tumor area on the slices of FFPE specimens with tumor cell ratio (tumor purity) > 70%; (2) the histological subtypes of sarcoma were diagonalized by pathologists according to WHO classification of soft Tissue & Bone tumor[28]. For NAT samples: (1) pathologists evaluated and defined the tumor-adjacent areas on the slices of FFPE specimens with no observed tumor cells; (2) for different histological sarcoma subtypes, NATs were chosen based on tumor locations and the original lineages of tumors, according to WHO classification of soft Tissue & Bone tumor[28]. The specific NATs for different histological sarcomas were presented in Supplementary Data. 1. The representative H&E-stained slices showed the regions of tumors with their paired NATs, which confirmed the NAT types for distinctive tumors, and also indicated over 70% of tumor cellular purities for tumor regions, and no tumor cells in NATs (Supplementary Fig. 2A). We further estimated the tumor purities by ESTIMATE algorithm[29]. As a result, in concordant with the histologically evaluated tumor purity, the ESTIMATE algorithm calculated tumor purity ranged from 70 to 98% (median 78%) ("Methods", Fig. 1D). Survival analysis indicated that most STS histological subtypes couldn't be distinguished from each other by OS or disease-free survival (DFS) (Fig. 1C). Clinical data of the cohort, including the gender, age at diagnosis, histological subtype, FNCLCC grade, survival time, etc. were summarized in Fig. 1B and Supplementary Data 1.

A mass spectrometry (MS)-based proteomic analysis was conducted for all 363 samples (tumors, $n = 272$; NATs, $n = 91$). A phosphoproteomic analysis was conducted for 138 samples (tumors, $n = 114$; NATs, $n = 24$) using a Fe-NTA phosphopeptides enrichment technology ("Methods"). Quality control was applied on both peptide and protein levels with less than 1% FDR. As a result, 10,118 proteins and 37,842 phosphosites were identified, with 5593 proteins and 6483 phosphosites per sample on average ("Methods"; Supplementary Fig. 1G, Supplementary Data 1). All samples passed the quality control and showed good consistency in terms of proteome and phosphoproteome quantification (Supplementary Fig. 1E, F), exhibiting a typical unimodal (Gaussian or normal) distribution (dip statistical test). Among all phosphosites, 28,034 (74.1%) were serine (S), 8,762 (23.2%) were threonine (T) and 1,046 (2.7%) were tyrosine (Y), which was uniform with previous research (Supplementary Fig. 1H)[30,31]. Principal component analysis (PCA) showed that there were no time-related batch effects (Supplementary Fig. 1I). To surveil stability of the mass spectrometry, the HEK293 cell samples were utilized as the control samples ("Methods"). The correlations of these control samples were 0.83-0.95 and the median coefficient of variation (CV) was 0.14

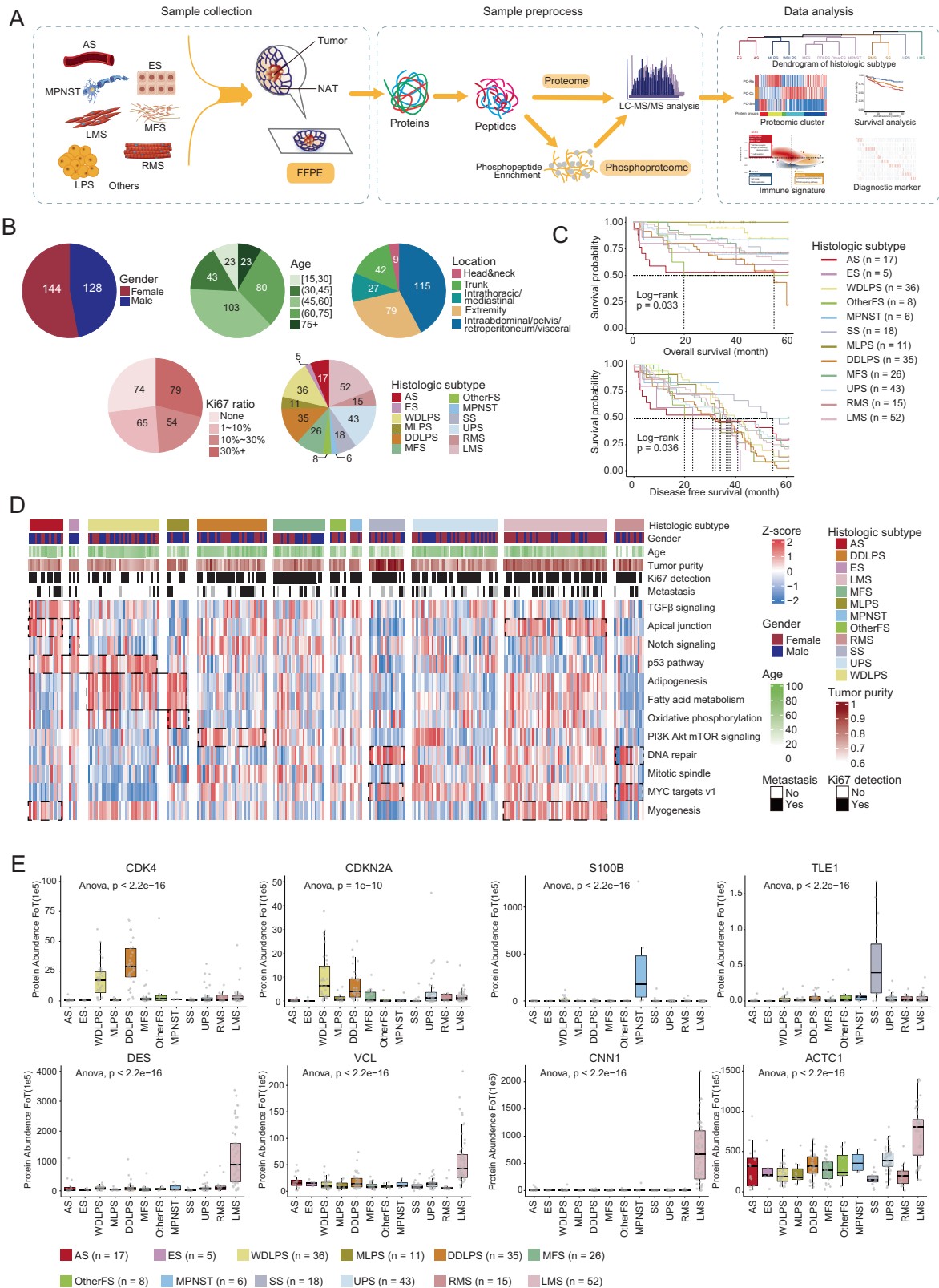

("Methods", Supplementary Fig. 1A–C), which is comparable to previously published papers[32], presenting the stability of the mass spectrometry across quality controls.

### Proteomic features of STS and NAT

To elucidate proteomic molecular alterations upon STS tumorigenesis, we performed a comparative analysis between STS (*n* = 272) and NAT

(*n* = 91) using proteomic and phosphoproteomic data. PCA analysis revealed a slight overlap between tumors and NATs (Supplementary Fig. 3A). To further illustrate the separation of tumor samples and NATs, we then conducted unsupervised clustering. As a result, 2 clusters (cluster1: NAT-distance and cluster2: NAT-similar) were determined ("Methods"). We then calculated specificity and purity of the two clusters ("Methods"). As a result, in concordant with the PCA

**Fig. 1 | Proteomic landscape of the soft tissue sarcoma cohort. A** Schematic representation of the experimental design, including sample collection, sample pre-process, LC-MS/MS analysis, and data analysis. **B** Pie plots indicated clinical features of the STS cohort, including gender, age, tumor location, KI67 ratio, and STS histological subtypes. **C** Kaplan-Meier curves for OS (up) and DFS (down) of tumor patients stratified by the 12 histological subtypes (log-rank test). **D** Heatmap of enriched cancer hallmarks in STS histological subtypes. The significantly up-regulated hallmarks in some sarcoma histological subtypes are marked by black dotted box. **E** Boxplots indicated protein abundances of known markers of specific STS histological subtypes across 12 histological subtypes in our cohort. The biologically independent samples for each histological subtypes are as follows: AS ($n = 17$), ES ($n = 5$), WDLPS ($n = 36$), MLPS ($n = 11$), DDLPS ($n = 35$), MFS ($n = 26$), otherFS ($n = 8$), MPNST ($n = 6$), SS ($n = 18$), UPS ($n = 43$), RMS ($n = 15$), LMS ($n = 52$). The middle bar represents the median and the box represents the interquartile range. Bars extend to 1.5× the interquartile range. Two-way analysis of variance (ANOVA) was used for statistical test. Source data are provided as Source Data files.

analysis, around 89% of the NATs were grouped into cluster1, and 56% of the tumors were grouped into cluster2. Forty-four percent of tumors were grouped with NATs, implying that these tumors might not show significantly diverse proteomic features compared to NATs (Supplementary Fig. 3B). Since our cohort contained diverse histological types of sarcomas and NATs, we then hypothetically assumed that the overlap between tumors and NATs might be caused by the diverse tumor heterogeneity of different STS histological subtype. To verify this assumption, we separately conducted PCA analysis for each histological type of sarcomas. As a result, the tumors were perfectly separated with NATs in each histological type of sarcomas (Supplementary Fig. 3C). These results confirmed that the overlap between tumors and NATs was probably caused by the tumor heterogeneity of diverse histological sarcomas, further revealed the value of research in deciphering the tumor heterogeneity of different histological sarcomas.

To illustrate the distinctive features of tumors and NATs, we conducted differential expression analysis, utilizing all control samples. It was identified that 1885 proteins and 2450 phosphosites were significantly overrepresented in tumor samples of STSs (Wilcoxon rank test, fold change >1.5; adjusted $p$ value < 0.05). Gene Ontology (GO) enrichment analysis based on proteomic and phosphoproteomic data revealed that proteins of some classical oncogenic pathways were significantly elevated in STS, including RNA splicing, NF-kappaB signaling, JNK cascade, and cell growth. Meanwhile, the protein decreased in STS participated in ATP metabolic process, glycogen metabolic process, and actin filament organization were (Supplementary Fig. 4A). Besides, the distinctive features of tumors and NATs were further confirmed by the comparison analysis of sarcoma and paired NAT in a patient-specific manner (Supplementary Fig. 4B).

Meanwhile, pair-wised comparative analysis between tumors and NATs among 12 histological sarcoma subtypes revealed besides the common features of sarcoma tumors, the different histological sarcoma tumors and their corresponding NATs showed specific features (Supplementary Data. 2). For instance, the pathways enriched in WDLPS included the VEGFA & VEGFR2 signaling pathway and HOXA1 target signaling pathway, whereas pathways enriched in its pair-wised NATs (lipid tissues) included organic acid catabolic process, carboxylic acid catabolic process, and ATP synthesis coupled electron transport. Meanwhile, the pathways enriched in RMS include MYC targets up, signaling by interleukins and DNA replication, while, pathways enriched in their pair-wised NATs (skeletal muscle tissues) were muscle system processing, muscle contraction, etc. Along with these findings, the pathways dominantly enriched in MPNST were MAPK cascade, P53 regulation pathway, and cell cycle, whereas pathways enriched in its pair-wised NATs (nerve tissues) were intermediate filament organization and collagen fibril organization.

To further illustrate the clinical relevance of our data, we referred to a published drug database (https://www.cancerrxgene.org/) and identified ten proteins that showed high expression in STS and could be targeted by FDA-approval drugs, including ACLY, CDK4, PGD, and etc. (Supplementary Fig. 4C). We filtered 1,169 phosphosites that showed higher phosphorylation level in STS (Supplementary Fig. 4D). We further inferred kinase activities based on these phosphosites ("Methods") and observed a total of 13 kinases had significantly

increased activities in STS, including targets of approved inhibitors (CDK4 and CDK6) (Supplementary Fig. 4E). Most of these kinases were cyclin-dependent kinases (CDK1, CDK2, etc.) and casein kinases (CSNK1A1, CSNK2A1, and CSNK2A2), which mainly participated in cell cycle regulation[33,34]. Survival analysis also revealed poor-prognosis-associated proteins/phosphoproteins were enriched in cell-cycle related pathways, including regulation of G2/M transition of mitotic cell cycle, mRNA catabolic process, DNA replication, and cell cycle G1/S phase transition (Supplementary Fig. 4F), validating a relationship between the activity of cell cycle and adverse clinical outcome. We further constructed the kinase-substrate regulation network consisting of these kinases and important substrates of them, including RB1, AKT1, CTNNB1, STAT1, and JUN (Supplementary Fig. 4G). Together, these results showed the elevation of cell proliferation, WNT signaling pathway, and NF-kappaB signaling, in STS, implying their role in tumorigenesis and might potentially be targeted for STS treatment.

## Proteomics identifies histology-related molecular signatures of STS subtypes

To further unravel the difference of molecular features among STS histological subtypes, we performed single sample gene set enrichment analysis (ssGSEA) of cancer hallmarks based on proteomic data. The result indicated that the TGFβ signaling pathway were most enriched in AS than other histological subtypes. Fat-metabolism-related pathways, including adipogenesis and fatty acid metabolism, were dominantly enriched in WDLPS and MLPS, whereas, DDLPS had an enriched PI3K-Akt-mTOR signaling pathway (Fig. 1D, Supplementary Fig. 5A, B). Meanwhile, the pathway, DNA repair process, was overrepresented in SS and RMS. We also verified the histological specific features of sarcomas by the comparing analysis of sarcoma and control tissue in a patient-specific manner (Supplementary Fig. 5A, B).

We also evaluated the expression of reported diagnostic markers across various subtypes, including CDK4 (a marker for WDLPS), CDKN2A (a marker for DDLPS), S100B (a marker for MPNST), TLE1 (a marker for SS), DES (a marker for LMS), and others[35–40] (Fig. 1E, Supplementary Fig. 6A–C). Consistent with previous reports, CDK4 and CDKN2A showed opposite distribution tendencies in DDLPS and WDLPS: CDK4 was relatively higher in DDLPS and CDKN2A was relatively higher in WDLPS (Fig. 1E). Combined with patients' prognosis, we found that patients with DDLPS showed poorer prognosis than patients with WDLPS. In consistent with this phenomenon, survival analysis revealed that CDK4 was negatively correlated with OS, whereas, CDKN2A was positively correlated with OS (Supplementary Fig. 6C). We further validated this observation in TCGA sarcoma cohort[9] (Supplementary Fig. 6B). The expression of S100B and TLE families was observed to be separately elevated in MPNST and SS, which was consistent with previous reports at transcriptome level (Fig. 1E, Supplementary Fig. 6A). Besides the above reported diagnostic biomarkers, we identified a series of proteins that showed dominant expressions in different histological subtypes of STS and presented high sensitivities and specificities, such as PECAM1, ICAM2, and NOS3 in AS (Supplementary Fig. 7A, B, Supplementary Data 1). Immunohistochemistry (IHC) staining further validated the prognostic effects of some proteins, including PECAM1, CD36, and IGFBP6 (Supplementary Fig. 7C). In sum, these results suggested that the panel of biomarker

candidates could be used to assist diagnosis in supplement to other known biomarkers to distinguish different STS histological subtypes in clinic.

## Unsupervised hierarchical clustering revealed common and distinctive features of different histological sarcomas

Histologic clustering is the gold standard for STS diagnosis, but the molecular similarity and diversity among subtypes are still unclear. To further investigate the intrinsic common features of STS histological subtypes, we employed hierarchical clustering on the 12 STS histological subtypes ("Methods"). As a result, 6 clusters (HC1 - HC6) were selected based on both silhouette coefficient and clinical relevance (Fig. 2A, C, Supplementary Fig. 8A–D). HC1 contained AS and ES; HC2 included MLPS and WDLPS; HC3 consisted of MFS, DDLPS, otherFS, and MPNST; HC4 contained RMS and SS; HC5 was UPS and HC6 was LMS (Fig. 2A). Survival analysis of these hierarchical clusters revealed that patients belonging to HC1 (AS and ES) had the poorest OS among all clusters. Meanwhile, patients belonging to HC2 (MLPS and WDLPS) had a significantly longer OS (Fig. 2B).

Aimed at exploring the molecular features of each hierarchical cluster, we performed ssGSEA based on proteomic data (Fig. 2C). Statistical results of ssGSEA indicated that pathways, including TGFβ signaling, integrin pathway, epithelial cell migration, and actin cytoskeleton reorganization, were dominantly enriched in HC1 (Fig. 2C, Supplementary Fig. 9A). SHC1, PTK2, and PECAM1, which participated in these HC1-enriched pathways, are significantly correlated with poor prognosis (Supplementary Fig. 9B). Pathways related to metabolism, including metabolism of vitamins & cofactors, PPARα pathway were dominantly enriched in HC2. Some proteins participating in the metabolism process, such as APOVA4 and ACADVL, are significantly correlated with good prognosis (Supplementary Fig. 9B). HC3 had enriched transport-related pathways, including intra-Golgi traffic, membrane trafficking, and trafficking regulated by Rab and elevated relevant kinases including GAK and SCYL1. RNA process and metabolism pathways, such as tRNA processing, RNA degradation, and spliceosome, were observed with significant distribution tendency in HC4. HC5, uniquely consisting of UPS, was featured by immune-related pathways, including T cell receptor signaling pathway and Fc receptor signaling. Cell-cycle-related proteins (MCM7, MCM5, CDK2, PCNA, etc.) were enriched in HC6 (LMS).

Noticeably, we found that our hierarchical clustering divided the lipid sarcoma (WDLPS, MLPS, and DDLPS) into two clusters. Particularly, DDLPS were clustered together with fibrosarcomas (MFS and otherFS) and MPNST in HC3. WDLPS and MLPS were clustered into another cluster (HC2). Considering different differentiation levels of WDLPS, MLPS, and DDLPS, these findings revealed the difference of tumor differentiation within lipid sarcomas might lead to the diverse molecular features between DDLPS and WDLPS, further implying that the degree of tumor differentiation might serve as an important factor in determining the molecular features of sarcomas within lipid sarcomas. Because DDLPS is more metastatic and proliferative than WDLPS[35], we compared the ratio of KI67-positive tumor cells in WDLPS and DDLPS. DDLPS showed an obviously higher ratio of KI67-positive tumor cells than WDLPS (Supplementary Fig. 9C). Consistently, HC3 also presented the higher ratio of KI67-positive tumor cells than HC2, implying that HC3 was featured with the fast cell proliferation characteristics (Supplementary Fig. 9C).

GSVA analysis revealed that DDLPS (HC3) could be distinguished from WDLPS and MLPS (HC2) by elevated enrichments of Rab pathway (Fig. 2C, Supplementary Fig. 9D). The elevated protein expression of Rab GTPases including RAB14, RAB5A, RAB2A, etc. in HC3 confirmed the increased Rab pathway in HC3 (Supplementary Fig. 9E). Moreover, among the Rab GTPases that showed elevated expression in HC3, we observed that the protein abundance of RAB2A and RAB14 were significantly correlated with patients' prognosis (Supplementary Fig. 9F).

Previous researches have reported that Rab GTPases participate in cell autophagy and RAB2A has been proved to regulate the formation of autophagosome and autolysosome[41–43]. As researches have indicated that the elevated autophagy might be associated with tumor proliferation[44], we then hypothetically assumed that the elevated autophagy might lead to faster tumor cell proliferation in HC3.

Aim to confirm this assumption, we compared the autophagy pathway between HC2 and HC3, and found that both the autophagy pathway enrichment scores as well as autophagy markers (ATG5, ATG7, MTOR, WIPI1) showed elevation in HC3 than HC2 (Supplementary Fig. 9G, H). Moreover, proliferation index of sarcoma is both correlated with protein expression of RAB2A and autophagy pathway GSVA enrichment scores (Supplementary Fig. 9I). These findings illustrated that comparing to WDLPS and MLPS (HC2), DDLPS (HC3), showed fast tumor cell proliferation features, which might be caused by the RAB2A-associated autophagy process.

In sum, our data revealed clinical relevance and could help to illustrate the common features among different histological sarcomas and could further decipher the distinctive biological features of lipid sarcomas varies with degrees of differentiation.

## SHC1 contributes to poor prognosis in AS and ES depending on phosphorylating ADD2

To investigate cluster-specific proteins which were correlated with survival outcomes, we focused on proteins highly expressed in HC1 or HC2 (the hierarchical clusters with the poorest and best OS). Pathways enriched in HC1/HC2 and associated proteins were then filtered out for further survival analysis. To be more specific, HC1-enriched proteins (student's t test, fold-change >1.5, adjusted $p$ value <= 0.05) participating in TGFβ signaling, integrin pathway, epithelial cell migration, or actin cytoskeleton reorganization pathways, were selected. Meanwhile, HC2-enriched proteins (student's $t$ test, fold-change >1.5, adjusted $p$ value <= 0.05) participating in pathways correlated with metabolism (as shown in Fig. 2C) were filtered out. The correlations between abundances of these filtered proteins and patients' OS were calculated using the Cox proportional hazard model (Supplementary Data 2). High expressions of SHC1, PTK2, and ITGB2 were significantly correlated with the poorer prognosis (Fig. 2D).

Among these proteins related to OS, SHC1 presented the highest hazard ratio, suggesting SHC1 could play an important role in survival outcomes. SHC1, as an adaptor protein, has been reported to be recruited and activated by growth factor signaling transduction and cascade, including TGFβ signaling[45–47]. After activation, SHC1 could regulate diverse biological processes involved in cell growth and proliferation, cell migration, and angiogenesis[48]. Consistently, we found a significantly positive correlation between the protein abundance of SHC1 and the TGFβ signaling pathway enrichment score (Pearson's correlation, r = 0.15, $p$ value = 0.028), suggesting an association between SHC1 and the TGFβ signaling in sarcoma (Fig. 2E). We then investigated the expression of proteins participating in TGFβ signaling and found among the TGFβ families, TGFβ3 showed a statistically positive correlation with SHC1 (Pearson's correlation, r = 0.25, $p$ value = 0.026), suggesting the potential association between TGFβ3 and SHC1, and implying they might cooperate to impact downstream signaling pathways (Fig. 2E). To illustrate the downstream pathways which might be regulated by TGβ3 and SHC1 and led to poor prognosis of patients in HC1, we conducted correlation analysis and found there're significantly positive correlations between protein abundance of SHC1 and two HC1-enriched biological pathways, actin cytoskeleton reorganization (Pearson's correlation, r = 0.21 $p$ value = 0.0049) and epithelial cells migration (Pearson's correlation, r = 0.22 $p$ value = 0.0027) (Supplementary Fig. 10C).

These results promote us to further explore the potential relationship among SHC1, TGFβ and elevated tumor cell migrations of HC1 cluster, we utilized ASM cell line, the cell line of AS, to represent the

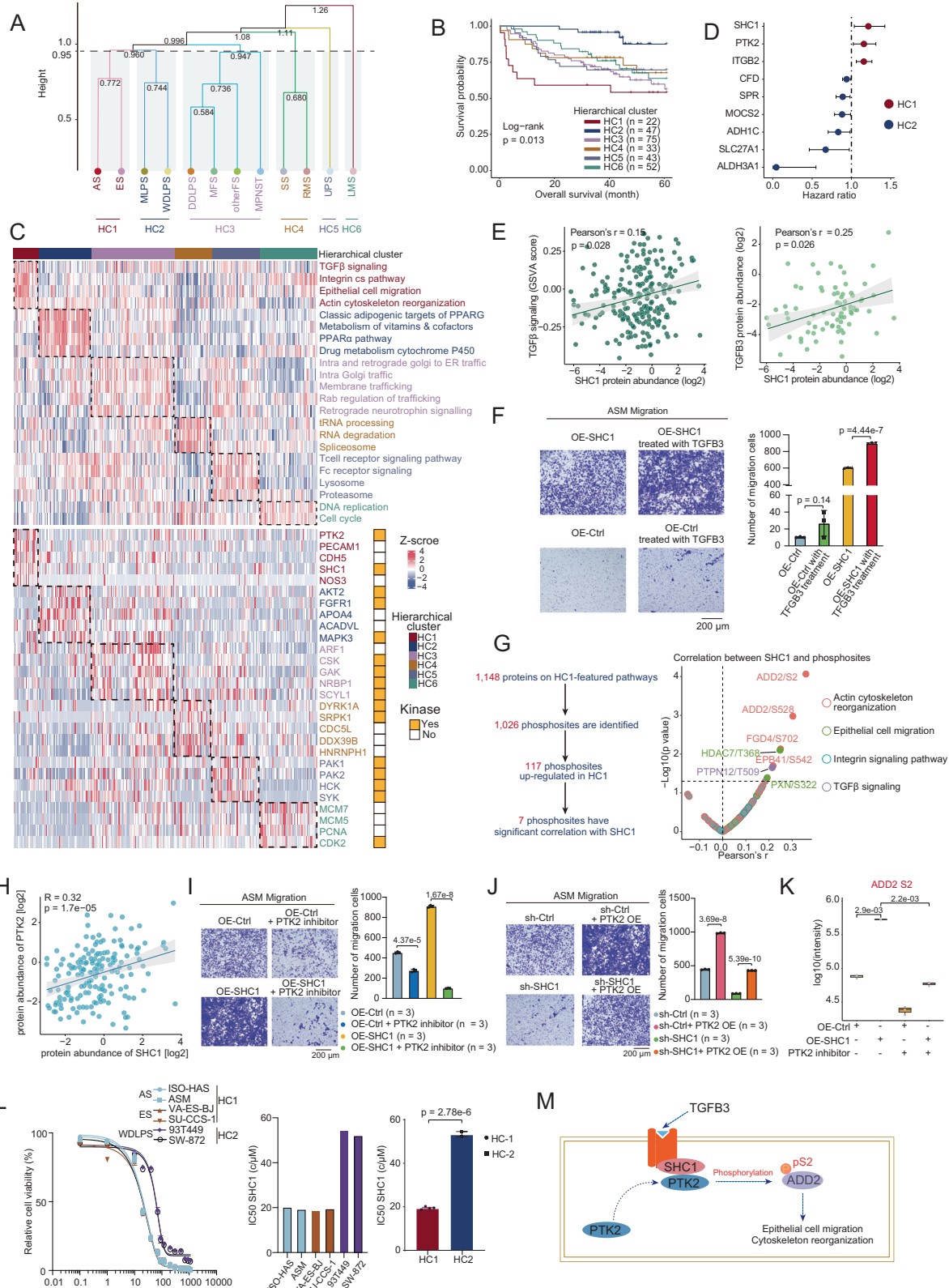

HC1 cluster. We constructed the SHC1-overexpressed vector and transfected it into the ASM cell line (SHC1-OE-ASM). Meanwhile, we also utilized shRNA to knock down SHC1 in ASM (SHC1-KD-ASM). RT-PCR analysis confirmed the significantly elevated expression of SHC1 in SHC1-OE-ASM and significantly decreased expression of SHC1 in SHC1-KD-ASM (Supplementary Fig. 10A). We then evaluated the cell migration rates using transwell assay. As a result, SHC1-OE-ASM showed

increased cell migration ability, whereas SHC1-KD-ASM exhibited decreased cell migration ability (Supplementary Fig. 10B).

We then treated SHC1-OE-ASM and OE-Ctrl-ASM (transfected with empty vectors as the control) with TGFB3 and evaluated the tumor cell migration rates. As a result, SHC1-OE-ASM treated with TGFB3 showed significantly elevated tumor cell migration rates, whereas OE-Ctrl-ASM showed no significantly changes in tumor cell migration rates after

**Fig. 2 | Hierarchical clusters of STS histological subtypes. A** Dendrogram of hierarchical clusters (HCs). **B** Kaplan-Meier curves for OS of tumor patients stratified by HCs (log-rank test). **C** The heatmap indicates differentially enriched pathways (up panel) and key proteins (down panel) in different HCs. Two-sided student's t test is used for detecting the enrichment of pathways and proteins. **D** The forest plot presents the prognosis-related proteins in HC1 and HC2. Multivariate Cox proportional hazard models is used for survival analysis. The dots represent the hazard ratio values and the bars represent the 95% confidence intervals. The survival information from all 272 sarcoma patients were used for calculating the hazard values. **E** The scatter plot describes the correlation between the SHC1 abundance and TGFβ signaling GSVA scores (left, $n = 214$) or the TGFB3 abundance (right, $n = 170$). **F** The effect of TGFB3 on the migration of ASM cells is confirmed by transwell assay. The bar plot indicates the migrated ASM cells under different treatments. **G** The flow chart and the volcano plot present the phosphosites both up-regulated in HC1 and significantly correlated with SHC. Pearson's correlation is used for associated analysis. **H** The scatter plot presents the positive correlation between the protein expression of PTK2 and SHC1 ($n = 170$). **I, J** The transwell assay and bar plots indicate the migrated cell counts of ASM cells under different treatments. **K** The boxplot indicates the phosphorylation intensity of ADD2 S2 under different treatments (n = biological repeats per group). The middle bar represents the median and the box represents the interquartile range. Bars extend to 1.5× the interquartile range. **L** Dose-response curves and IC50 values presents the different response effects to the SHC1 inhibitor in 6 cell lines. **M** The diagram displays the mechanism of SHC1-dependent cell migration. For (**F, I, J, L**), three biologically independent experiments were performed for each group. Data are presented as mean values ± SE. Unpaired two-side Student's $t$ test was used for statistical test. For (**E, H**), the Pearson's correlation is used for associated analysis. The error band represents the 95% confidence interval of the regression line. Source data are provided as Source Data files.

treated with TGFB3 (Fig. 2F). These results confirmed the role of TGFB3 in activating SHC1-mediacted tumor cell migrations.

Published researches have indicated that SHC1 participates in various biological process, and might regulate downstream pathways through phosphorylation[45,49,50]. To this end, we investigated phosphoproteins and corresponding phosphosites which were enriched in HC1 and participated in SHC1-correlated pathways (Fig. 2G). As a result, among the 7 phosphosites which had a significant positive correlation with SHC1 abundance, ADD2 Ser2 had the highest correlation with SHC1 (Pearson's correlation, $r = 0.36$, $p$ value = 6.79e−5) (Fig. 2G). ADD2 Ser2 presented high expression in HC1, independent of the protein abundance of ADD2 (Supplementary Fig. 10D, E). ADD2 belongs to a family of membrane skeletal proteins involved in cell-cell adhesion, cell motility, and cell signaling[51]. Consistently, we found an obvious correlation between ADD2 Ser2 and ssGSEA enrichment scores of actin cytoskeleton reorganization or positive regulation of epithelial migration (Supplementary Fig. 10F). Survival analysis also uncovered that the high phosphorylation level of ADD2 Ser2 was associated with poor prognosis (Supplementary Fig. 10G).

To elucidate the kinase that related to SHC1 and might regulate the phosphorylation of ADD2 at Ser2 in HC1, we referred to the public database (PhosphoSite [https://www.phosphosite.org/homeAction.action], Phos-pho.ELM [http://phospho.elm.eu.org/dataset.html], and PhosphoPOINT [http://kinase.bioinformatics.tw/]) and conducted correlation analysis. As a result, among the kinases reported to regulate phosphorylation of ADD2, PTK2 was identified as the kinase showing the most significantly correlation with SHC1 and comparatively higher expression in HC1 cluster (Fig. 2H, Supplementary Fig. 10H). To further investigate the role of PTK2 in impacting cell migration, SHC1-OE-ASM and OE-Ctrl-ASM cell lines were treated with PTK2 inhibitors and the cell migration was evaluated by transwell assay. As a result, inhibiting PTK2 could significantly decrease the cell migration rates increased by SHC1 overexpression (Fig. 2I). Moreover, overexpression of PTK2 in SHC1-KD-ASM significantly increased cell migration which was inhibited by knocking down SHC1(Fig. 2J). These results implied that PTK2 participated in cell migration driven by SHC1. We further performed comparative phosphoproteomic analysis between SHC1-OE-ASM treated with or without PTK2 inhibitor. As a result, the phosphorylation of proteins such as ADD2 Ser2, FGD4 Ser702 and EPB41 Ser542, which participate in actin cytoskeleton reorganization and epithelial cell migration, showed significant elevation in SHC1-OE-ASM and significant reduction in SHC1-OE-ASM treated with PTK2 inhibitor (Fig. 2K, Supplementary Fig. 10I). These observations confirmed the role of PTK2 in phosphorylating ADD2 at Ser2 and elevating actin cytoskeleton reorganization pathways.

To test the clinical relevance of SHC1 targeting for HC1 (AS and ES), we collected 6 different sarcoma cell lines, including 2 AS cell lines (ISO-HAS and ASM), 2 ES cell lines (VA-ES-BJ and SU-CCS-1), and 2 WDLPS cell lines (93T449 and SW-872). All six cell lines were cultured and treated with different degrees of carbamoylcholine (the inhibitor of SHC1). Then effects of carbamoylcholine on cell viability were measured. Compared with WDLPS cell lines, AS and ES cell lines were more sensitive to the SHC1 inhibitors with lower $IC_{50}$ values (median $IC_{50}$: 19.41 μM [AS & ES] vs 42.08 μM [WDLPS]) (Fig. 2L, Supplementary Data 2), proving that SHC1 had a more effective impact on cell viability in AS and ES. In sum, after activated by TGFB3, SHC1 could recruit PTK to phosphorylate ADD2 Ser2 and other phosphosites participating actin cytoskeleton reorganization and epithelial cell migration, which is correlated with poor prognosis (Fig. 2M).

## Proteomics subtypes unravel the heterogeneity within STS histological subtypes

Besides hierarchical clustering based on proteomic data which had unraveled the relationships among STS histological subtypes, we further performed proteomic-based subtyping to characterize biological themes that cross histological boundaries and unite individual tumors of disparate histologies. Consensus clustering based on global proteomics data of 272 tumor samples identified three clusters (PC-Ra, PC-Cc, and PC-Sm) with distinct clinical outcomes, proliferation score, stroma score, and pathway enrichment ("Methods", Fig. 3A). Remarkably, the proteomic clusters significantly differed in OS (log-rank test, $p$ value = 9.6e-3), in which PC-Ra and PC-Cc had a worse prognosis than PC-Sm ("Methods", Fig. 3C). To explore the potential clinical factors might impact the correlation between prognosis and proteomic clusters, we compared the baselines among the three clusters. As a result, there were no significant differences of basic clinical characteristics between the three proteomic clusters, including age, gender, postoperative treatment, and tumor location, implying that the proteomic cluster is an independent risk factor relevant with prognosis (Supplementary Data. 3).

To explore the diversity of proteomic characteristics and signaling pathways among proteomic clusters, we utilized consensus clustering based on 2757 proteins differently expressed among proteomic clusters (ANOVA analysis, adjusted $p$ value <= 0.001, Supplementary Data 3) and identified 6 protein sets that illustrate different expression models of proteins among the three protein clusters ("Methods", Fig. 3A). PC-Ra had over-represented RAS-MAPK cascade and angiogenesis, with a low-grade oxidation−reduction process. PC-Cc were featured by high expression of proteins participating in cell-cycle-related pathways, including mRNA processing, DNA repair, cell population proliferation, and chromosome organization. Consistently, PC-Cc had the highest cell cycle score (PC-Ra:0.23, PC-Cc:0.29, PC-Sm:0.15; Kruskal-Wallis's test: $p$ value < 2.2e−16) (Fig. 3A). Meanwhile, PC-Cc had the down-regulated component process. PC-Sm had enriched extracellular matrix and lipid metabolism, with the highest stroma score (PC-Ra:255, PC-Cc: 93, PC-Sm: 494; Kruskal-Wallis's test: $p$ value = 3.8e−10) (Fig. 3A). Moreover, compared

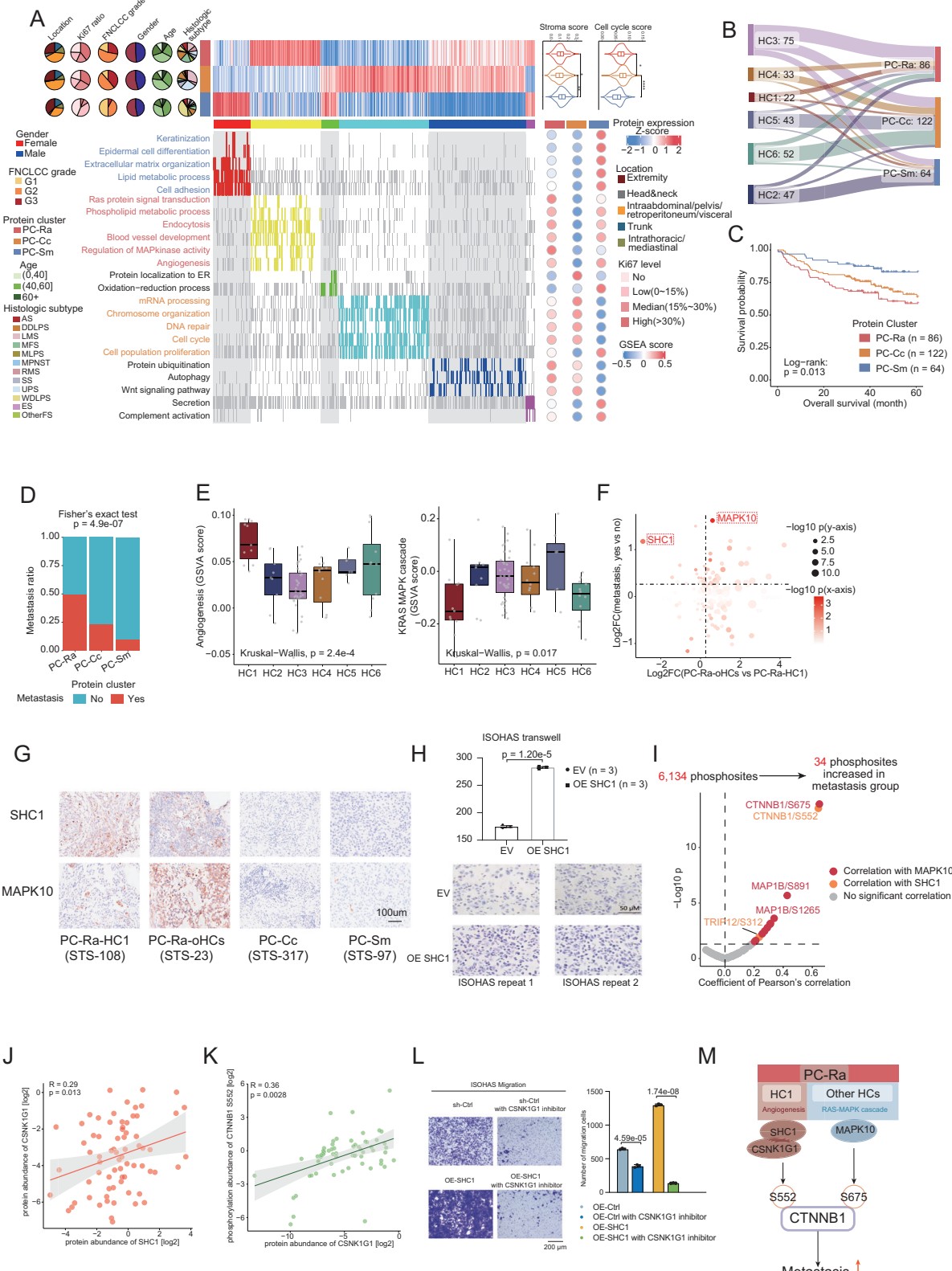

with PC-Ra and PC-Cc, PC-Sm had lower enrichment of proteins participating in the Wnt signaling pathway.

To assess the intersection of our proteomic clusters with histological subtypes, we compared subtypes assignment of 272 STS patients using each of the two classifiers. Intriguingly, the LMS histological subtype (HC6) was distributed orthogonally across our three proteomic clusters, implying that this subtype is not restricted to a

distinctive proteomic feature (Fig. 3A, B). By conducting further integrative analysis of proteomic clusters and patients' clinical features, we found that patients belonging to PC-Ra had the highest metastasis ratio (49.4%) after the primary surgeries than PC-Cc and PC-Sm in the following 5 years (Fig. 3D). Pathways enriched in PC-Ra included positive regulation of MAPK cascade (Kruskal-Wallis's test, *p* value = 8.5e-8), RAS protein signaling transduction

**Fig. 3 | Proteomics clusters of STSs. A** The heatmap presents 3 proteomic clusters (PCs), 6 protein groups and enriched pathways in protein groups. Clinical and molecular features are also presented, including location, gender et al. **B** The Sankey diagram illustrates relationships between hierarchical clusters and proteomic clusters. **C** Kaplan-Meier curves for OS of proteomic subtypes (log-rank test). **D** The bar plot indicates the metastasis ratio of proteomic clusters. The number of patients with available metastasis information: PC-Ra (metastasis: 41, non-metastasis: 42), PC-Cc (metastasis: 22, non-metastasis: 73), PC-Sm (metastasis: 5, non-metastasis: 49). Fisher's exact test is used for statistical analysis. **E** Boxplots indicates GSVA scores of angiogenesis and KRAS-MAPK cascade among hierarchical clusters. The sample number for each group: HC1($n = 11$), HC2($n = 9$), HC3($n = 35$), HC4($n = 12$), HC5($n = 7$), HC6($n = 12$). The middle bar represents the median and the box represents the interquartile range. Bars extend to 1.5× the interquartile range. Kruskal-Wallis's test is used for statistical analysis. **F** Scatter plot presents the proteins with different expression models in the four groups:

metastasis, non-metastasis, PC-Ra-oHCs, and PC-Ra-HC1. The Wilcoxon rank test is used for statistical analysis. **G** IHC images of SHC1 and MAPK10, Scale bar = 100 μm. **H** The transwell experiments confirm the promotion of SHC1 on cell migration of ISOHAS cell line. Scale bar for images of the transwell experiments, 50 μm. **I** The volcano plot illustrated the phosphosites enriched in metastasis group and correlated with SHC1 or MAPK10. Th Pearson's correlation test is used for associated analysis. **J** The abundance's correlation between CSNK1G1 and SHC1 ($n = 72$). **K** The correlation between CSNK1G1 abundance and phosphorylation level of CTNNB1 Ser552 ($n = 58$). **L** The effect of the SHC1-CSNK1G1 axis on the migration of ISOHAS is confirmed by transwell assay. **M** The diagram illustrates two different mechanisms associated with metastasis in PC-Ra. For (**H, L**), three biological repeats are performed per group. The data is presented as mean value ± SE. The two-sided student's $t$ test is used for statistical analysis. For (**J, K**), the error band represents the 95% confidence interval of the regression line. Pearson's correlation test is used for associated analysis. Source data are provided as Source Data files.

($p$ value = 7.9e−10), and sprouting angiogenesis ($p$ value = 1.2e−11). To further explore the mechanisms of metastasis in PC-Ra, we focused on the dominant pathways in PC-Ra. As a result, we observed higher ssGSEA enrichment scores of the three pathways, angiogenesis (Wilcoxon's test, $p$ value = 3.4e−7), MAPK cascade ($p$ value = 5.2e−8), and RAS signaling transduction ($p$ value = 4.3e−5) (Supplementary Fig. 11B) in patients with metastasis.

Histologically, PC-Ra contained all six hierarchical clusters, we then divided the PC-Ra according to HCs and evaluated the ssGSEA enrichment scores of these three metastasis-related pathways. As a result, HC1 (AS and ES) was distinguished from other HCs by the highest enrichment of angiogenesis (Kruskal-Wallis's test, $p$ value = 2.4e−4) and the lowest enrichment of the RAS-MAPK cascade pathway ($p$ value = 0.017) in HC1 (Fig. 3E). Thus, we hypothesized that the metastasis of patients in PC-Ra and HC1 (PC-Ra-HC1) might be mainly driven by angiogenesis, and the metastasis of patients in PC-Ra and other HCs (PC-Ra-oHCs) might be mainly driven by RAS-MAPK signaling pathway. Consistently, focusing on the proteins participating in these PC-Ra featured pathways, we analyzed different expression tendencies of them in two aspects (PC-Ra-HC1 vs PC-Ra-oHCs, metastasis vs non-metastasis) to explore key proteins related to metastasis (Fig. 3F). We found the enrichment of SHC1 in metastasis samples of PC-Ra-HC1 (Student's $t$ test: p value = 0.042) and MAPK10 in metastasis samples of PC-Ra-oHCs (Student's $t$ test: $p$ value = 2.1e−4) (Fig. 3F). IHCs also validated the exclusively the high expression of SHC1 and MAPK10 separately in PC-Ra-HC1 and PC-Ra-oHCs (Fig. 3G). To verify the promotion of SHC1 for metastasis in PC-Ra-HC1, we constructed SHC1-overexpressed ISOHAS cell lines (ISOHAS-SHC1-OE). As a result, the transwell migration assay showed increased migration ability of ISOHAS-SHC1-OE cell lines (Fig. 3H). Focusing on the PC-Ra-HC1, since we have confirmed that as an adaptor protein, SHC1 could interact with PTK2 and phosphorylated ADD2 to elevate the actin cytoskeleton reorganization pathway in HC1, we then evaluated the expression of PTK2 and phosphorylation of ADD2 in HC1-PC-Ra. As a result, comparing to HC1-oPCs (HC1 samples which were grouped into other proteomic clusters), PTK2 and phosphorylation of ADD2 at S2 showed no significant elevation in HC1-PC-Ra (Supplementary Fig. 11C-D), implying that PTK2 phosphorylated ADD at S2 might be the common features shared by both HC1-PC-Ra and HC1-oPCs, and SHC1 might cooperate with other kinases to promote metastasis of HC1-PC-Ra.

We then combined different expression and protein-phosphosite correlation analyses to find phosphosites might be downstream participators in phosphorylation signaling transduction of SHC1 or MAPK10 (Fig. 3I). Noticeably, CTNNB1 Ser552 had a positive correlation with SHC1 (Pearson's correlation: r = 0.64, $p$ value = 1.45e−14) and CTNNB1 Ser675 had a positive correlation with MAPK10 (Pearson's correlation: r = 0.65, $p$ value = 5.87e−15) (Fig. 3I, Supplementary Data 3). Both phosphorylation of CTNNB1 Ser552 and Ser675 showed

elevated expression in metastasis patients and have been proven to be related to the activation of CTNNB1[52–55] (Supplementary Fig. 11E). Functionally, CTNNB1 is one of the core proteins in the Wnt signaling pathway, whose activation has been reported to be related to metastasis, in gastric, colon, and breast cancer[56–58]. To validate the correlation between SHC1 and CTNNB1 Ser552, we cultured patient-derived cancer cells (PDCs) from patients of PC-Ra with different expression levels of SHC1 (high-SHC1 group: $n = 3$, average SHC1 abundance = 9.81; control group: $n = 3$, average SHC1 abundance = 0.0028) and collected these PDCs to performed LC-MS based phosphoproteomic analysis (Supplementary Fig. 11F). Among all phosphosites elevated in the high-SHC1 group, CTNNB1 Ser552 showed the most significant fold change of abundance between the high-SHC1 group and control group, verifying the potential association between SHC1 on CTNNB1 Ser552 (Supplementary Fig. 11G). To identify the potential kinase that associated with SHC1, the phosphorylation of CTNNB1 at Ser552, and the tumor metastasis, we referred to the public database and performed further correlation analysis. As a result, among the public reported kinases of CTNNB1, CSNK1G1 showed the significantly positive correlation with both SHC1 and the phosphorylation of CTNNB1 at Ser552 (Fig. 3J–K). Consistently, the phosphorylation of CSNK1G1 also showed elevated expression level in PC-Ra (Supplementary Fig. 11H). To further investigate the role of CSNK1G1 in impacting tumor metastasis, we utilized the constructed SHC1-OE-ISOHAS and Ctrl-OE-ISOHAS cells and treated them with the CSNK1G1 inhibitor. We then evaluated the cell migration by transwell assay. As a result, inhibiting CSNK1G1 could significantly decrease the cell migration rates increased by SHC1 (Fig. 3L). These results implied that CSNK1G1 participates in tumor metastasis in PC-Ra-HC1 driven by SHC1. We further performed phosphoproteomic analysis between SHC1-OE-ISOHAS treated with or without the CSNK1G1 inhibitor. As a result, the phosphosites of proteins participating in angiogenesis, especially CTNNB1 Ser552, significantly decreased in SHC1-OE-ISOHAS treated with the CSNK1G1 inhibitor (Supplementary Fig. 11I). These observations confirmed the role of CSNK1G1 in phosphorylating CTNNB1 at Ser552. The above results confirmed our assumption that SHC1 could lead to PC-Ra-HC1 tumor migration through phosphorylating CTNNB1 at Ser552 mediated by CSNK1G1.

Moreover, as for the impact of MAPK10 on the phosphorylation of CTNNB1 at Ser675. We constructed the MAPK10 overexpressed vector and transfected it into SW872 cell line (MAPK10-OE-SW872) which showed similar expression patterns with PC-Ra-oHCs. We then treated MAPK10-OE-SW872 cells and treated with or without MAPK10 inhibitor. We also conducted phosphoproteomic analysis, and observed the phosphorylation of proteins such MAPK13, CTNNB1 and MAPK14 which participate in MAPK signaling pathway, showed significantly elevated expression in MAPK10 overexpressed cells and down-regulated in MAPK10 inhibitor treated cell lines (Supplementary

Fig. 1lJ). The above results confirmed our assumption that MAPK10 could lead to PC-Ra-oHCs tumor migration through phosphorylating CTNNB1 at Ser675. All the above observations implied that although both PC-Ra-HC1 and PC-Ra-oHCs showed elevated metastatic rates, they were mediated by diverse phosphorylation cascade, with SHC1 dominant in PC-Ra-HC1 and MAPK10 dominant in PC-Ra-oHCs, respectively (Fig. 3L).

## APEX1 promotes cell proliferation in the PC-Cc

Although both PC-Cc and PC-Ra were associated with poor prognosis, PC-Cc showed an elevated enrichment of the cell cycle pathway (Fig. 3A). To further explore the relationships of prognosis and proteins involved in the cell cycle, we performed Cox proportional hazard model analysis ("Methods") for cell-cycle related proteins and found APEX1 had the most significant correlation with prognosis (hazard ratio = 1.016, $p$ value = 1.18e−3) (Fig. 4A, Supplementary Data 4). In addition, the prognostic value of APEX1 at the mRNA level was also verified in the TCGA sarcoma cohort (Supplementary Fig. 12A)[9]. APEX1 is a key enzyme of the base excision repair (BER) pathway, the function of which is to repair DNA damage caused by oxidizing or alkylating agents[59]. Besides BER and DNA damage repair, APEX1 could also act as a redox coactivator of different transcription factors such as EGR1, p53, and NF-κB[60,61]. It has been reported that APEX1 plays a key role in the progression of a variety of cancers, including gastric, colon, and hepatocellular carcinoma[62–64]. To further explore whether APEX1 promotes poor prognosis through cell proliferation, we performed the correlation analysis and found a positive correlation between abundance of APEX1 and cell proliferation scores in PC-Cc (Pearson's correlation, r = 0.23, p value = 0.01) (Fig. 4B). In concordantly, since KI67 is an indicator of cell proliferation, which has been universally used in pathological screening, we stratified samples into different groups based on KI67 ratios (no: 0%, low: 0−15%, median: 15−30%, high: >30%) and compared expression level of APEX1 in these groups. APEX1 showed a higher abundance in groups with more KI67 ratio (Kruskal−Wallis's test, $p$ value = 1.2e−05), confirming the positive correlation of APEX1 abundance and cell proliferation (Fig. 4B). The positive correlation between the mRNA expression level of APEX1 and cell proliferation scores in two independent cohorts also confirmed the promotion of APEX1 for cell proliferation in LMS and UPS (the two dominant histological subtypes in PC-Cc)[65,66] (Supplementary Fig. 12B). Moreover, to verify the promoting effect of APEX1 for cell proliferation in PC-Cc, we generated APEX1-overexpressed cell lines (RKN-APEX1 and SK-UT-1B-APEX1) utilizing two LMS cell lines (RKN and SK-UT-1B), since LMS was one of the dominant histological subtypes in PC-Cc (29.5%). The results showed that after the overexpression of APEX1, these two cell lines both showed higher cell proliferation rate (RKN: fold change = 2.35, $p$ value = 2.56e-6; SK-UT-1B: fold change = 1.82, $p$ value = 0.024), proving the promoting effect of APEX1 for cell proliferation in vitro (Fig. 4C).

Based on the findings above, we then explored the proteins that might cooperate with APEX1 in the cell cycle pathway and observed proteins including BANF1, RUVBL1, and NPM1 showed the positive correlation with APEX1 (Fig. 4D, Supplementary Data 4). Among these proteins potentially interacting with APEX1, NPM1 (Nucleophosmin1) had the highest functionally combined score (0.992) with APEX1 (Supplementary Fig. 12C, "Methods")[67]. NPM1 involves in several cellular processes, including cell proliferation, DNA repair, and cell senescence, and elevated expression of NPM1 is associated with the progress of several cancers[68–70]. Similar to our findings, the interaction of APEX1 and NPM1 has also been reported by previous research in breast cancer and this interaction contributes to drug resistance in triple negative breast cancer[71]. Moreover, high expression of NPM1 was correlated with poor OS in our cohort (Supplementary Fig. 12F).

For purpose of validating the interaction of APEX1 and NPM1 in PC-Cc, we collected PDCs from patients of PC-Cc with different expression levels of APEX1 (high-APEX1 group: $n$ = 3, average APEX1 abundance = 93.43; control group: $n$ = 3, average APEX1 abundance = 4.64), and performed Immunoprecipitation-Mass Spectrometry (IP-MS) experiments utilizing the APEX1 antibody (anti-APEX1) ("Methods", Fig. 4G). As a result, the comparison between high-APEX1 and the control group identified 48 proteins specifically interacting with APEX1. Among these proteins, NPM1 showed the highest elevated expression in the high-APEX1 group compared with the control group, confirming the close interaction of APEX1 and NPM1 in PC-Cc (Fig. 4E, F). In concordant with APEX1, we also proved the positive correlation between NPM1 and cell proliferation in PC-Cc at the protein level in our cohort and the mRNA level in public independent cohorts (Fig. 4E, G, Supplementary Fig. 12E)[65,66]. Survival analysis further revealed that patients with both high expression levels of NPM1 and APEX1 showed worse prognosis (Cox proportional hazard model, $p$ value = 1.19e−3) (Fig. 4H).

As a phosphoprotein, NPM1 is phosphorylated by various kinases during different stages of the cell cycle, regulating its subcellular localization and functions[72,73]. We hypothesized that phosphorylation of NPM1 might help its interaction with APEX1. End of this assumption, we then evaluated the abundance of NPM1 Ser125 had the highest correlation with APEX1 (Pearson's correlation, r = 0.29, $p$ value = 0.0061) (Fig. 4I, Supplementary Fig. 12G, Supplementary Data 4). In addition, we further explored the upstream kinases of NPM1 Ser125 through kinase-substrate correlation analysis and thus found a significantly positive correlation between NPM1 Ser125 and RIOK1, suggesting that NPM1 Ser125 might be a substrate of RIOK1 (Fig. 4J, Supplementary Data 4). Similar to NPM1 Ser125, RIOK1 also showed an elevated expression in PC-Cc (Supplementary Fig. 12H). We also utilized PDCs from patients of PC-Cc with different expression levels of RIOK1 (high-RIOK1 group: $n$ = 3, average RIOK1 abundance = 1.97; control group: $n$ = 3, average RIOK1 abundance = 0.032) and conducted phosphoproteome approach to depict the functional correlation between RIOK1 and NPM1 Ser125 (Supplementary Fig. 12I). Among all phosphosites participating in the cell cycle, NPM1 Ser125 showed the most elevated abundance in the high-RIOK1 group compared with the control group, verifying the phosphorylation function of RIOK1 on NPM1 Ser125 (Supplementary Fig. 12J). We also constructed RIOK1-overexpressed RKN cell line (RIOK1-OE-RKN) and RIOK1-knocking-down RKN cell line (RIOK1-KD-RKN) to validate the impact of RIOK1 in promoting cell proliferation and phosphorylating NPM1 Ser125. CCK8 cell proliferation assay revealed that RIOK1-OE-RKN showed most significantly elevated cell proliferation rates and RIOK1-KD-RKN had significantly decreased cell proliferation rates (Fig. 4K). The RIOK1 inhibitor could also significantly decrease the proliferation of RIOK1-OE-RKN (Fig. 4K). These observations confirmed the impact of RIOK1 on promoting sarcoma tumor cell proliferation. We then performed comparative proteomic and phosphoproteomic analysis among RKN sarcoma cell lines with different treatments. As a result, besides APEX1, the proteins participating in DNA base excision repair including XRCC1, XRCC4, POLB, as well as cell proliferation index KI67 showed elevated expression in RIOK1-OE-RKN (Fig. 4L, Supplementary Fig. 12K). Consistent with the result of high-RIOK1 PDCs, NPM1 Ser125 was significantly increased in RIOK1-OE-RKN, verifying the correlation of RIOK1 and NPM1 Ser125 (Fig. 4L).

To further investigate the impact of NPM1 phosphorylation on cell proliferation as well as on its interaction with APEX1, we then constructed mutant NPM1 Ser125 plasmid (NPM1^S125A), and transfected it into RIOK1-KD-RKN cells (NPM1^S125A-OE-RIOK1-KD-RKN). The wildtype NPM1 plasmid was also transfected into RIOK1-KD-RKN cells (NPM1-OE-RIOK1-KD-RKN) as controls. Comparing to RIOK1-KD-RKN, NPM1-OE-RIOK1-KD-RKN showed elevated cell proliferation rates, whereas the cell proliferation rates of NPM1^S125A-OE-RIOK1-KD-RKN showed no significant elevation (Fig. 4M). Consistently, the cell proliferation

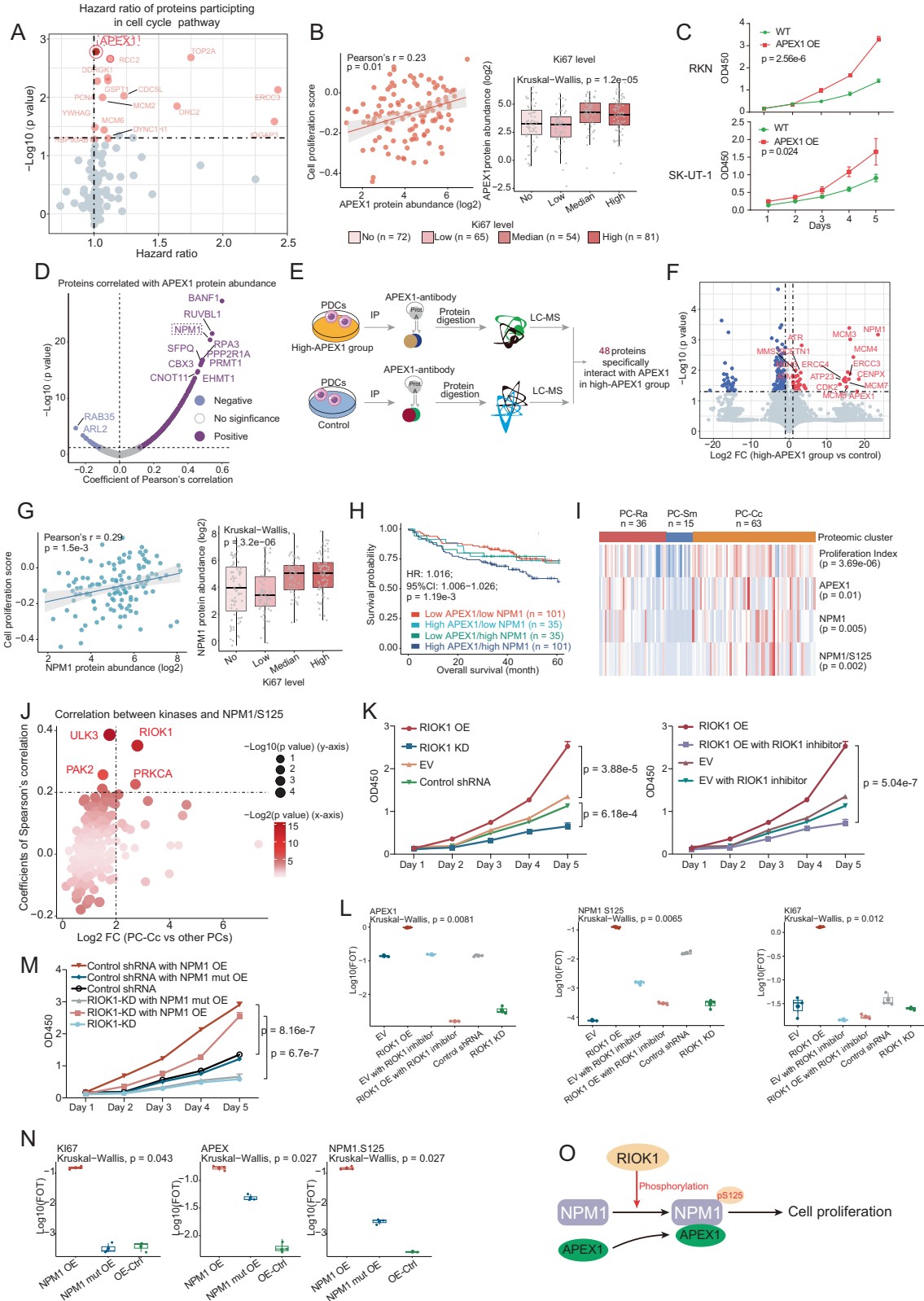

index, KI67 was also observed to be elevated only in NPM1-OE-RIOK1-KD-RKN (Fig. 4N). Meanwhile, comparative proteomics and phospho-proteomic data confirmed the increased expression of APEX1 as well as the increased phosphorylation of NPM1 at Ser125 in NPM1-OE-RIOK1-KD-RKN cells (Fig. 4N). These results indicated the decreased cell proliferation rates led by knocking down RIOK1 could only be rescued by the wildtype NPM1 overexpression, which further emphasized the

role of NPM1 Ser125 phosphorylation in mediating RIOK-dependent regulation of the tumor cell proliferation.

We further performed IP-MS using both NPM1$^{S125A}$-OE-RIOK1-KD-RKN and NPM1-OE-RIOK1-KD-RKN to further illustrate whether the phosphorylation of NPM1 affects its interaction with APEX1 (Supplementary Fig. 12L). As a result, 17 proteins were identified to interact with the wildtype NPM1, but not NPM1$^{S125A}$. Among them, APEX1

**Fig. 4 | Characteristic proteins and the driver pathway of the PC-Cc. A** The volcano plot illustrates prognosis-related proteins which are also enriched in PC-Cc. The Cox proportional hazards model is used for survival analysis. **B** The scatter plot illustrates the correlation between APEX1 abundance and cell proliferation ($n = 122$ samples). The boxplot presents the correlation between APEX1 and Ki67 levels. **C** The plots illustrate the increased proliferation of RKN and SK-UT-1B cell lines after SHC1 overexpression ($n = 3$ biological repeats per group). **D** The volcano plot illustrates proteins significantly associated with APEX1. The Pearson's correlation test is used for associated analysis. **E** Flow chart shows IP-MS steps of primary patient-derived cancer cells (PDCs) isolated from patients with different APEX1 expression. **F** The volcano plot shows up-regulated proteins in high-APEX1 PDCs group. The two-sided student's *t* test is used for the statistical comparison. **G** The plots illustrate the correlation between NPM1 abundance and cell proliferation. Data is presented as Fig. 4B. **H** The Kaplan-Meier curve illustrate the interacted impact of APEX1 and NPM1 on patient's overall survival times. Cox proportional hazards model is used for survival analysis. **I** The heatmap presents the enrichment of APEX1, NPM1 and NPM1 Ser125 in PC-Cc. Wilcoxon rank test is used for statistical analysis. **J** The scatter plot illustrates kinases both correlated with NPM1 Ser125 and

enriched in PC-Cc. The Spearman's correlation is used for associated analysis and the two-sided student's *t* test is used for different expression analysis. **K** Proliferation rate of the RKN cell line is promoted by RIOK1 ($n = 4$ biological repeats per group). **L** Boxplots reveal the influence of RIOK1 on APEX1, Ki67 and NPM1 Ser125 in the RKN cell line ($n = 3$ biological repeats per group). **M** Proliferation rate of the RKN cell line promoted by RIOK1 is associated with NPM1 Ser125 ($n = 4$ biological repeats per group). **N** Boxplots reveal the impact of NPM1 Ser125 mutation on APEX1, Ki67 and NPM1 Ser125 in the RKN cell line ($n = 3$ biological repeats per group). **O** The diagram illustrates the comprehensive mechanism of RIOK1, APEX1, and NPM1 in promoting cell proliferation. For scatter plots in (**B**, **G**), the Pearson's correlation is used for associated analysis. The error band represents the 95% confidence interval of the regression line. For boxplots in (**B**, **G**, **L**, **N**), the middle bar represents the median and the box represents the interquartile range. Bars extend to 1.5× the interquartile range. Kruskal-Wallis's test is used for statistical analysis. For (**C**, **K**, **M**), data is presented as the mean value +/- SE. The two-sided student's *t* test is used for statistical analysis at the 5th day. Source data are provided as Source Data files.

presented the highest abundance, proving that NPM1 Ser125 is the pivotal site for the interaction between NPM1 and APEX1 (Supplementary Fig. 12M). The above results illustrated the potential mechanism that RIOK1 could impact sarcoma tumor cell proliferation through phosphorylating NPM1 which then interacted with APEX1 and promoted tumor cell proliferation accordingly (Fig. 4O).

## Immune infiltration in soft tissue sarcomas

We performed a cell-type deconvolution analysis based on the xCell[74] algorithm to infer the STS TME("Methods")[63]. UPS had the highest mean M1 macrophage score among STS histological subtypes; ES had the highest monocyte infiltrated scores (Supplementary Fig. 13D). Furthermore, we evaluated the prognostic power (Cox proportional hazard model; adjusted *p* value < 0.05) of these infiltrated cells in our cohort (Supplementary Fig. 13B, Supplementary Data 5). The elevated signature of the Th1 cell was associated with the good outcome in DDLPS and oppositely correlated with the poor outcome in LMS and SS. Focusing on macrophages, we observed M1 macrophages were generally associated with favorable outcomes of patients, whereas, the impact of M2 macrophages on prognosis differed among histological subtypes. For instance, the enrichment of M2 macrophages in LMS and MFS was related to poor prognosis, while correlated with favorable prognosis in UPS and DDLPS. These results implied the diversity of immune features within hierarchical clusters. In concordantly, we also observed a similar phenomenon in the TCGA sarcoma cohort[9], implying the diverse impacts of immune cell infiltration on prognosis among different STS subtypes.

Consensus clustering based on inferred cell type proportions defined three immune subtypes with distinct immune and stromal features: IM-S-1 (stroma-enriched: $n = 96$), IM-S-2 (immune-deficiency: $n = 93$), IM-S-3 (immune-enriched: $n = 83$) ("Methods", Fig. 5A, C; Supplementary Fig. 13A, 13E). Survival analysis indicated the immune subtypes significantly differed in OS (log-rank test, *p* value = 0.0013), suggesting that different types of immune cell infiltration could lead to diverse prognostic outcomes (Fig. 5B). The stroma-enriched subtype (IM-S-1), containing mainly PC-Sm samples (52%) (Fig. 5D), showed the highest stromal score and was characterized by multiple types of stromal cells, such as keratinocytes, adipocytes, and mesangial cells (Fig. 5A). In concordant with the features of PC-Sm, this subtype also had enriched complement & coagulation cascade and PPAR signaling pathway (Fig. 5A). The immune-deficiency subtype (IM-S-2), predominantly containing PC-Cc and LMS, represented the lowest immune score (Supplementary Fig. 13E). Consistently, this subtype was enriched with proteins involved in the cell cycle, such as PCNA, CDK2, MCM4, and MCM6. The immune-enriched subtype (IM-S-3), containing most UPS samples, was characterized by the highest immune score,

the presence of CD8 + T cells, CD4 + T cells, macrophages, etc., and increased expression of the immune evasion markers CD274 (PD-L1), CD80 (Fig. 5A, F, Supplementary Fig. 13E). SsGSEA analysis indicated immune-related pathways, including the Toll-like receptor signaling pathway, T cell receptor signaling pathway, and RIG-like receptor signaling pathway were also enriched in this subtype (Fig. 5A). Intriguingly, the samples in IM-S-3 showed a significantly higher metastasis ratio than the other two immune subtypes (Fisher's exact test, *p* value = 6.1e−4) (Fig. 5E).

To confirm our cellular deconvolution analysis by xCell algorithms, we utilized ESTIMATE and CIBERSORT methods to infer each patient's total immune cell infiltration scores and distinctive cell type enrichment scores. We then compared both total immune scores and cell-type specific enrichment scores among the three immune subtypes. The results confirmed the consistent conclusion inferred by the three deconvolution methods. As for the total immune and stroma scores inferred by ESTIMATE confirmed the analysis results of xCell, the immune subtype that harbored the highest immune infiltration score was IM-S-3, and the immune subtype that held the highest stromal scores was IM-S-1 (Supplementary Fig. 14A). Meanwhile, as for cell-type specific enrichment scores among the three immune subtypes, in concordant with the distinctive cell-type enrichments revealed by xCell analysis, CIBERSORT also indicated that the IM-S-3 showed the highest enrichment scores of CD8+ T cell, M1 macrophage and M2 macrophage, and IM-S-2 showed the highest memory B cell enrichment scores (Supplementary Fig. 14B). These results confirmed the feasibility of our proteomic-based xCell deconvolution analysis in predicting the distinctive cell type enrichment in sarcoma tumor microenvironments. We further evaluated the expression of cell-type-specific markers among three immune clusters. As a result, the keratinocyte markers (KRT14, KRT19, and KRT5) were enriched in IM-S-1. The B cell markers (CD19 and IgM) and endothelial cell markers (MCAM, etc.) were significantly elevated in IM-S-2. The CD4+ T cell markers (CD4 and ISG20) and the macrophage markers (CD14, FCGRA, and etc.) were significantly elevated in IM-S-3 (Supplementary Fig. 14C). To further verify our TME deconvolution analysis, we randomly selected several markers for distinctive cell types of each immune subtype (KRT5 & KRT9 for Keratinocyte, CD4 & ISG20 for CD4+ T cells, CD19 & IgM for B cells) and obtained their expression through IHC staining. These markers showed consistent enrichment in immune clusters with related xCell-enriched cell types (Supplementary Fig. 14D). For example, CD4+ T cells had the highest infiltrated scores in the IM-S-3 group. Consistently, the IHC results also presented the highest CD4 and ISG20 expressions in the IM-S-3 group. Meanwhile, IHC staining using CD19 and IgM confirmed the elevated expressions of these two B cell

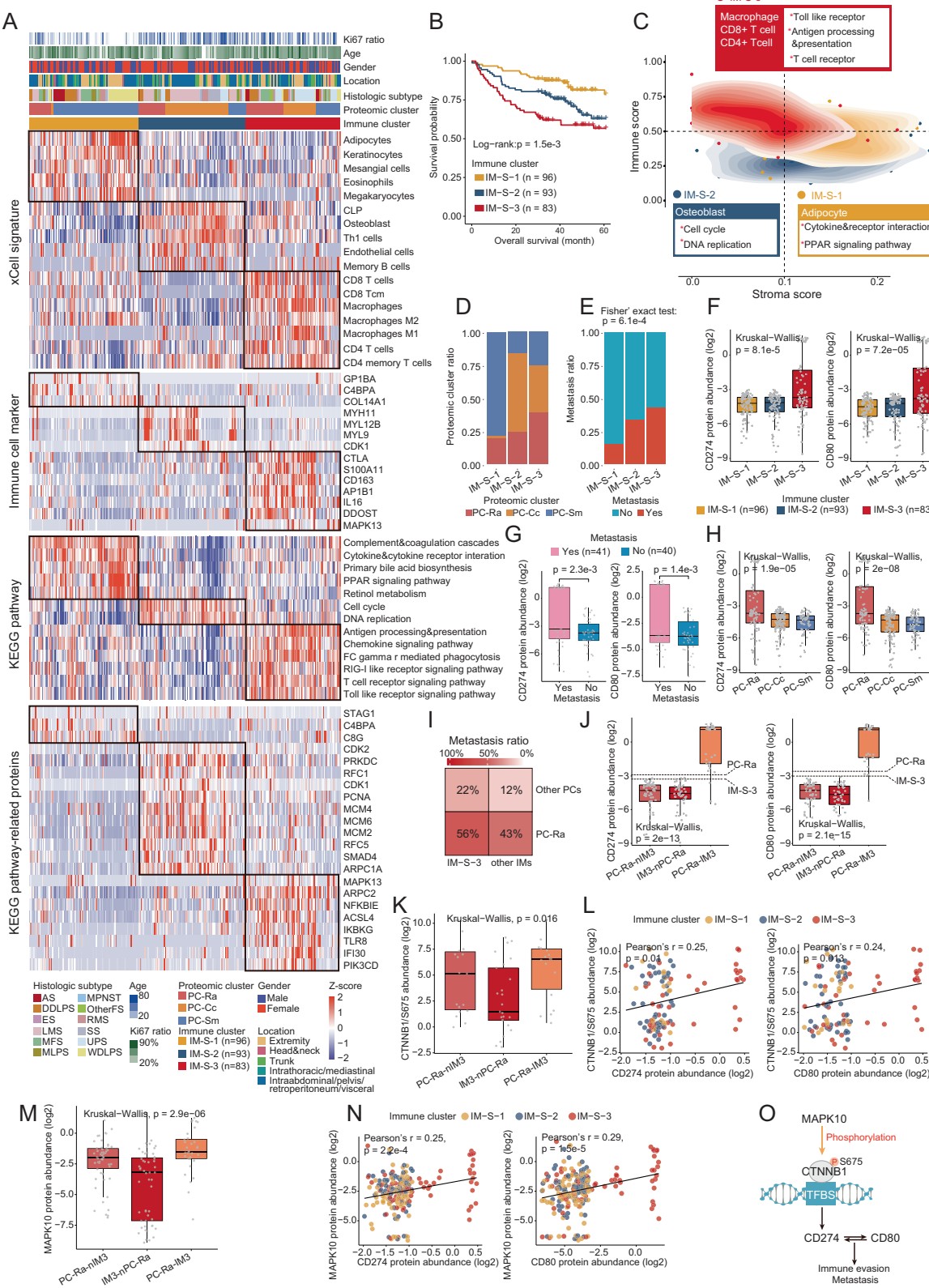

markers in the IM-S-2. Moreover, IHC staining using KRT5 and KRT9 verified the dominant expression of these keratinocytes in IM-S-1. In sum, these IHC staining provided a convincible proof for our TME convolution result.

Previous research has indicated that immune evasion could promote distance metastasis in many cancers, such as colon and breast[75-77]. To further explore whether higher metastatic rates in

IM-S-3 were associated with the elevated expression of CD80 and CD274, we compared expression levels of CD274 and CD80 between samples with and without metastasis in IM-S-3 and found both of them had increased expression levels in the metastasis group (Student's *t* test, CD274: *p* value = 2.3e-3, CD80: *p* value = 1.4e-3), which confirmed the association between CD80/CD274 and metastasis in IM-S-3 (Fig. 5G).

**Fig. 5 | Immune subtypes of STS. A** The heatmap illustrates enriched cell type compositions, proteins and pathways among three immune clusters. **B** The Kaplan-Meier curve illustrate distinguished OS of three immune clusters. Log-rank test is used for survival analysis. **C** The density contour plot of different immune clusters based on stroma and immune scores. **D** The bar plot indicates the proportion of proteomic clusters in each immune cluster. Per group number: IM-S-1 (PC-Ra = 33, PC-Cc = 13, PC-Sm = 50), IM-S-2 (PC-Ra = 21, PC-Cc = 66, PC-Sm = 6), IM-S-3 (PC-Ra = 32, PC-Cc = 43, PC-Sm = 8). **E** The bar plot indicates the proportion of patients with and without metastasis in each immune cluster. The number for each group: IM-S-1 (metastasis = 13, non-metastasis = 73), IM-S-2 (metastasis = 26, non-metastasis = 52), IM-S-3 (metastasis = 29, non-metastasis = 39). Fisher's exact test is used for statistical analysis. **F** Boxplots present CD274 and CD80 abundance in different immune clusters. **G** Boxplots present CD274 and CD80 abundance in patients with and without metastasis of IM-S-3. Per group number: metastasis ($n = 41$) and non-metastasis ($n = 40$). The two-sided student's $t$ test is used for statistical analysis. **H** Boxplots present CD274 and CD80 abundance in different proteomic clusters.

**I** The heatmap shows the metastasis ratio in different proteomic and immune clusters. **J** Boxplot presents CD274 and CD80 abundance in three groups: PC-Ra-nIM3 ($n = 54$), IM3-nPC-Ra ($n = 51$), and PC-Ra-IM3 ($n = 32$). **K** Boxplot presents phosphosite abundance of CTNNB1 Ser675 in three groups: PC-Ra-nIM3 ($n = 18$), IM3-nPC-Ra ($n = 25$), and PC-Ra-IM3 ($n = 18$). **L** Scatter plots present the correlation between CTNNB1 pSer675 and CD274 or CD80 abundance. **M** The boxplot shows log2-transformed MAPK10 protein abundance across PC-Ra-nIM3 ($n = 54$), IM3-nPC-Ra ($n = 51$), and PC-Ra-IM3 ($n = 32$). **N** Scatter plots present the correlation between MAPK10 and CD274/CD80. **O** The diagram illustrates the mechanism that MAPK10 phosphorylate CTNNB1 Ser675 to regulate CD274, resulting in promote metastasis. For (**F, H, J, K, M**), the Kruskal−Wallis's test is used for statistical analysis. For (**F−H, J, K, M**), the middle bar represents the median and the box represents the interquartile range. Bars extend to 1.5× the interquartile range. For (**L, N**), Pearson's correlation is used for associated analysis. Source data are provided as Source Data files.

## The impact of MAPK10 on tumor immune infiltrations

Noticeably, the samples of PC-Ra also featured by high metastasis ratio but were not predominantly overlapped with samples of IM-S-3 (Figs. 3D, 5D). Based on the intersection of PC-Ra and IM-S-3, we found patients belonging to both PC-Ra and IM-S-3 (PC-Ra-IM3) had the highest metastasis ratio (56%) and elevated abundances of CD274/CD80 than PC-Ra-nIM3 (patients belonged to PC-Ra, but not IM-S-3) and IM3-nPC-Ra (patients belonged to IM-S-3, but not PC-Ra) (Fig. 5H–J). Since we have proven that metastasis in PC-Ra depended on activating the CTNNB1 through phosphorylating Ser552 or Ser675 (Fig. 3L), we then analyzed the expression of CTNNB1 Ser552 and Ser675 in three groups: PC-Ra-IM3, PC-Ra-nIM3, and IM3-nPC-Ra. We found a high expression of CTNNB1 Ser675 in PC-Ra-IM3 (Kruskal-Wallis's test, $p$ value = 0.016), but not CTNNB1 Ser552 (Fig. 5K, Supplementary Fig. 13F). We next evaluated the association between CD274/CD80 and the phosphorylation of CTNNB1 Ser675 and observed a significant correlation between the abundance of CTNNB1 Ser675 and the expression of CD274 or CD80 (Fig. 5L). Consistent with our findings, previous research has reported that CTNNB1 could induce transcriptional expression of CD274[78], which meant the phosphorylation of CTNNB1 Ser675 might activate CTNNB1 for inducing transcription of CD274. Furthermore, in concordant with the correlation of CTNNB1 Ser675 and MAPK10 in PC-Ra, we also observed the increased expression of MAPK10 in PC-Ra-IM3 and a significantly positive correlation between MAPK10 and CD274/CD80 (Fig. 5M, N).

We further validated the impact of MAPK10 on tumor immune infiltration using C57/BL6J mice, which usually used as the model for immune microenvironment analysis[79–81]. We constructed xenograft mice models using SW872 cells in which MAPK10 were stably over-expressed or knocked down. Twenty C57/BL6J mice were randomized into four groups ($n = 5$ per group), and separately injected MAPK10-overexpressed and MAPK10-knocked-down SW872 cell lines (OE-MAPK10 and sh-MAPK10) and control cell lines (OE-Ctrl and sh-Ctrl) to form subcutaneous tumors. Tumor size and weight were measured throughout the tumor growth process and tumor volume was calculated. After 4 weeks, mice were sacrificed and tumors were collected for further proteomic and IHC staining analysis. As a result, tumors from mice transplanted with OE-MAPK10-SW872 showed significantly increased immune cell infiltrations, which were evidenced by elevated expression of T cell and macrophage markers (CD4, CD8, and CD163) (Supplementary Fig. 15A). Moreover, the immune checkpoint proteins such as CD274 (PD-L1) and CD80 were also observed to be elevated in OE-MAPK10-SW872 mice (Supplementary Fig. 15A). On the contrary, mice transplanted with sh-MAPK10-SW872 showed obviously decreased immune cell infiltrations, with decreased expression of both immune cell markers and immune checkpoint proteins (Supplementary Fig. 15A). IHC staining of immune markers (CD8, CD163, CD274, and etc.) further confirmed the increased immune cell infiltrations in

OE-MAPK10-SW872 mice and decreased immune cell infiltrations in sh-MAPK10-SW872 mice (Supplementary Fig. 15B). In sum, our data implied MAPK10 might activate CTNNB1 through phosphorylating CTNNB1 Ser675 to induce transcriptional expression of CD274 and then result in a high risk of metastasis in PC-Ra-IM3 patients (Fig. 5O).

To further clarify the relationships among subgroups from different aspects, we performed an integrative analysis of the twelve histological subtypes, six hierarchical clusters, three proteomic clusters, and three immune clusters for STS (Fig. 6A, Supplementary Fig. 16A, B). Noticeably, three hierarchical clusters (HC4, HC5, HC6) and five histological subtypes (RMS, SS, UPS, LMS, DDLPS) were mainly enriched in PC-Cc. For these histological subtypes, almost all of the patients distributed into PC-Cc showed higher proliferation scores, higher APEX1 abundance, and higher NPM1 abundance (Fig. 6B), indicating despite the diverse histological subtypes, the consistent molecular features could still be observed at the proteomic level. Furthermore, combined with immune clustering, we found a high overlap between Pc-Cc and IM-S-2, implying the potential impact of tumors on TMEs. Intriguingly, immune clustering grouped PC-Ra mainly into two immune clusters with different TME features: PC-Ra-IM1 and PC-Ra-IM3 (Fig. 6C). PC-Ra-IM3 showed a higher metastasis ratio with the cooperation of kinase (MAPK10) and immune features (CD274 and CD80), which illustrated that the combination of proteomic clusters and immune clusters could uncover the interaction between tumor biological process and TME to give a detailed division of STS. This conclusion is still tenable in specific histological subtypes and hierarchical clusters. For example, we estimated the proteomic and immune features of HC3, which is mainly clustered into 2 proteomic clusters (PC-Ra and PC-Cc) and 2 immune clusters (IM-S-1 and IM-S-3), showing proteomic and immune environment diversity. According to the distribution of HC3 in proteomic and immune clusters, we classified HC3 into 4 subgroups: HC3-Ra-IM1, HC3-Ra-IM3, HC3-Cc-IM1, and HC3-Cc-IM3. Consistent with the conclusion gotten from the whole proteomic clusters and immune clusters, we also observed the elevated MAPK10 and CTNNB1 S675 in HC3-Ra-IM3 (Supplementary Fig. 16C, D). For immune features, CD274 and CD80 were also elevated in HC3-Ra-IM3 (Supplementary Fig. 16C). In HC3, CTNNB1 S675 still had positive correlations with MAPK10, CD274, and CD80, as the same as the results observed in whole samples (Supplementary Fig. 16E). These results proved the consensus of clustering at different levels.

Moreover, through the combined analysis among the clusters from different aspects, we could also further explore the intrinsic features of STS comprehensively. Specifically, HC5 (UPS) and HC6 (LMS) were both clustered into PC-Cc and featured with fast tumor cell proliferation, which could be confirmed by the elevated cell proliferation index (Supplementary Fig. 16F). Yet, the two HCs showed distinctive immune features. Particularly, the HC5 showed elevated

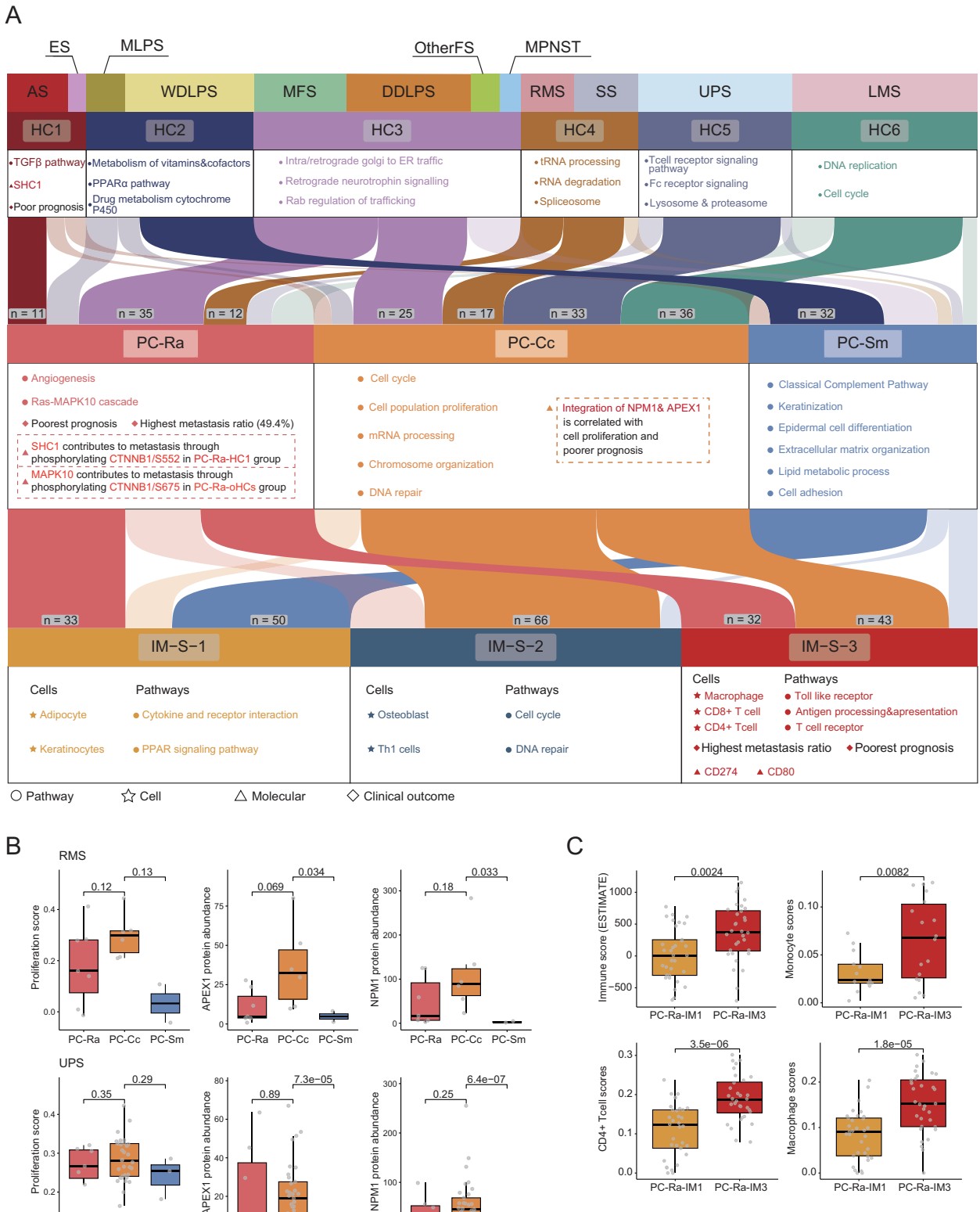

**Fig. 6 | Summary of molecular characteristics based on proteomic clusters.**
**A** The Graphical summary presents the characteristic pathways and major molecular findings of different level subtypes including histological subtypes, hierarchical clusters, unbiased consensus proteomic clusters, and immune clusters. The relationships of these subtypes are also displayed.
**B** Boxplots indicate the proliferation score and APEX1/NPM1 protein abundance across different protein clusters separately in RMS (top) and UPS (bottom). For the number of each group in RMS: PC-Ra (*n* = 7), PC-Cc (*n* = 6),

and PC-Sm (*n* = 2). For the number of each group in UPS: PC-Ra (*n* = 7), PC-Cc (*n* = 33), and PC-Sm (*n* = 3). **C** Boxplots indicate the immune score and signature scores of monocyte/CD4 + T cell/macrophage in PC-Ra-IM1 (*n* = 33) and PC-Ra-IM3 (*n* = 32). For (**B**, **C**), the student's test is used for statistical analysis. The middle bar represents the median and the box represents the interquartile range. Bars extend to 1.5× the interquartile range. Source data are provided as Source Data files.

CD8 + T cell infiltration (Supplementary Fig. 16G). To illustrate the potential mechanism, we compared the protein expression and pathway enrichment scores of immune-related processes between HC5 and HC6. As a result, we observed the dominant enrichment of the TCR signaling pathway in HC5. TCR-related proteins such as PTPN6, NFKBIE, IKBKG, BCL10, etc. were also significantly elevated in HC5 (Supplementary Fig. 16G). These observations suggested that even presenting the same proteomic features, the hierarchical clusters could have different TME features, which supported the necessity of clustering from different levels.

## Discussion

In this study, we establish a Chinese pan-sarcoma cohort including 272 patients and 12 sarcoma histological subtypes. We performed integrate proteomic and phosphoproteomic data to reveal the differentially overrepresented signaling pathways in STS histological subtypes, metastasis-related proteins, and therapeutically relevant subgroups. Our study with this cohort would serve as a complement to the previous multi-omics studies, exhibit a range of clinic-histological spectrums of pan-sarcoma.

Although, in the histological level, WDLPS, MLPS, and DDLPS all belong to the category of liposarcoma, our proteomic-based hierarchical clustering revealed the DDPLS showed the similar proteomic features with MFS than with MLPS and WDLPS. Specifically, the cell proliferation scores were significantly elevated in both MFS and DDLPS. These findings confirmed the previous transcriptomic research that indicated the DDLPS showed comparatively elevated cell proliferation features at mRNA level[82]. Importantly, by performing comparative analysis, we found the RAB signaling pathway was dominantly enriched in DDLPS, and further illustrated that RAB2A might lead to tumor cell proliferation of DDLPS by increasing autophagy process. These results implicated that inhibiting autophagy might be a promising therapeutical option for patients with DDLPS.

MFS was once considered a subset of UPS (myxoid malignant fibrous histiocytoma), but they have been classified as distinct clinical entities based on their different clinicopathologic features[83]. Despite the clinical classification, the molecular diversity of these two subtypes have not been uncovered, thus for now, the treating strategies for UPS and MFS remain the same. Our research revealed that MFS showed enriched transport-related pathways, whereas UPS showed enriched RNA process and metabolism pathways. The diverse proteomic features of UPS and MFS implied the two different histological sarcoma subtypes could be benefited from distinctive therapeutical approaches in the feature.

AS represents a rare group of soft-tissue sarcomas and are aggressive endothelial cell tumors of vascular or lymphatic origin[84,85]. Angiogenesis is thought to be associated with the pathogenesis of AS and is regarded as a potential target for treatment. However, some clinical trials of anti-angiogenesis drugs in AS don't have positive results or only showed limited improved DFS, including bevacizumab (VEGF-A antibody), trebananib (an angiopoietin-1 and −2 peptibody), and sorafenib (VEGFR and B-Raf inhibitor)[86,87]. By performing integrative analysis and functional experiments, our study identified SHC1 as the key regulator, which could elevate actin cytoskeleton reorganization and lead to unfavorable outcomes of AS patients. Meanwhile, hierarchical clustering revealed the similar proteomic features between AS and ES, especially in the elevated SHC1 expression, which implied ES and AS patients might benefit from SHC1 targeting therapy.

The diverse immune features have been reported to be associated with the prognosis of sarcoma patients, but the majority of these researches were either done in animal models or have one layer of omics data. For instance, Magrini and colleagues have utilized transcriptomic data from sarcoma mice model to illustrate that the sarcoma tumor cells could express C3 which could then recruit macrophages through C3-C3aR axis, thus C3 deficiency-associated

signatures of macrophages could lead to favorable prognosis in sarcoma[88]. Since we have also observed elevated C3 protein expression in tumor tissues (Supplementary Fig. 17A), we then investigated the potential association among C3 protein expression, the recruitment of macrophages, and patients' prognosis. As a result, the significantly positive correlation between C3 and macrophage enrichment was observed in our pan-sarcoma dataset and in histological subtypes LMS, SS, WDLPS, and AS (Supplementary Fig. 17B). Further integrative analysis with patients' prognosis revealed that the C3-deficiency macrophage signature based on proteomic was associated with patients' prognosis, consistent with the result from transcriptome previously[88] (Supplementary Fig. 17C). Meanwhile, previous research conducted by Petitprez et al. have utilized transcriptomic data based immune analysis to decipher the immune diversity in pan-sarcomas[89]. They have presented an immune classification of soft tissue sarcomas and identified B cells as a prognostic factor for sarcomas. Taking advantages of our cohort including 12 histological sarcoma subtypes, we further investigated the association between B cells and patients' prognosis. As a result, although we didn't observe a significant association between B cell enrichment and patients' prognosis in our whole pan-sarcoma cohort, we observed that LMS, UPS, MFS, and AS patients with high B cell signatures trended to have longer overall survival times (Supplementary Fig. 17D).

Moreover, to further elevate the clinical applicable of utilizing B cells to prognostic index, we further evaluated the prognostic relevance of the B cell markers' protein expression in our sarcoma cohort and TCGA SARC cohort. As a result, among the 12 B cell markers that have been detected in our dataset, 7 B cell markers showed significant associations with patients' prognosis in our pan-sarcoma cohort. 3 of these B cell markers (PTPRC, CD9, IGLL5) showed consistent prognostic relevance at transcriptomic level in TCGA cohort[9] (Supplementary Fig. 17E). These results implying the potential clinical utilization of these 3 B cell markers for prognostic prediction in feature.

Immune therapy has been applied to many malignancies and presents improved clinical outcomes, such as melanoma. Some clinical studies for immune therapy in STS have been completed and get positive for advanced, metastatic, or unresectable STS[26,90]. Despite the progression of immune therapy in STS, the heterogeneity of TME components within STS histological subtypes makes it a challenge to distinguish patients responding to immune therapy. Intriguingly, based on TME components, we defined a subtype of STS (IM-S-3) with enriched immune infiltration and immune evasion markers (CD274 and CD80) which might respond to immune therapy, especially PD-L1 inhibitors. Besides the heterogeneity in STS histological subtypes, the interaction between tumor biologic process and TME in STS is quite important for the potential combination therapies for sarcoma[91]. Our results implied that the CTNNB1 may contribute to the transcription of CD274 in the immune-enriched group of STS. Meanwhile, MAPK10 participates in this process by phosphorylation of CTNNB1 Ser675. Based on our research, we provide a viewpoint that combined blockade of MAPK10 or CTNNB1 with CD274 might apply in STS cooperatively, which requires further research.

The aim of this study is to provide a proteomic and phosphoproteomic landscape to decipher the sarcomas' heterogeneity, the prognosis-related markers, and abnormally changed biology pathways. There are some limitations due to the sample collection and technology as follows:

1. The sarcoma cohort in this study is single-centered and included only Chinese patients, so the conclusions may lead to potential selection bias. Additional prospective studies are needed to validate our findings in multi-center and cohort of other ethnicities.

2. We found specific subtype-enriched proteins which might be serviceable in early diagnosis and histological subtype detection,

but we couldn't exclude the possibility that this protein could have stemmed from other affected organs or may be indirectly induced by the effects of the tumors on their microenvironment or even systemically. Further experiments or clinical data are necessary complements to validate the roles of this proteins in sarcoma.

3. The proteomic data in this study was generated through bulk proteomic approach from tumor and NAT tissues and couldn't fully reflect the heterogenous tumor regions and the tumor-NAT boundary regions. Integrating single cell and spatial omics would be useful to further explore the intra-tumoral heterogeneity in the future research.

4. The samples in this study were all collected from treat-naïve patients and were all primary tumors without remote metastasis or local relapse. The information about metastasis and local relapse comes from 60-month follow up. The conclusion in this study, that SHC1 and MAPK10 promotes metastasis, required further confirmatory studies on metastatic samples. Other conclusions were also just based on localized diseases, it will have to be determined if these conclusions are also tenable in locally relapsed and metastatic tumors.

## Methods
The present study was carried out in compliance with the ethical standards of Helsinki Declaration II and approved by the Institution Review Board of Fudan University Zhongshan Hospital (B2019-200R).

### Experimental model and subject details
**Sample acquisition.** The two hundred seventy-two formalin-fixed, paraffin-embedded (FFPE) sarcoma tumor tissues and 91 paired NATs were acquired from Zhongshan Hospital, Fudan University from 2010 to 2019. All cases were collected regardless of histological grade or surgical stage. Clinical information of 272 STS patients, including gender, age, tumor location, survival status, and FNCLCC classification, is listed in Supplementary Data 1. Written informed consent was obtained by participants. All the patients received primary resection for sarcomas without any anti-cancer treatments prior to surgery. Postoperative surveillance and treatment were conducted consistently according to Zhongshan Hospital's guidelines. Specifically, 64 patients received chemotherapies, and 27 patients received target therapies after sugary. We compared the overall survival between patients with and without postoperative treatments and observed no significant difference, implying that the patient survival was not impacted by their treatment after surgery (Log-rank test, $p$ value > 0.1). Each sample was assigned a new research ID, and the patient's name or medical record number used during hospitalization was deidentified.

One 4 μm trick slide from each FFPE block was sectioned and stained by hematoxylin and eosin (H&E) for histological evaluation. Specifically, each tumor/ tumor-adjacent sample was checked by three expert pathologists to confirm the sample quality according to the following criteria: For tumor samples: (1) pathologists evaluated and defined tumor area on the slices of FFPE specimens with tumor cell ratio (tumor purity) >70%; (2) the histological subtypes of sarcoma were diagonalized by pathologists according to WHO classification of soft Tissue & Bone tumor[28].

For NAT samples: (1) pathologists evaluated and defined the tumor-adjacent areas on the slices of FFPE specimens with no observed tumor cells; (2) for different histological sarcoma subtypes, NATs were chosen based on tumor locations and the original lineages of tumors, according to WHO classification of soft Tissue & Bone tumor[28]. The specific NATs for different histological sarcomas were presented in Supplementary Data 1. The representative H&E-stained slices showed the regions of tumors with their paired NATs, which confirmed the NAT types for distinctive tumors, and also indicated over 90% of tumor cellular purities for tumor regions, and no tumor cells in NATs

(Supplementary Fig. 2A). Moreover, the same NAT collecting criteria were also utilized by previous published sarcoma studies[9,92–94].

### Cell lines
Eight human sarcoma cell lines were used for functional experiments, including SW-872 (ATCC no. HTB-92), SK-UT-1B (ATCC no. HTB-115), RKN (ITI BioChem, Cat ITI04946), ASM (obtained from Chinese Academy of Science [Shanghai, China]), ISO-HAS (obtained from Bio-Vector Science Lab), VA-ES-BJ (ATCC no. CRL-2138), SU-CCS-1 (ATCC no. CRL-2971), and 93T449 (ATCC no. CRL-3043). The HEK-293T was obtained from Chinese Academy of Sciences and used for quality control. All cell lines were routinely tested for mycoplasma contamination and authenticated by Short Tandem repeat (STR) profiling. Cells were maintained in recommended medium, Roswell Park Memorial Institute-1640 (RPMI-1640, Corning) or Dulbecco's modified Eagle's medium (DMEM, ATCC) supplemented with 10% fetal bovine serum (FBS, Sigma-Aldrich) and 1% penicillin–streptomycin antibiotic (Sigma- Aldrich) and incubated at 37 °C and 5% $CO_2$ in a humidified atmosphere in an incubator.

### Primary cells
Patient-derived primary cell cultures were grown in DMEM supplemented with 1% Penicillin/ Streptomycin (GIBCO), 1X Glutamax (GIBCO), 20 ng/mL EGF and 20 ng/mL bFGF (FGF2). The details for cell isolation and culture were presented in "Method" details.

### Primary sarcoma cell cultures
Surgical specimens were analyzed by pathologists and processed within 3 h after surgical resection. Before the experiment, all experimental tools and instruments were disinfected by ultraviolet ray for at least 20 min. Then the tumor mass was washed in sterile phosphate-buffered saline (PBS) at least twice to totally remove blood. To avoid infection, the tumor mass could be put into the penicillin & streptomycin solution for 30–60 min before washing. Then the tumor mass was cut into 1–2 mm3 sections utilizing the sterile surgical scalpel to facilitate digestion in next step. The sections were added with 40x volume 2 mg/mL type I collagenase (Millipore Corporation, Billerica, MA, USA) and then were digested at 37 °C for 15–20 min. During digestion, the mixture was stirred every 5 min. Then, the mixture was digested overnight at room temperature and was added with DMEM supplemented with 10% fetal bovine serum, 1% glutamine, and 10% penicillin/streptomycin to block the digestion. Then the suspended cells were separated from the undigested mass utilizing a 100 μm sterile filter (CellTrics, Partec, Münster, Germany). Then collect the supernatant through low-speed centrifugation (500–1000 r/min). The supernatant was then added with complete DMEM medium. The cells were counted and seeded at a density of 80,000 cells/cm2.

### Peptide desalination
After digestion of samples, 26 μL 10% formic acid (FA) was added to each tube which was then vortexed for 3 min and centrifuged at 12,000 × $g$ for 10 min. The supernatant was collected in a new 1.5 mL tube with 350 μL extraction buffer (0.1% FA in 50% acetonitrile [ACN]) and extracted by a vortex for 3 min and centrifuging at 12,000 × $g$ for 5 min. The supernatant was transferred into a new tube for drying in a vacuum drier at 60°C. Then, 100 μL 0.1% FA was added for dissolving the peptides, vortexed for 3 min, and centrifuged at 12,000 × $g$ for 5 min to separate out supernatant. The supernatant was collected in a new tube and then desalinated. A pillar filled with two slides of octadecyl (C18) (Empore, Lot #3M-2215) was used for desalination. Before desalination, the C18 slides were activated and balanced by 100 μL 100% ACN twice, 100 μL 50% ACN twice and 100 μL 0.1% FA thrice. For desalination, the supernatant was loaded in the pillar twice, and decontaminated with 100 μL 0.1% FA twice. Lastly, 100 μL elution buffer (0.1% FA in 50% ACN) was added into the pillar for elution twice

and only the effluent was collected for MS. The collected liquid was evaporated to dryness in a vacuum drier at 60°C and stored at −80 °C until LC-MS/ MS analysis.

## Enrichment of phosphorylated peptide

Peptides were extracted and collected using the methods described above. To prevent dephosphorylation, phosphatase inhibitor cocktail 3 (Sigma, #P0044) was additionally added into the tube when digestion. Then phosphopeptides were enriched with High-Select™ Fe-NTA Phosphopeptides Enrichment Kit (Thermo Fisher Scientific, #A32992) following the manufacturer's recommendations. Briefly, the peptides were suspended with binding/wash buffer (provided in the enrichment kit), mixed with the equilibrated resins, and incubated at 21–25 °C for 30 min. After incubation, the resins were washed thrice with 100 μl binding/wash buffer and twice with 100 μl water (MS grade). The enriched peptides were eluted with elution buffer (contained in the enrichment kit) and dried in a vacuum drier at 30 °C.

## ESI-LC-MS/MS analysis

**Proteome and phosphoproteome analysis with liquid chromatography–tandem mass spectrometry.** The Orbitrap Exploris 480 Mass Spectrometer (Thermo Fisher Scientific) is equipped with an Easy nLC-1200 (Thermo Fisher Scientific) and a Nanoflex source (Thermo Fisher Scientific). The peptides were re-dissolved in 12 μL loading buffer (0.1% FA). Peptide samples were loaded onto a trap column (100 μm × 2 cm, homemade; particle size, 3 μm; pore size, 120 Å; SunChrom, USA), separated by a homemade silica microcolumn (150 μm × 30 cm, particle size, 1.9 μm; pore size, 120 Å; SunChrom, USA) with a gradient of 4–100% mobile phase B (80% acetonitrile and 0.1% formic acid) at a flow rate of 600 nL min⁻¹ for 150 min.

LC−MS/MS based proteomic and phosphoproteomic experiments were conducted with Field Asymmetric Ion Mobility Spectrometry (FAIMS). FAIMS voltages were set to −45 V and −65 V, respectively, and other parameters were consistent and set as follows: protein quantification consisted of an MS1 scan at a resolution of 120,000 (at 400 m/z). The automatic gain control (AGC) for full MS and MS/MS was set to 3e6 and 5e4, respectively, with maximum ion injection times of 80 and 22 ms, respectively. The signature was collected and recorded by the Xcalibur (v4.5) software.

## Database searching for proteomic and phosphoproteomic MS raw data

**Peptide identification and protein quantification.** Peptide identification was processed with the one-stop proteomic cloud platform, Firmiana[95] against the *homo sapiens* RefSeq protein database (updated on 04-07-2013) in the National Center for Biotechnology Information. The maximum number of missed cleavages was set to two. The mass tolerance allowed for precursor and production was 20 ppm and 0.05 Da, respectively. The fixed modification was carbamidomethyl (C), and the variable modifications were N-acetylation and methionine oxidation. For quality control of protein identification, a target-decoy-based strategy was applied to control the FDR of both peptides and proteins to <1%. Percolator was used to obtain the probability value (q-value) and validate that the FDR (measured by the decoy hits) of every peptide-spectrum match (PSM) was <1%. Thereafter, all peptides shorter than seven amino acids were removed. The cutoff ion score for peptide identification was 20. The PSMs in all fractions were combined for protein quality control, which was more stringent. The q-values of both target and decoy peptide sequences were dynamically increased until the corresponding protein FDR was <1% using the parsimony principle. Finally, to reduce the false-positive rate, proteins with at least one unique peptide were selected for further investigation. For all the analyses including hierarchical cluster, proteomic subtyping, tumor microenvironment analysis, etc. we utilized a protein matrix

that applied 1% FDR filtering at the protein level for all datasets, which contained 10,118 proteins in total.

For phosphoproteomic data, a label-free based quantification analysis was performed using Proteome Discover (version 2.3). Phosphorylation sites were localized with ptmRS module[96]. Peptide spectrum matches (PSMs) were filtered with 75% localization probability for all phosphorylation sites were included for further analysis. The maximum number of missed cleavages was set to 2. The mass tolerance allowed for precursor and production was 20 ppm and 0.05 Da, respectively. The fixed modification was carbamidomethyl (C), and the variable modifications were oxidation (M), acetylation (protein N-term), and phosphorylation (S/T/Y). For global phosphoproteomic analysis, the FDR at the peptide level and the protein level were also set as 1%.

## MS quantification of proteins and phosphoproteins

For the proteomic data, Firmiana was employed for protein quantification, and both the results and raw data from the mzXML file were loaded. Next, for each identified peptide, the extracted-ion chromatogram (XIC) was extracted by searching against the MS1 based on its identification information, and the abundance was estimated by calculating the area under the extracted XIC curve. For calculating protein abundance, the non-redundant peptide list was used to assemble the proteins by following the parsimony principle. Thereafter, the protein abundance was estimated using a traditional label-free, intensity-based absolute quantification (iBAQ) algorithm, which divided the protein abundance (derived from identified peptide intensities) by the number of theoretically observable peptides[97,98].

For calculating phosphoprotein abundance, the non-redundant phosphopeptide list was used to assemble the proteins by following the parsimony principle. Next, the phosphoprotein abundance was estimated using a traditional label-free, iBAQ algorithm, which divided the protein abundance (derived from the identified peptide intensities) by the number of theoretically observable peptides[97].

## Quality control of the MS data

**Quality control of the MS platform.** For the quality control of MS performance, the HEK293T cell lysate was measured every three days as the quality control standard. The standard was digested and analyzed using the same method and conditions as the STS samples. A pairwise Pearson's correlation coefficient was calculated for all quality control runs in the statistical analysis environment R (version 4.2.3), and the results were shown in Supplementary Fig. 1A. The average correlation coefficients of proteome standards were 0.83-0.95. We also calculated the CV values based on iBAQs and signal-to-noise ratio, to further determine the reproducibility. Fifteen raw data files of HEK293 standards were utilized. We performed a peptide filtering process, the peptides with CV values higher than 30% were filtered out, and the remaining peptides were utilized for protein quantifications. As a result, the median CV based on signal-to-noise ratio at the protein level was 0.18 (Supplementary Fig. 1J–K). These results revealed good reproducibility for repeat experiments with the same samples and demonstrated the consistent stability of the MS platform.

## Proteome and phosphoproteome data analysis

**Data normalization.** Identified proteins and phosphorylated peptides were normalized using the fraction of total (FOT) method, where a relative quantification value is defined as protein iBAQ divided by the total iBAQ of all identified proteins in one experiment. Then, the FOT was further multiplied by 1e6 for presentation ease. Finally, the FOT values were calculated for all samples and used in all subsequent quantitative analyses to correct sample loading differences.

## Missing value imputation

Proteins and phosphosites with <50% missing values in all samples were used for missing value imputation and further analysis. Missing values (NA) were imputed with 1e−6 to adjust extremely small values for avoiding subsequent algorithm analysis that could not handle missing values.

## Differential expression analysis

The proteomic data filtered by <50% missing values ($n = 6251$ proteins) was used as input data for differential expression analysis. Then, the protein expression matrix was used to identify proteins differentially expressed in STS and NAT using Contrasts functions implemented in the limma (v.3.48.3) R package. The $p$ value was adjusted with the Benjamini-Hochberg method and the adjusted $p$ value cutoff was set to 0.05. A total of 1,655 proteins were identified by differential analysis with fold change >2 in STS compared to NAT.

## Survival analysis

Kaplan-Meier survival curves and Cox proportional hazard model were used for survival analysis through the R packages: survival (v3.5-5) and survminer (v0.4.9). Log-rank test was used to test the differential survival outcomes between categorical variables and Cox proportional hazard model was used for continuous variables. p value less than 0.05 was considered as significantly different. Log-rank test also used to test the differential survival outcomes between continuous variables including a given protein, phosphosites, or GSVA score of pathways. Before the log-rank test, the optimal cut point for the selected samples was determined by survminer (v0.4.9) with the algorithm, maxstat (maximally selected rank statistics). Kaplan-Meier survival curves were then calculated (Kaplan-Meier analysis, log-rank test) based on the optimal cut point.

## Proliferation score

Proliferation score was represented by the ssGSEA normalized enrichment scores from the corresponding KEGG gene sets (KEGG cell cycle) calculated above (Pathway projection using ssGSEA).

## Functional enrichment analysis of gene sets and pathways

Pathway over-representation analysis, gene set enrichment analysis (GSEA)[99], and single sample gene set enrichment analysis (ssGSEA) were performed to calculate functional enrichment scores of gene sets or pathways. For pathway over-representation analysis, differentially expressed proteins defined in different clusters or subtypes were filtered as input data. For GSEA, fold changes between different clusters or subtypes were calculated as input data. For ssGSEA, proteomic data matrix was directly used as input data. The input data was subjected to corresponding functions in clusterProfiler (version 4.7.0)[100,101] or GSVA (version 1.46.0) R packages. Gene sets or pathways were gotten from GO[102], KEGG[103], Hallmark[104], Reactome[105], and Msigdbr[106] databases.

## Phosphopeptide analysis-kinase and substrate regulation

KSEA (https://casecpb.shinyapps.io/ksea/) was used to estimate the kinase activities based on phosphosite abundance. KSEA estimates the changes in kinase activity by measuring and averaging its identified substrate amounts instead of a single substrate, which enhanced the signal-to-noise ratio from inherently noisy phosphoproteomic data[107,108]. If the same phosphosites was shared by multiple kinases, it was used for estimating the activities of all known kinases. The use of all curated substrate sequences of a particular kinase minimized the overlapping effects from other kinases, thus improving the precise measurement of kinase activities. The information on kinase−substrate relationships was obtained from public databases including PhosphoSite[109], Phos-pho.ELM[110], and PhosphoPOINT[111]. The information on substrate sites was obtained from previous studies[112] or a KSEA dataset using Motif-X[107].

## Protein−protein interaction network construction

The interaction network among the proteins and phosphoproteins was generated with STRING (version 11.0) (https://string-db.org/) using medium confidence (0.4), experiments and databases as active interaction sources. The network was visualized using Cytoscape (version 3.5.1)[113].

## Immune cell deconvolution

The abundances of 64 different immune and stromal cell signatures for 272 STS samples were computed via xCell[74] (https://xcell.ucsf.edu/). 22 immune cell signatures predefined in CIBERSORT (version 0.1.0) R package were also calculated to validate the results of the xCell. ESTIMATE (version 1.0.13) R package[114] was also used to infer immune and stromal scores based on global proteomic data, with default algorithm parameters.

## Unsupervised clustering of NAT and STS samples

ConsensusClusterPlus (version 1.62.0) R package was utilized to conduct unsupervised consensus clustering of NAT and tumor samples[115]. The following detail settings were used: number of repetitions = 1000 bootstraps; pItem = 0.8 (resampling 80% of any sample); pFeature = 1 (resampling 100% of any protein); clusterAlg = K-means; and distance = Euclidean. As a result, 2 clusters were determined based on the average pairwise consensus matrix within consensus clusters, the delta plot of the relative change in the area under the cumulative distribution function (CDF) curve, and the average silhouette distance for consensus clusters.

Specificity and purity were calculated to estimate the separation of NAT and tumor samples. Specifically, for sample's specificity, the following formula was utilized: specificity = $\max\{N_{c1}/N_{total}, N_{c2}/N_{total}\}$. $N_{total}$ means the whole number of tumors or NAT samples. $N_{c1}$ and $N_{c2}$ mean the samples belonging to cluster1 or cluster2 in $N_{total}$. As for cluster purity, the following formula was utilized: purity = $\max\{C_N/C_{total}, C_T/C_{total}\}$. $C_{total}$ means the whole number of cluster1 or cluster2. $C_N$ and $C_T$ means the numbers of tumors or NATs in $C_{total}$.

## Hierarchical clustering analysis of histological subtypes

R (version 4.2.3) and the R package, factoextra (version 1.0.7) were employed to perform hierarchical clustering of STS samples to find heterogeneity among histological subtypes.

Firstly, 2536 proteins were filtered out with significant variance among histological subtypes (ANOVA analysis, adjust $p$ value <= 0.001). Then, we calculated the mean values of these filtered proteins for each sarcoma histology subtype. The coefficients of Pearson correlation between each two subtypes were calculated utilizing these mean values to represent subtypes' distances (Supplementary Data 2). Next, based on the Pearson's distances, we created the dendrogram with hclust (R basse) and fviz_dend (factoextra) functions. To find the appropriate cluster number (k), we cut the cluster dendrogram at different heights to get the cluster numbers from 2 to 10. The silhouette coefficient was employed to estimate the similarity of samples in one cluster and the difference of samples among different clusters. The silhouette coefficients reached the peak when the cluster number was 5 or 6 (Supplementary Fig. 8C). Combined with survival analysis, we selected 6 clusters as the final result of hierarchical clustering, since the 6 clusters had significantly different prognosis, which revealed the clinical relevance of this clustering standard (Supplementary Fig. 8D).

## Identification of proteomic clusters based on profiling

Consensus clustering was performed using the ConsensusClusterPlus (version 1.62.0) R package[115] with proteins significantly overexpressed in STS ($n = 2703$). The following detail settings were used for clustering: number of repetitions = 1000 bootstraps; pItem = 0.8 (resampling 80% of any sample); pFeature = 1 (resampling 100% of any protein); clusterAlg = "hc"; and distance = "spearman". The number of clustering

was determined by three factors, the average pairwise consensus matrix within consensus clusters, the delta plot of the relative change in the area under the cumulative distribution function (CDF) curve, and the average silhouette distance for consensus clusters. We selected three clusters as the best solution for the consensus matrix since k = 3 provided the clearest separation among the clusters. Additionally, the consensus CDF and delta plots showed a significant increase in the area for k = 3 than that in k = 2, whereas a smaller increase was observed in the area for k = 3 compared with that in k = 4. Based on the evidence above, the STS proteomic data were clustered into three groups.

### Identification of immune clusters based on cell type composition

The abundance of 64 different cell types in 272 STSs was computed via xCell[74]. Consensus clustering was performed based on these cells only detected in at least 50% of patients (adjusted $p$ value < 0.01). This filtering resulted in 53 cell types. To identify sample groups with similar immune/stromal characteristics, consensus clustering was performed using the R packages ConsensusClusterPlus (version 1.62.0) based on the normalized Z-score of these 53 xCell signatures selected above. Specifically, 80% of the original 272 samples were randomly subsampled without replacement and partitioned into three major clusters using the HC (hierarchical cluster) algorithm, which was repeated 1000 times.

### Quantification and statistical analysis

Statistical details of experiments and analyses were noted in the figure legends and supplementary data. Standard statistical tests were used to analyze the association between clinical information and multiomics data. Student's t test, Wilcoxon rank test, One-way ANOVA, and Kruskal-Wallis's Test were used for continuous data; Fisher's exact test was used for categorical data. Log-rank tests and Cox proportional hazard model were used in survival analysis. All statistical tests were two-sided, and statistical significance was considered when $p$ value < 0.05. The correlation between two sets of data was calculated using Pearson's correlation and Spearman's correlation. All the analyses were performed in R (version 4.2.3). A significance level of $p$ value < 0.05 was assumed for all statistical evaluations * $p$ value < 0.05, ** $p$ value < 0.01, *** $p$ value < 0.001.

### Functional experiments

Primers were listed as following. APEX1-F:5′-aacgggccctctagactcgagATGCCGAAGCGTGGGAA-3′

APEX1-R:5′-ctagtccagtgtgtggaattcATGGATCTCCTGCCCCC-3′

### The sequences for shRNA and overexpression were listed as flowing

shMAPK10: CCGGGGAGGAGTTCCAAGATGTTTACTCGAGTAAACATCTTGGAACTCCTCCTTTTTT

shSHC1:CCGGCCGCTTTGAAAGTGTCAGTCACTCGAGTGACTGACACTTTCAAAGCGGTTTTTT

shRIOK1:CCGGGGATGACATTCTGTTTGAAGACTCGAGTCTTCAAACAGAATGTCATCCTTTTTT

The full sequences of MAPK10, RIOK1 and SHC1 were referred to NCBI (Supplementary Data 5).

### Plasmids

The full-length sequences of MAPK10, SHC1 and RIOK1was inserted into pCDNA3.1-FLAG and pCDNA3.1-HA.

### Cell transfection

Plasmid transfections were carried out by either the polyethylenimine (PEI), Lipofectamine 3000 (Invitrogen), or calcium phosphate method. In the PEI transfection method, 500 μL of DMEM (serum-free medium) and the plasmid were placed in an empty EP tube and PEI (three times the concentration of the plasmid) was added into the medium, and followed by vigorous shaking. The mixture was incubated for 15 min. Meanwhile, the cell culture medium was replaced with 2 mL of fresh 10% FBS medium. After 15 min, the mixture was added to the cells, and the medium was replaced after 12 h. After 36 h, the transfection was completed and the cells were consequently treated. In the Lipofectamine 3000 transfection method, 250 μL of DMEM was added to two clean EP tubes and Lipofectamine 3000 was added to one of the tubes and mixed for 5 min. Next, the plasmid and P3000 reagent were added to the other tube, and then added to the medium containing Lipofectamine 3000, mixed, and allowed to stand for 5 min. Meanwhile, the cell culture medium was replaced with fresh 10% FBS medium. After 5 min, the mixture was added to the cells, and the fresh medium was replaced after 12 h. After 36 h, the transfection was completed and the cells were treated. In the calcium phosphate method, the medium was aspirated, 9 mL of fresh DMEM was added, and then the cells were placed back into the incubator for at least 1 h (this is important for balancing the pH for transfection efficiency). DNA in ddH2O (up to 450 μL) was mixed with 500 μL of 2× HEPES buffered saline buffer, and 50 μL of CaCl2 was added drop-by-drop along with shaking. The mixture was incubated on ice for 10 min, chloroquine (2000×, 5 μL) was added to the cells, and the mixture was added drop-by-drop into the plates gently. The plates were swirled and placed back into the incubator. After 5–6 h of transfection, the medium was aspirated and the cells were washed twice with PBS, and fresh medium was added. The cells were collected 24–48 h later.

### Transwell assay

Cells were seeded with serum-free RMPI-1640 medium or DMEM into the upper chamber coated with or without matrigel (corning, Corning City, USA), with RMPI-1640 medium containing 10% FBS added to the lower chamber. Following 24 h incubation, cells that remained in upper membrane were wiped, while cells that migrated or invaded were fixed in methanol, then stained with 0.1% crystal violet and counted under a microscope.

### Analysis of cell proliferation

Total 2000 cells were seeded onto a 96- well plate, and the proliferation activity of the cells was examined by a cell counting kit-8 (CCK-8) assay (Beyotime Institute of Biotechnology, Jiangsu, China) on days 1, 2, 3, 4, 5 post-inoculation. Briefly, 10 μL of CCK-8 solution was added into each well at the corresponding time points. Following incubation at 37 °C for 2 h, the absorbance at 450 nm was measured using a microplate reader (Bio-Rad Laboratories, Inc., Hercules, CA, USA).

### Quantitative RT-PCR

Superscript III RT Kit (Invitrogen) was used with random hexamer primers to produce cDNA from 4 μg of total RNA. GAPDH was used as the endogenous control for all samples. All the primers used for analysis were synthesized by Generay (Shanghai). The analysis was performed by using an Applied Biosystems 7900 HT Sequence Detection System, with SYBR green labeling. The primers sequences are listed as following:

QPCR SHC1-F:5′-CCAGCAGGCAGAGAGCTTTT-3′
QPCR SHC1-R:5′-TCCATGCTACTCCCAGCTCT-3′.

### IP-MS experiment

For IP-MS experiment, primary STS cells were previously selected based on protein expression (APEX1, SHC1, RIOK1, and NPM1) and cultured. The following antibodies were prepared: anti-APEX1 (1:100 dilution), anti-SHC1 (1:100 dilution), anti-RIOK1 (1:100 dilution) and anti-NPM1 (1:100 dilution). Then primary STS cells were lysed on ice in 0.5% NETN buffer (0.5% Nonidet P-40, 50 mM Tris-HCl (pH 7.4),

150 mM NaCl, 1 mM EDTA, and protease inhibitor mixture). After the removal of insoluble cell debris by high-speed centrifugation, protein concentration was then determined by Braford assay. Then 2 mg proteins were incubated with the antibody and rotated overnight at 4 °C. Further, 20 μl Pre-wash magnetic beads (Protein A Magnetic Beads, #73778) were added for another 20 min incubation at room temperature. Pellet beads using magnetic separation rack. Wash pellets five times with 500 μl of 1x cell lysis buffer. Keep on ice between washes. Beads were further washed twice with ddH2O, and three times with 50 mM NH$_4$HCO$_3$. Then, on-bead tryptic digestion was performed at 37 °C overnight. The peptides in the supernatant were collected by centrifugation and dried in a speed vacuum (Eppendorf). Lastly, the samples were redissolved in loading buffer containing 0.1% formic acid before being subjected to MS.

## PDC proteome and phosphoproteome

For the proteomic and phosphoproteomic analysis of PDCs cells, Cells were lysed in lysis buffer (8 M Urea, 100 mM Tris Hydrochloride, pH 8.0) containing protease and phosphatase Inhibitors (Thermo Scientific) followed by 1 min of sonication (3 s on and 3 s off, amplitude 25%). The lysate was centrifuged at 14,000 g for 10 min and the supernatant was collected as whole tissue extract. Protein concentration was determined by Bradford protein assay. Extracts from each sample (500 μg protein) was reduced with 10 mM dithiothreitol at 56 °C for 30 min and alkylated with 10 mM iodoacetamide at room temperature (RT) in the dark for additional 30 min. Samples were then digested using the filter aided proteome preparation (FASP) method with trypsin. Briefly, samples were transferred into a 30kD Microcon filter (Millipore) and centrifuged at $14,000 \times g$ for 20 min. The precipitate in the filter was washed twice by adding 300 μL washing buffer (8 M urea in 100 mM Tris, pH 8.0) into the filter and centrifuged at $14,000 \times g$ for 20 min. The precipitate was resuspended in 200 μL 100 mM NH$_4$HCO$_3$. Trypsin with a protein-to-enzyme ratio of 50:1 (w/w) was added into the filter. Proteins were digested at 37 °C for 16 h. After tryptic digestion, peptides were collected by centrifugation at $14,000 \times g$ for 20 min and dried in a vacuum concentrator (Thermo Scientific). 10% dried peptides were then used for proteomic analysis and 90% peptides were used for further phosphoproteomic analysis, following the protocol described above.

## Immunohistochemistry (IHC)

These antibodies were used for IHC: anti-SHC1 (1:1000 dilution), anti-MAPK10 (1:1000 dilution), anti-PECAM1 (1:1000 dilution), anti-CD36 (1:1000 dilution), anti-IGFBP6 (1:1000 dilution), anti-KRT5 (1:1000 dilution), anti-KRT9 (1:1000 dilution), anti-CD19 (1:1000 dilution), anti-IgM (1:1000 dilution), anti-CD4 (1:1000 dilution) anti-ISG20 (1:1000 dilution), anti-CD8 (1:1000 dilution), anti-CD163 (1:1000 dilution), anti-CD274 (1:1000 dilution).

Formalin-fixed, paraffin-embedded tissue sections of 10 μM thickness were stained in batches for detecting markers and special proteins in a central laboratory at the Zhongshan Hospital according to standard automated protocols. Deparaffinization and rehydration were performed, followed by antigen retrieval and antibody staining. IHC was performed using the Leica BOND-MAX auto staining system (Roche). Slides were imaged using an OLYMPUS BX43 microscope (OLYMPUS) and processed using a Scanscope (Leica).

## In vivo tumorigenesis experiments

Five-week-old male C57/BL6J nude mice were obtained (Shanghai SLAC Laboratory Animal Co., Ltd, Shanghai, China) for in vivo xenografts. Mice were housed in pathogen free, temperature-controlled environment, scheduled with 12–12 h light–dark cycles. The feeding conditions were specific pathogen free animal laboratory with 28 °C and 50% humidity 12/12, providing sufficient water

and diet. Empty-overexpressing-vector, Empty-shRNA-vector, stably MAPK10 overexpressing, and stable MAPK10 knockdown SW872 cell lines ($2 \times 10^6$) were resuspended in PBS and subcutaneously injected into the right flank of C57/BL6J mice (day 0). Tumor size was measured using a caliper, and tumor volume was determined by using the formula: L×W$^2$×0.52, where L is the longest diameter and W is the shortest diameter. This study is under the guidelines of Institutional Animal Care and Use Committee (IACUC), Fudan University. The maximal permitted tumor size is 20 mm in an average diameter for mice, in accordance with guidelines of IACUC. At the end of the experiment, following euthanasia with excessive carbon dioxide (CO2) inhalation, tumors were excised, weighed, and imaged. All procedures were approved by IACUC, Fudan University. Ethical review approval number 2018JS024 was obtained from the Department of experimental animal science, Fudan University.

## Reporting summary

Further information on research design is available in the Nature Portfolio Reporting Summary linked to this article.

## Data availability

All the proteome and phosphoproteome datasets for the cohort study can be accessed through the ProteomeXchange ID: PXD047297. For functional studies, all the raw data can be accessed through the ProteomeXchange ID accession: PXD047429. The entire proteome and phosphoproteome datasets from these functional experiments were uploaded to OMIX and can be accessed through the accession: OMIX005327. The raw files of Annotated gene sets were collected from GO (https://geneontology.org/docs/download-go-annotations/). For molecular signatures database, KEGG database and Reactome database, we got access to them by the R package: msigdbr (version 7.5.1). The public transcriptomic data for validation were downloaded from supplementary files of published articles (https://doi.org/10.1016/j.cell.2017.10.014, https://doi.org/10.1074/mcp.M110.000240, https://doi.org/10.1038/modpathol.3800794). The information of kinase-substrate relationships was available in PhosphoSite [https://www.phosphosite.org/homeAction.action], Phos-pho.ELM [http://phospho.elm.eu.org/dataset.html], and PhosphoPOINT [http://kinase.bioinformatics.tw/]. Software and publicly available resources used in this study were described in "Methods" section. The remaining data are available within the Article, Supplementary Information, or Source Data files. Source data are provided with this paper.

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

## Acknowledgements

This work is supported by National Key R&D Program of China (2022YFA1303200 [C.D.] and 2022YFA1303201 [C.D.]), National Natural Science Foundation of China (32330062 [C.D.] and 31972933 [C.D.]), Program of Shanghai Academic/Technology Research Leader (22XD1420100 [C.D.]), the Major Project of Special Development Funds of Zhangjiang National Independent Innovation Demonstration Zone (ZJ2019-ZD-004 [C.D.]), Shanghai Municipal Science and Technology Major Project (2017SHZDZX01 [C.D.]), the Fudan Original Research Personalized Support Project [C.D.], Young Scientists Fund of the National Natural Science Foundation of China (32201212 [Y.Z.W]), and the China Postdoctoral Science Foundation (2023TQ0063 [H.X.] and 2023TQ0084 [N.X.]).

## Author contributions

Chen Ding, Yingyong Hou, Chen Xu, Weiqi Lu, Yuhong Zhou, Shaoshuai Tang, Yunzhi Wang, Rongkui Luo, Rundong Fang, Yufeng Liu, conceived the work and designed the experiments. Rongkui Luo, Yufeng Liu, Sujie Akesu, Chen Xu, Weiqi Lu, Zixiang Yu and Yingyong Hou collected the clinical information of participants and performed follow-up. Shaoshuai Tang, Yunzhi Wang, Rongkui Luo, Rundong Fang, Yufeng Liu collected the tissue samples. Shaoshuai Tang, Yunzhi Wang, Rundong Fang, Peng Ran, and Hang Xiang. performed the experiments and acquired the MS data. Shaoshuai Tang, Yunzhi Wang, Rundong Fang performed data analysis. Shaoshuai Tang provided technical support on mass spectrometry and quality control of samples. Yexin Tong, Mingjun Sun, Jiacheng Lv, Ning Xu, Zhenmei Yao, Qiao Zhang, Ziyan Xu, and Xuetong Yue made substantial contributions to interpretation of data and provided necessary advices to revise the work. Rongkui Luo, Yufeng Liu, Wen Huang, Jie Huang, Zixiang Yu, and Yuqin Ding, provided support of clinical knowledge and performed immunohistochemistry experiments. Yunzhi Wang, Rundong Fang and Subei Tan drafted the work. Chen Ding, Yingyong Hou, Chen Xu, Weiqi Lu, Yuhong Zhou substantively revised the work. Chen Ding and Yingyong Hou supervised the research.

## Competing interests

The authors declare no competing interests.
