## [Peer Review File · Nature Communications]

REVIEWER COMMENTS

Reviewer #1 (Remarks to the Author): Expert in soft tissue sarcoma functional genomics and preclinical models

In the manuscript entitled "Proteomic characterization identifies clinically relevant subgroups of soft tissue sarcoma", Tang et al perform proteomic analysis of 272 STSs representing 12 major subtypes. They also perform phosphoproteomics for a subset of these samples. Based on this dataset the authors accomplish interesting analyses including hierarchical sample classification, proteomic clustering classification and TME deconvolution and immune-based tumor classification. These analyses uncover interesting correlations. One being the association of SHC1 with poor prognosis in angiosarcoma (AS) and epithelial sarcoma (ES) and its potential role in phosphorylating and regulating AAD2 and CNTBB1. Another being the involvement of APEX1 and NPM1 in promote cell proliferation in one of the proteomic clusters identified (PC-Cc). The classification based on immune signatures defines three immune subtypes with distinctive tumor microenvironments.

The amount of work and analyses performed in this study is impressive. The dataset will be of interest for better understand the diversity of STS but also will provide a valuable resource for clinicians and biologist working on STS. The amount or work and analysis done is a bit overwhelming. Additionally, and although in general the paper is clear, there are some parts where the English language needs to be improved particularly in the discussion. Some comments which I think should be addressed:

1. One of the interesting aspects of this study is the inclusion of normal adjacent tissue for a number of samples. It is clear from the GO presented in supplementary Fig. 2 that this is probably enriched in normal muscle tissue. Since sarcomas can occur in many different locations and the adjacent normal tissue can be quite variable, I wonder if this was taken into consideration. Are the normal tissues primarily corresponding to muscle or is there a combination of more connective tissue types (adipose, muscle, cartilage, etc). The protein was extracted from FFPE material and the regions were probably selected based on histology?

2. It is unclear in Figure 2A how the 6 clusters were defined. Why do the authors define AS, ES, MLPS, WDLPS as two distinct clusters (HC1 and HC2), while MFS, DDLPS, othersFS and MPNST form a single cluster (HC3)? Based on the dendrogram y-axis (height) the HC3 should be further subdivided into three clusters: MFS, DDLPS (as one cluster), othersFS and MPNST. Is that correct? Or which parameters/cutoffs were used for determining the 6 HCs?

3. Line 239: "Meanwhile, *HC1-enriched proteins (student's t test, fold-change > 1.5, adjusted P value <=0.05) participating in pathways correlated with metabolism (as shown in figure 2C) were filtered out.

Is it supposed to be *HC2-enriched proteins? Or did the authors filter out some of the H1-enriched proteins? This is unclear and confusing.

4. The work involves many different analyses touching on some relationships only superficially. I think to further test these models/hypotheses is out of the scope of the paper, however, the authors should take care in their concluding remarks in the results section. The findings support or are suggestive of a particular mechanism but were not functionally tested. The conclusive remarks in the result section should reflect that.

For example:

Line 283: "In sum, the upregulation of SHC1 drives poorer diagnoses of patients diagnosed with AS or ES through promoting actin cytoskeleton reorganization and epithelial cell migration by phosphorylation of ADD2 Ser2".

Although the integration between proteomics and phosphoproteomics is very interesting, and suggestive of a role for SHC1, the data shown does not directly implicate SHC1 in poor prognosis by promoting cytoskeleton reorganization or cell migration. In fact, only cell viability was measured upon SHC1 inhibition at this point in the paper. Also, the authors did to confirm that indeed SHC1 is able to phosphorylate ADD2 and later in the paper it is suggested that SHC1 regulates metastasis by phosphorylating CTNNB1 (instead or in addition?) (Fig.3K, L). This conclusion should be rephrased.

Line 458: "Taken together, our data illustrated that RIOK1 phosphorylates NPM1 on Ser125 to assist the interaction of NPM1 and APEX1 resulting in cell proliferation in PC-Cc"

Again, the authors show increase proliferation by NPM1 overexpression, and that NPM1 interacts with APEX1 but the rest of the data is correlative.

5. The TME deconvolution analysis suggests some interesting relationships. Since this type of analysis is usually done from transcriptomic data, I wonder to what extent is well established for proteome analysis. Was this validated with an alternative deconvolution method or by a couple of IHC markers to validate enrichment of immune cell populations in some of the samples analyzed?

6. The discussion is too long and has some paragraphs that are not well written. See for example the paragraph starting on line 647.

7. Although the last integrative analysis presented in Figure 6 brings together the different aspects analyzed in this study, it is unclear what it means in the perspective of heterogeneity of STS subtypes. Some subtypes are enriched in specific proteomic clusters which are then enriched in different immune signatures, but still, there is a lot of variability on how HC are distributed.

8. There are some typos and some sentences are not well constructed or are unclear. This is particularly noticeable in the discussion. Some examples below:

Line 73. "A potential explanation is that these mechanisms could not reflect the functional effects, as they reside many regulatory layers away from the protein."

Line 106. "It is necessarily required for immune therapy that more detailed information about the characteristics of immune infiltration and the effective immune components."

Line 594. "When considering targeting the molecular in the TGF β signaling pathway (such as SHC1), ES might have a similar response with AS."

Minor comments:

1. Line 257: "We found the activity level of two pathways enriched in HC1, actin cytoskeleton reorganization (Pearson's correlation, $r = 0.21$ pvalue = 0.0049) and epithelial cells migration (Pearson's correlation, $r = 0.22$ p value = 0.0027), changed *tightly followed the abundance variation of SHC1 (Figure 2F).

I would not say there is a *tight correlation between SHC1 abundance and Epithelial cell migration. It is just a correlation.

2. Fig. 2K it would be better to show in the plot the IC50 of all cell lines individually and they are only 6.

3. The literature references are not always correct. For example, reference for CellX should be 62 (not 63 as mentioned in the text).

Reviewer #2 (Remarks to the Author): Expert in tumour immunology and immune landscapes in sarcoma

The present study undertakes a comprehensive proteomic profiling of 272 STS patients representing 12 major subtypes.

The authors identify six subtypes on the base of hierarchical classification, three subtypes based on proteomic analysis and three subtypes based on immune signatures. For some clusters they identified some mechanism/s relevant for patient prognosis. Interestingly, some of the main mechanisms identified with bioinformatics approaches are verified by wet laboratory experiments.

The study provides a valuable proteomic resource for the scientists working on sarcomas. The study is correctly written, although the logic of the analyzes carried out is not always fluent and sometimes it is difficult to follow.

Moreover, there are some concerns:

1) The three main clustering analysis should be performed also taking into account the anatomical site distribution and the therapies applied to the patients, in order to verify if the clustering may be influenced by the location of the tumor or by the therapy.

2) in figure 5A complement and coagulation cascade pathways are enriched in the IM-S-1 cluster corresponding to the stroma-enriched subtype and B cells in the IM-S-2. Results already published on the role of complement activation and B cells in sarcomas (doi: 10.1038/s43018-021-00173-0 and doi: 10.1038/s41586-019-1906-8) should be mentioned and discussed. Are the main findings of these two papers true by proteomic point of view? For example, is the C3aR or complement soluble proteins/receptors expression associated with M2-like macrophages and/or UPS patient survival? Are B cell markers associated with increased overall survival? Do they correlate with metastasis?

3) The authors should discuss some limitations of the study, such as:

- the requirement of future validation in independent cohorts.

- considering the extensive intra-tumoural heterogeneity, the inability of bulk proteomic approach to dissect the contribution of distinct heterogenous tumour regions.

- the study is based on localised disease, thus it will have to be determined if these findings will be true also for locally relapsed and metastatic tumours.

Reviewer #3 (Remarks to the Author): Expert in MS-based cancer proteomics

Comments on “Proteomic characterization identifies clinically relevant subgroups of soft tissue sarcoma” by Tang et al.

The authors present proteome data from 272 soft tissue sarcoma tissues and 91 matched tumor-adjacent tissues (total of 363 samples). In addition, phosphoproteome data were generated from 138 sarcoma and 24 tumor-adjacent tissues. Data analysis is based on clustering the data, extract functional predictions from the clusters, and follow-up with some cell line experiment to understand the role of top-scoring proteins in the specific functional categories. The authors are - in general – overstating the evidence from the molecular mechanisms they are interrogating (see comments). Overall, I did not find the study to be very exciting. I think that Nature Communications is a good place for resource-style papers like this, and proteomics studies on soft tissue sarcoma have the potential to help us better understand the diseases and to identify new treatment strategies. Also, 361 sample is a quite large number. What I am missing is evidence that proteomics is adding crucial information beyond what we know about the disease. I also think that the follow-up experiments need more depth. I am on the fence regarding recommending to consider a publication after major revisions, but I am happy to look the manuscript after the below comments have been addressed.

(1) The authors state that 15,552 proteins were identified across all samples with an average of 5,593 proteins being quantified per sample on average. It is very unlikely that 5.5 k proteins per sample using unfractionated sample leads to a total of > 15 k proteins across 363 samples. I wonder if the false-discovery filtering at the protein level was done for each individual sample but not for all datasets combined. It is the latter, that should have been done. Merely filtering for each individual run will greatly inflate the protein FDR for the entire dataset (as false assignments will be different for each run). It is also not clear if a parsimony filtering was used on the identified proteins. This should also be done the combined dataset. The same question applies to the phosphoproteomics analysis: was the filtering done on the combined dataset (which it should have been) or only on each individual dataset?

(2) Peptides/proteins were quantified using a label-free approach (iBAQ). Reproducibility is shown in Supp Fig 1 A. I would like to see the median CV across all the HEK standard samples as well as the CV in dependence to the signal-to-noise ratio.

(3) Supp Fig 2 A. The PCA plot shows quite an overlap of NAs and tumor samples. It would be great to see a unsupervised clustering of NAs and tumor sample and some cluster purity measurement to evaluate the separation of tumor and normal samples.

(4) What criteria were used to define the clusters (HC1-6)? This is not clear based on the dendrogram alone. The dendrogram implies that there was very clean clustering histological subtypes. I am missing a plot showing how well the subtypes were separated from each other using unsupervised clustering (see also comment 3).

(5) line 250. A correlation between TGFbeta proteins and SHC1 does not necessarily mean that SHC1 plays a key role in TGFbeta signaling. It may suggest that it plays a role, but this needs more evidence. This should be re-worded.

(6) Line 283: In sum, None of that is shown with enough evidence. The language should be toned down. Higher kinase expression does not necessarily mean higher kinase activity. Did ADD2 S2 phosphorylation level drop with inhibition of SHC1? How specific is the inhibitor. What is the kinase phosphorylating ADD S2?

(7) Fig 3 and Supp Fig 6: Is the inhibition of SHC1 and MAPK10 affecting the phosphorylation levels at CTNNB1Ser552 and Ser675?

(8) Fig 7P and line 457. There is lots of evidence missing for RIOK1 phosphorylating NPM1 and thereby regulating the interaction of APEX1 and NPM1. Does inhibition/KD of the kinase affect the phosphorylation level (phosphoproteomics, WB)? Does the inhibition affect the interaction of the 2 proteins (IP-MS, WB)? Does it affect the co-regulation of the two proteins (proteomics)?

(9) Fig 7O and line 527: Evidence is missing. Does inhibition/KD of MAPK10 affect the CTNNB1 Ser657 phosphorylation level. Does the inhibition of MAPK10 in cells derived from the according strain affect immune infiltration (xenograft model)?

(10) As the control samples are matched tumor-adjacent tissue, the authors may consider comparing sarcoma and control tissue in a patient-specific manner to better understand tumor/normal differences (does it matter if I normalize the sarcoma proteome by the adjacent tissue proteome for each patient, rather than compare all control samples with all sarcoma samples?).

Minor comments:

(a) The KSEA algorithm should be cited in the main text when stating that kinase-substrate networks were generated.

**Reviewer #1 (Remarks to the Author): Expert in soft tissue sarcoma functional**
**genomics and preclinical models**

**In the manuscript entitled “Proteomic characterization identifies clinically**
**relevant subgroups of soft tissue sarcoma”, Tang et al perform proteomic analysis**
**of 272 STSs representing 12 major subtypes. They also perform**
**phosphoproteomics for a subset of these samples. Based on this dataset the authors**
**accomplish interesting analyses including hierarchical sample classification,**
**proteomic clustering classification and TME deconvolution and immune-based**
**tumor classification. These analyses uncover interesting correlations. One being**
**the association of SHC1 with poor prognosis in angiosarcoma (AS) and epithelial**
**sarcoma (ES) and its potential role in phosphorylating and regulating AAD2 and**
**CNTBB1. Another being the involvement of APEX1 and NPM1 in promote cell**
**proliferation in one of the proteomic clusters identified (PC-Cc). The classification**
**based on immune signatures defines three immune subtypes with distinctive tumor**
**microenvironments.**

**The amount of work and analyses performed in this study is impressive. The**
**dataset will be of interest for better understand the diversity of STS but also will**
**provide a valuable resource for clinicians and biologist working on STS. The**
**amount or work and analysis done is a bit overwhelming. Additionally, and**
**although in general the paper is clear, there are some parts where the English**
**language needs to be improved particularly in the discussion. Some comments**
**which I think should be addressed:**

**Response:**

**We are grateful for the constructive comments that the reviewer has provided, which**
**truly help us in improving our work. In this revision, according to the reviewer’s**
**comments, we have conducted a deeper bioinformatic analysis, performed a series of**
**functional experiments, and also revised the manuscript carefully. We have provided**
**specific point-to-point response as follows:**

**1. One of the interesting aspects of this study is the inclusion of normal adjacent**
**tissue for a number of samples. It is clear from the GO presented in supplementary**
**Fig. 2 that this is probably enriched in normal muscle tissue. Since sarcomas can**
**occur in many different locations and the adjacent normal tissue can be quite**
**variable, I wonder if this was taken into consideration. Are the normal tissues**
**primally corresponding to muscle or is there a combination of more connective**
**tissue types (adipose, muscle, cartilage, etc). The protein was extracted from FFPE**
**material and the regions were probably selected based on histology?**

**Response:**

We appreciate the reviewer's constructive comments. We apologize for the unclear
description of the sample collection and assessments and the neglected presentation of
how we performed a comparative analysis between tumors and normal tissues adjacent
to tumor (NAT). To systematically respond to the comment, we divided the responses
into two parts:

**1. The criteria for sample collection and assessment**

In this study, for tumor samples, 272 formalin-fixed, paraffin-embedded (FFPE)
sarcoma tumor tissues and 91 paired NATs were acquired from Zhongshan Hospital,
Fudan University from 2010 to 2019. One 4 µm thick slide from each FFPE block was
sectioned and stained by hematoxylin and eosin (H&E) for histological evaluation.
Specifically, each tumor/ tumor-adjacent sample was checked by three expert
pathologists to confirm the sample quality according to the following criteria:

For tumor samples: (1) pathologists evaluated and defined tumor area on the slices of
FFPE specimens with tumor cell ratio (tumor purity) > 90%; (2) the histological
subtypes of sarcoma were diagonalized by pathologists according to WHO
classification of soft Tissue & Bone tumor (*Adv Anat Pathol*, PMID: 32960834).

For NAT samples: (1) pathologists evaluated and defined the tumor-adjacent areas on
the slices of FFPE specimens with no observed tumor cells; (2) for different histological
sarcoma subtypes, NATs were chosen based on tumor locations and the original

lineages of tumors, according to WHO classification of soft Tissue & Bone tumor (*Adv*
*Anat Pathol*, PMID: 32960834). The specific NATs for different histological sarcomas
were presented in **Table RL1**. The representative H&E-stained slices showed the
regions of tumors with their paired NATs, which confirmed the NAT types for
distinctive tumors, and also indicated over 90% of tumor cellular purities for tumor
regions, and no tumor cells in NATs (**Figure RL1**). Moreover, the same NAT collecting
criteria were also utilized by previous published sarcoma studies (*Cell*,
PMID:29100075; *Nature Genetics*, PMID:20601955; *Curr Treat Options Oncol*,
PMID: 35171456; *Cancer Research*, PMID:17638873).

**Table RL1. The histological subtypes of sarcoma and tissue types of their paired NATs**

Tumors	NATs
WDLPS	Lipid tissue
MLPS	Lipid tissue
DDLPS	Lipid tissue
AS	Connective tissue
UPS	Connective tissue
MFS	Connective tissue
otherFS	Connective tissue
LMS (gastrointestinal tract & uterus)	Smooth muscle tissue
LMS (other organs)	Connective tissue
RMS	Skeletal muscle tissue
MPNST	Nerve tissue
SS	Connective tissue
ES	Connective tissue

Figure RL1. The criteria for sample collection and assessments

(A) H&E-stained slices presents the regions of tumor and paired NATs. Different histological
 subtypes have distinguished tissue types of NATs.

**2. The molecular features of tumors and NATs**

In our previous version, to investigate the common molecular features of sarcoma
 tumors and NATs, we conducted a comparative analysis between all tumor tissues and
 all NATs. As a result, Gene Ontology (GO) enrichment analysis based on proteomic
 data revealed that proteins of some classical oncogenic pathways, including RNA
 splicing, NF-kappaB signaling, JNK cascade, and cell growth were significantly
 elevated in tumor samples, whereas, the protein participating in ATP metabolic process,
 glycogen metabolic process, and actin filament organization were decreased in tumor
 samples.

In the revision, following the reviewer's suggestion, we performed pair-wised tumors
 and NATs comparative analysis among 12 histological sarcoma subtypes to clearly
 elucidate the molecular features of different histological tumor subtypes and features
 of their corresponding NATs. As a result, besides the common features of sarcoma
 tumors such as cell cycle, NF-kappaB signaling pathway, and the general characteristics
 of NATs such as ATP metabolic process, we observed the distinctive features of

different histological sarcoma tumors and their corresponding NATs (**Table RL2**). For
instance, the pathways enriched in WDLPS included the VEGFA & VEGFR2 signaling
pathway and HOXA1 target signaling pathway, whereas pathways enriched in its pair-
wised NATs (lipid tissues) included organic acid catabolic process, carboxylic acid
catabolic process, and ATP synthesis coupled electron transport. Meanwhile, the
pathways enriched in RMS include MYC targets up, signaling by interleukins and DNA
replication, while, pathways enriched in their pair-wised NATs (skeletal muscle tissues)
were muscle system processing, muscle contraction, etc. Along with these findings, the
pathways dominantly enriched in MPNST were MAPK cascade, P53 regulation
pathway, and cell cycle, whereas pathways enriched in its pair-wised NATs (nerve
tissues) were intermediate filament organization and collagen fibril organization. The
specific pathways for distinctive tumors and NATs were listed as follows:

**Table RL2. Significantly enriched pathways in NAT tissue and tumor tissue for each**
**histological subtype**

	NAT		Tumor	
	Pathway	p value	Pathway	p value
DDLPS	intermediate filament organization	7.61E-10	cell cycle	2.55E-07
	cellular lipid catabolic process	5.08E-09	MTORC1 mediated signalling	2.83E-07
	lipid catabolic process	3.09E-08	PI3KCI pathway	7.89E-06
	lipid modification	9.68E-07	AKT targets	1.18E-05
MLPS	actomyosin structure organization	3.71E-07	MYC targets up	1.07E-12
	myofibril assembly	3.01E-06	complement and coagulation cascades	1.18E-10
	striated muscle cell development	3.19E-06	PTEN regulation	2.75E-08
	muscle contraction	8.83E-06		
LMS	actin filament capping	1.22E-07	MYC targets up	7.01E-28
	regulation of actin filament depolymerization	1.43E-07	PDGFRB pathway	1.14E-21
	regulation of actin filament polymerization	4.03E-06	metastasis up	1.19E-20
	regulation of actin filament length	4.57E-06	focal adhesion	2.66E-20
UPS	energy derivation by oxidation of organic compounds	3.86E-29	MYC targets up	4.33E-38
	ATP synthesis coupled electron transport	3.64E-27	B cell receptor signaling pathway	2.49E-11
	electron transport chain	6.5E-27	T cell receptor signaling pathway	1.53E-08
	ATP metabolic process	2.14E-20	cell cycle M to G1	2.59E-08
WDLPS	organic acid catabolic process	6.41E-25	transport to the golgi and subsequent modification	5.07E-24
	carboxylic acid catabolic process	6.41E-25	VEGFA&VEGFR2 signaling pathway	4.2E-23
	ATP synthesis coupled electron transport	1.54E-23	HOXA1 targets up	1.35E-13
	purine ribonucleotide metabolic process	1.33E-19		
MFS	muscle contraction	2.86E-23	protein targeting to membrane	5.11E-42
	muscle system process	5.83E-23	MYC targets up	8.33E-32
	myofibril assembly	5.91E-13	TGFB1 targets up	2.16E-17
	muscle cell development	2.32E-12	PI3KCI pathway	1.21E-08
ES	primary alcohol metabolic process	4.81E-11	selective autophagy	5.19E-11
	olefinic compound metabolic process	1.59E-09	Rho GTPase effectors	4.14E-05
	fatty acid metabolic process	3.13E-09	Notch1 targets up	1.69E-04
	hormone metabolic process	3.96E-09	TGFB1 targets up	6.14E-04
SS	positive regulation of cell adhesion	1.8E-20	ribosome biogenesis	8.01E-26
	leukocyte mediated immunity	5.72E-18	translational initiation	2.94E-23
	reactive oxygen species metabolic process	2.34E-17	DNA replication	5.32E-14
	regulation of cell-cell adhesion	2.91E-17	DNA conformation change	4.14E-10
AS	energy derivation by oxidation of organic compounds	3.8E-13	stabilization of P53	3.69E-09
	electron transport chain	1.09E-11	VEGFA&VEGFR2 signaling pathway	4.78E-07
	aerobic respiration	5.57E-11	collagen fibril organization	5.2E-06
	oxidative phosphorylation	8.62E-11	TGFβ receptor signaling in EMT	4.92E-05
MPNST	intermediate filament organization	1.57E-54	MAPK cascade	1.16E-08
	electron transport chain	3.16E-12	P53 regulation pathway	3.27E-06
	extracellular matrix organization	2.59E-08	cell cycle	2.60E-04
	collagen fibril organization	7.36E-08		
otherFS	intermediate filament organization	1.63E-26	BRCA1 targets up	1.43E-16
	muscle contraction	3.95E-20	MYC targets up	1E-15
	muscle system process	2.46E-18	TGFB1 targets up	2.85E-15
	cell-substrate adhesion	9.71E-15	DNA replication	1.91E-07
RMS	muscle system process	1.26E-16	MYC targets up	1.27E-33
	muscle contraction	8.34E-16	signaling by interleukins	1.81E-21
	muscle cell development	1.26E-08	interleukin1 signaling	1.11E-17
	muscle filament sliding	1.68E-07	DNA replication	5.72E-08

The above results indicated that besides the common features of sarcoma tumors (cell
proliferation, MAPK signaling pathways, etc.) and NATs (ATP metabolic process),
pair-wised comparative analysis revealed specific features for distinctive histological
sarcomas and their corresponding NATs. In the revision, besides the common features
of tumors and NATs which have been described in our previous version, we have added
the distinguished features of different histological sarcoma tumor types and their
corresponding NATs in the “**Result**” section, on lines 127-144 and 201-213. Moreover,

we carefully checked our previous version of the manuscript, and in our research, the
omics data of NATs were utilized to investigate the differential expression features of
sarcoma tumors and NATs (supplementary figure2 in the previous version), thus we
have updated the supplementary figure 2 accordingly. Also, we have added the criteria
for selection and assessment of NATs in the “**Method**” section on lines 933-960 in the
revised manuscript.

**2. It is unclear in Figure 2A how the 6 clusters were defined. Why do the authors**
**define AS, ES, MLPS, WDLPS as two distinct clusters (HC1 and HC2), while MFS,**
**DDLPS, othersFS and MPNST form a single cluster (HC3)? Based on the**
**dendrogram y-axis (height) the HC3 should be further subdivided into three**
**clusters: MFS, DDLPS (as one cluster), othersFS and MPNST. Is that correct? Or**
**which parameters/cutoffs were used for determining the 6 HCs?**

**Response:**

We thank the reviewer for the critical comment. We apologize for the unclear
presentation of the clustering cutoffs and details in our previous manuscript. To
systematically respond to the comment, we will address this comment from 3 aspects:

- 1. The process to create the dendrogram;
- 2. The criteria to determine the cluster number;
- 3. Biological insights based on hierarchical clusters.

**1. The process to create the dendrogram**

To investigate the intrinsic common features of STS histological subtypes, we
employed hierarchical clustering on the 12 STS histologic subtypes. R (version 4.2.0)
and the R package “factoextra” (version 1.0.7) were utilized for data process and
visualization.

Firstly, we performed ANOVA analysis to filter proteins with high variable values
among different histology subtypes. The protein expression matrix had been processed
as described in the “**Method**” section of the manuscript. 2536 proteins were finally
filtered out with significant variance among histological subtypes (p-value ≤ 0.001).

Then, we calculated the mean values of these filtered proteins for each sarcoma
histology subtype. The “Pearson” distances between each two subtypes were calculated
utilizing these mean values (**Supplementary Table 2**). Next, based on the “Pearson”
distances, we created the dendrogram with “hclust” and “fviz_dend” functions in R
using default parameters (**Figure RL2A**).

**Figure RL2. Process and details of hierarchical clustering**

(A) The cluster dendrogram of 12 histological subtypes of sarcoma

2. The criteria to determine the cluster number

The cluster number of hierarchical clustering is determined by the height where the
cluster dendrogram is cut. To find the appropriate cluster number (k), we cut the cluster
dendrogram at different heights to get the cluster numbers from 2 to 10 (**Figure RL2B**).
Referring to previous research, we utilized the silhouette coefficient to estimate the
similarity of samples in one cluster and the difference of samples among different
clusters. The silhouette coefficients reached the peak when the cluster number was 5 or
6 (**Figure RL2C**).

To further investigate the clinical availability of our hierarchal cluster, we evaluated the
association between hierarchal clustering with patients’ prognosis. As a result, when the
cluster number is 6, patients belonging to different clusters presented distinguished
overall survival time (log-rank test, $p < 0.03$) (**Figure RL2D**), suggesting its potential
clinical utilization. Therefore, we cut the dendrogram at 0.95 and clustered the 12
histological subtypes of sarcoma into 6 subgroups: HC1 (AS and ES), HC2 (MLPS and
WDLPS), HC3 (MFS, DDLPS, and otherFS), HC4 (RMS and SS), HC5 (UPS), and

HC6 (LMS) (Figure RL2E).

**Figure RL2.** (B) The circled cluster dendrograms of sarcoma histological subtypes with cluster

numbers from 2 to 10. (C) The scaled mean values of silhouette coefficients for different cluster

numbers. (D) Kaplan-Meier curves for overall survival times when cluster number is 5 or 6.

**Figure RL2 (E)** Cluster dendrogram for hierarchical clustering when cluster number is 6

**3. Biological insights based on hierarchical clusters**

Besides clinical availability, our HC clustering showed strong biological relevance,
 each subgroup showed distinctive biological features, helping to uncover the intrinsic
 common features of different histological subtypes belonging to the same hierarchical
 cluster. Particularly, in our previous version, we found that HC1 contains AS and ES,
 both of which could be distinguished from other clusters with elevated expression of
 SHC1-TGF β signaling pathways.

In the revision, we conducted further analysis to investigate how hierarchical clusters
 could decipher the common features and heterogeneity among 12 histological subtypes
 of sarcoma. As a result, we found that our hierarchical clustering divided the lipid
 sarcoma (WDLPS, MLPS, and DDLPS) into two clusters. Particularly, DDLPS were
 clustered together with fibrosarcomas (MFS and otherFS) and MPNST in HC3.
 WDLPS and MLPS were clustered into another cluster (HC2). Considering different
 differentiation levels of WDLPS, MLPS, and DDLPS, these findings revealed the
 difference of tumor differentiation within lipid sarcomas might lead to the diverse
 molecular features between DDLPS and WDLPS, further implying that the degree of
 tumor differentiation might serve as an important factor in determining the molecular

features of sarcomas within lipid sarcomas. Because DDLPS is more metastatic and
 proliferative than WDLPS (*Adv Anat Pathol*, PMID: 32960834), we compared the ratio
 of KI67-positive tumor cells in WDLPS and DDLPS. DDLPS showed an obviously
 higher ratio of KI67-positive tumor cells than WDLPS (**Figure RL2F**). Consistently,
 HC3 also presented the higher ratio of KI67-positive tumor cells than HC2, implying
 that HC3 featured fast cell proliferation characteristics (**Figure RL2F**).

**Figure RL2.** (F) Boxplots illustrating the ratio of KI67-positive tumor cells in HC2 and HC3
 (left) and histological subtypes belonging to HC2/HC3 (right).

 GSVA analysis revealed that DDLPS (HC3) could be distinguished from WDLPS and
 MLPS (HC2) by elevated enrichments of Rab pathway (**Figure RL2G-H**). The
 elevated protein expression of Rab GTPases including RAB14, RAB5A, RAB2A, etc.
 in HC3 confirmed the increased Rab pathway in HC3 (**Figure RL2I**).

**Figure RL2.** (G) The heatmap of specifically enriched pathways in hierarchical clusters; (H)
 Boxplots showing GSVA scores of Rab regulation of trafficking and Rab pathway in
 histological subtypes belonging to HC2/HC3.

Moreover, among the Rab GTPases that showed elevated expression in HC3, we
observed that the protein abundance of RAB2A and RAB14 were significantly
correlated with patients' prognosis (**Figure RLJ**).

**Figure RL2.** (I) The heatmap presenting Rab GTPases enriched in HC3; (J) The forest plot
showing the hazard ratios of Rab GTPases enriched in HC3.

Previous researches have reported that Rab GTPases participated in cell autophagy
(*Cell Death Differ*, PMID: 24440914; *Cell Biosci*, PMID: 33557950). RAB2A has
been proved to regulate the formation of autophagosome and autolysosome (*Autophagy*,
PMID: 30957628). Researches have indicated that the elevated autophagy might be
associated with tumor proliferation (*Clin Cancer Res*, PMID: 26567363), we then
hypothetically assumed that the elevated autophagy might lead to significantly fast
tumor cell proliferation and cell proliferation index in HC3.

Aim to confirm this assumption, we compared the autophagy pathway between HC2
and HC3, and found that both the autophagy pathway enrichment scores as well as
autophagy markers (ATG5, ATG7, MTOR, WIPI1) showed elevation in HC3 than HC2
(**Figure RL2K-M**). Moreover, proliferation index of sarcoma is both correlated with
protein expression of RAB2A and autophagy pathway GSEA scores (**Figure RL2N**).
These findings illustrated that comparing to WDLPS and MLPS which belong to HC3,
DDLPS, which belongs to HC2, showed fast tumor cell proliferation features, which
might be caused by the RAB2A-associated autophagy process.

**Figure RL2.** (K) The scatter plot presenting the positive correlation between RAB2A and
 autophagy pathway; (L) The boxplot presenting the enrichment scores of autophagy in different
 clusters; (M) Boxplots presenting the abundances of autophagy markers in different clusters;
 (N) The scatter plot presenting the positive correlation between proliferation index and
 autophagy pathway (left) or abundance of RAB2A (right)

In sum, our hierarchical clustering showed clinical relevance and could help to illustrate
 the common features among different histological sarcomas and could further decipher
 the distinctive biological features of lipid sarcomas varies with degrees of
 differentiation. In the revised manuscript, we have updated the methods for hierarchical
 clustering in the “**Methods**” section and updated our analysis on the HC2 and HC3 in
 the “**Result**” section (line297-330). Also, we updated **Figure RL2** in the revised
 **Figure2, Supplementary Figure 8&9.**

**3. Line 239: “Meanwhile, *HC1-enriched proteins (student’s t test, fold-change >**
 **1.5, adjusted P value <=0.05) participating in pathways correlated with**
 **metabolism (as shown in figure 2C) were filtered out.**

**Is it supposed to be *HC2-enriched proteins? Or did the authors filter out some of**
 **the H1-enriched proteins? This is unclear and confusing.**

**Response:**

We thank the reviewer for the comment and apologize for the incorrect phrasing of line
 239. We have revised the manuscript as follows: “Meanwhile, HC2-enriched proteins

(student's t-test, fold-change > 1.5, adjusted p-value <= 0.05) participating in pathways
correlated with metabolism (as shown in Figure 2C) were filtered out.”

**4. The work involves many different analyses touching on some relationships only**
**superficially. I think to further test these models/hypotheses is out of the scope of**
**the paper, however, the authors should take care in their concluding remarks in**
**the results section. The findings support or are suggestive of a particular**
**mechanism but were not functionally tested. The conclusive remarks in the result**
**section should reflect that.**

**For example:**

**Line 283: “In sum, the upregulation of SHC1 drives poorer diagnoses of patients**
**diagnosed with AS or ES through promoting actin cytoskeleton reorganization and**
**epithelial cell migration by phosphorylation of ADD2 Ser2”.**

**Although the integration between proteomics and phosphoproteomics is very**
**interesting, and suggestive of a role for SHC1, the data shown does not directly**
**implicate SHC1 in poor prognosis by promoting cytoskeleton reorganization or**
**cell migration. In fact, only cell viability was measured upon SHC1 inhibition at**
**this point in the paper. Also, the authors did to confirm that indeed SHC1 is able**
**to phosphorylate ADD2 and later in the paper it is suggested that SHC1 regulates**
**metastasis by phosphorylating CTNNB1 (instead or in addition?) (Fig.3K, L). This**
**conclusion should be rephrased.**

**Response:**

We thank the reviewer for this critical comment. We agree with the reviewer that more
evidence could help to elucidate the mechanism of how SHC1 regulates cell migration
through phosphorylation and leads to metastasis and poor prognosis in sarcoma. To
systematically respond to the comments, we divided the comments into 2 parts:

**Part 1. The association between SHC1 and cell migration in the HC1 cluster.**

In our previous version, we grouped the 12 histological types of sarcomas into 6

hierarchical clusters (HC), among which HC1 containing both AS and ES showed the
worst prognosis. Differential expression analysis combined with GO pathway analysis
revealed HC1 featured with enrichment of the TGF β signaling pathway. To further
elucidate the mechanism underlying the poor prognosis of HC1 patients, we focused on
the HC-elevated proteins that enriched in the TGF β signaling pathway and identified
SHC1 as the top-ranked elevated protein in HC1 associated with patients' poor
prognosis. We then hypothetically assumed that SHC1 might play an important role in
leading the poor prognosis of HC1 sarcoma, through enhancing TGF β mediated tumor
cell migration.

To confirm this assumption, in the revision, we conducted the following experiments:

**1.1. SHC1 could impact the HC1 tumor cell migration.**

To investigate the role of SHC1 in impacting the tumor cell migration in HC1 cluster,
we utilized the ASM cell line (the cell line originating from AS), since AS is the main
component of HC1. We constructed the stable SHC1-overexpressed ASM cell line
(SHC1-OE-ASM) utilizing the pCDH-SHC1-copGFP vector and also knocked down
*SHC1* (SHC1-KD-ASM) utilizing pLKO.1-CMV-shSHC1-copGFP. RT-PCR analysis
was utilized to verify the expression of SHC1 in SHC1-OE-ASM and SHC1-KD-ASM.
The results confirmed the significantly elevated expression of SHC1 in SHC1-OE-
ASM and the significantly decreased expression of SHC1 in SHC1-KD-ASM (**Figure**
**RL3A**). We then evaluated the cell migration rates using transwell assay. As a result,
the SHC1-OE-ASM cell line showed increased cell migration ability, whereas the
SHC1-KD-ASM cell line exhibited decreased cell migration ability compared with
control cells (**Figure RL3B**).

**Figure RL3. Functional experiments to validate the impacts of the SHC1-PTK2 and**
 **SHC1-CSNK1G1 axis in sarcoma**

(A) The expression of SHC1 in SHC1-OE-ASM, SHC1-KD-ASM, and the control group by
 RT-PCR. (B) The effects of SHC1 on the migration of ASM cells were confirmed by the
 transwell assay. The bar plots (right panel) indicate counts of migrated ASM cells under
 different treatments.

 **1.2. Comparative analysis revealed PTK2 as the core kinase that linked SHC1 and**
 **the phosphorylation of ADD2.**

Published researches have indicated that SHC1 participated in various biological
 process, and might regulate downstream pathways through phosphorylation (*Nature*,
 PMID: 23846654; *Nat Commun*, PMID: 28276425; *Front Cell Dev Biol*, PMID:
 33693003). Therefore, in our previous version, to further illustrate how SHC1 led to
 cell migration, we performed correlation analysis and observed that the
 phosphorylation of ADD2 (functions in cytoskeleton reorganization and epithelial
 migration) at Ser2 showed the most significant correlation with SHC1. Combined with
 clinical information, we found that the phosphorylation of ADD2 at Ser2 was
 significantly associated with patients' poor prognosis.

 Functionally, SHC1 is an adapter protein that could interact with different kinases and
 participate in signal transduction pathways (*Nature*, PMID: 23846654). In the revision,
 to elucidate the kinase that related to SHC1 and might regulate the phosphorylation of
 ADD2 at Ser2 in HC1 cluster, we referred to the public database (PhosphoSite [<https://>

www.phosphosite.org/homeAction.action], Phos-pho.ELM [[http://](http://phospho.elm.eu.org/dataset.html)
 phospho.elm.eu.org/dataset.html], and PhosphoPOINT [[http://](http://kinase.bioinformatics.tw/)
 kinase.bioinformatics.tw/] and conducted correlation analysis. As a result, among the
 kinases reported to regulate phosphorylation of ADD2, PTK2, was identified as the
 kinase that showed the most significant correlation with SHC1 and comparatively
 higher expression in the HC1 cluster (**Figure RLC-D**).

**Figure RL3.** (C) The scatter plot presenting the significantly positive correlation between the
 protein expression of PTK2 and SHC1 (Spearman's rank correlation). (D) The violin plot
 indicated the PTK2 protein expression among HC clusters.

1.3. Inhibiting PTK2 could impact the increased cell migration leading by SHC1.

To further investigate the role of PTK2 in impacting cell migration, we utilized SHC1-
 OE-ASM and OE-Ctrl-ASM cells and treated them with PTK2 inhibitors. We then
 evaluated the cell migration by transwell assay. As a result, inhibiting PTK2 could
 significantly decrease the cell migration rates increased by SHC1 (**Figure RL3E**).
 Moreover, overexpression of PTK2 in SHC1-KD-ASM significantly increased cell
 migration which was inhibited by knocking down SHC1 (**Figure RL3F**). These results
 implied that the kinase, PTK2 participated in cell migration driven by SHC1.

**Figure RL3.** (E-F) The transwell assay confirms effects of the SHC1-PTK2 axis on the

migration of ASM cells. The bar plots indicated the migrated cell counts of ASM cells under
different treatments.

We further performed phosphoproteomic analysis between SHC1-OE-ASM treated
with or without PTK2 inhibitor. As a result, the phosphorylation of some proteins
participating in actin cytoskeleton reorganization and epithelial cell migration showed
a significant elevation in SHC1-OE-ASM and a significant decrease in SHC1-OE-ASM
treated with the PTK2 inhibitor, such as ADD2 at S2, FGD4 at S702, and EPB41 at
S542 (**Figure RL3G**). These observations confirmed the role of PTK2 in
phosphorylating ADD2 at S2 and elevation actin cytoskeleton reorganization pathways.

**Figure RL3.** (G) The boxplots indicate the phosphorylation intensity of ADD2 S2 and other
phosphosites participating in actin cytoskeleton reorganization under different treatments.

In sum, the above experiments confirmed our assumption that SHC1 could impact the
cell migration through phosphorylating ADD2 at Ser2, mediated by PTK2.

**Part2 The association between SHC1 expression and high metastatic rates of PC-** 371 **Ra-HC1.**

Additionally, in our previous manuscript, to investigate the heterogeneity within and
across the histological subtypes of sarcomas, we performed proteomic-based subtyping
and divided the sarcoma into three proteomic subtypes (PC-Ra, PC-Cc, and PC-Sm)
with patients belonging to PC-Ra had the highest metastatic rates. Importantly, the
integrative analysis of hierarchical and proteomic clusters revealed that PC-Ra
contained samples from both HC1 (PC-Ra-HC1) and other HCs (PC-Ra-oHCs). We

then conducted further analysis and found that the elevated metastatic rates of PC-Ra-
HC1 might be caused by SHC1-mediated angiogenesis (student's t-test: p-value =
0.042), while PC-Ra-oHCs might be caused by MAPK10-mediated MAPK signaling
pathway (student's t-test: p-value = 2.1e-4).

In previous version, to verify the impact of SHC1 for metastasis in PC-Ra-HC1, we
have constructed SHC1-overexpressed vector and transfected it into the ISOHAS cell
line (SHC1-OE-ISOHAS) which showed similar expression patterns with PC-Ra-HC1,
and conducted the transwell migration assay. As a result, the increased migration ability
of SHC1-OE-ISOHAS was observed, confirming the role of SHC1 in enhancing
metastasis of tumors belonging to PC-Ra-HC1 (shown in the original **Figure 3I**).

In the revision, we then tried to further illustrate the mechanism underlining this
phenomenon. Since we have confirmed that as an adaptor protein, SHC1 could interact
with PTK2 and phosphorylated ADD2 to elevate the actin cytoskeleton reorganization
pathway in HC1, we then evaluated the expression of PTK2 and phosphorylation of
ADD2 in HC1-PC-Ra. As a result, comparing to HC1-oPCs (HC1 samples which were
grouped into other proteomic clusters), PTK2 and phosphorylation of ADD2 at S2
showed no significantly elevation in HC1-PC-Ra (**Figure RL3H**), implying that PTK2
phosphorylated ADD at S2 might be the common features shared by both HC1-PC-Ra
and HC1-oPCs, and SHC1 might cooperate with other kinases to promote metastasis of
HC1-PC-Ra.

**Figure RL3.** (H) Boxplots illustrate the abundances of PTK2 and ADD2 S2 in HC1-PC-Ra
and HC1-oPCs

We then explored the potential phosphosites that might be regulated by SHC1 and lead
 to metastasis in PC-Ra-HC1. As a result, the phosphosite CTNNB1 at Ser552 was
 identified to be the potential regulatory phosphosites of SHC1. Further validation
 experiments were conducted and verified that elevated expression of SHC1 could
 elevate the phosphorylation of CTNNB1 at Ser552 (**Figure RL3I**).

**Figure RL3. (I)** Heatmap illustrating elevated phosphosites in high-SHC1 patient derived
 cells.

In the revision, to further explore the kinase that associated with SHC1, the
 phosphorylation of CTNNB1 at Ser552 and the tumor metastasis. The following
 experiments were performed:

**2.1 Comparative and correlation analysis revealed CSNK1G1 as the core kinase**
 **that linked SHC1 and the phosphorylation of CTNNB1.**

We referred to the public database and performed further correlation analysis. As a
 result, among the public reported kinases of CTNNB1, CSNK1G1 showed the
 significantly positive correlation with both SHC1 and the phosphorylation of CTNNB1
 at Ser552 (**Figure RL3J-K**). Consistently, the phosphorylation of CSNK1G1 also
 showed elevated expression level in PC-Ra (**Figure RL3L**).

**Figure RL3.** (J) The correlation of the expression of CSNK1G1 with SHC1 expression
 (Spearman' s rank correlation). (K) The correlation of the expression of CSNK1G1 with the
 phosphorylation of CTNNB1 at Ser552 (Spearman' s rank correlation). (L) The boxplot
 indicates the expression of CSNK1G1 in different proteomic clusters.

**2.2. Inhibiting CSNK1G1 could impact the increased cell migration leading by**
 **SHC1.**

To further investigate the role of CSNK1G1 in impacting tumor metastasis, we utilized
 the constructed SHC1-OE-ISOHAS and Ctrl-OE-ISOHAS cells and treated them with
 the CSNK1G1 inhibitor. We then evaluated the cell migration by transwell assay. As a
 result, inhibiting CSNK1G1 could significantly decrease the cell migration rates
 increased by SHC1 (**Figure RL3M**). These results implied that CSNK1G1 as a kinase
 participated in tumor metastasis in PC-Ra-HC1 driven by SHC1.

**Figure RL3.** (M) The effects of the SHC1-CSNK1G1 axis on the migration of ISOHAS cells
 were confirmed by transwell assay. The bar plots indicated the migrated cell counts of ISOHAS
 cells under different treatments.

We further performed phosphoproteomic analysis between SHC1-OE-ISOHAS treated
 with or without the CSNK1G1 inhibitor. As a result, the phosphosites of proteins
 participating in angiogenesis, especially CTNNB1 Ser552, significantly decreased in
 SHC1-OE-ISOHAS treated with the CSNK1G1 inhibitor (**Figure RL3N**). These
 observations confirmed the role of CSNK1G1 in phosphorylating CTNNB1 at Ser552.

**Figure RL3.** (N) The boxplots indicate the phosphorylation levels of CTNNB1 Ser552 and
 other phosphosites participating in angiogenesis under different treatments.

The above results confirmed our assumption that SHC1 could lead to PC-Ra-HC1
 tumor migration through phosphorylating CTNNB1 mediated by CSNK1G1.

In sum, our data illustrated the two potential functions of SHC1, by interacting with
 PTK2 and phosphorylating ADD2 at Ser2, SHC1 will enhance cell migration, and lead
 to poor prognosis of HC1 patients. Meanwhile, for a group of HC1 patients that showed
 proteomic features of Pc-Ra, SHC1 will further interact with CSNK1G1 and
 phosphorylating CTNNB1 at Ser552, and lead to tumor metastasis (**Figure RL3O**).

According to the reviewer’s comments, we also toned down our statements as follows:

“In sum, the upregulation of SHC1 might interact with kinase PTK2, phosphorylating
 ADD2 at Ser2, enhanced cell migration. This phosphorylation cascade might be
 associated with the poor prognosis with HC1 patients (AS or ES).”

**Figure RL3.** (O) Sankey plot illustrates the distribution of HC1 in proteomic clusters and
 related phosphorylation process.

In the revision, we have updated **Figure RL3** in the revised **Supplementary Figure**
 **10&11** and the “**Result**” section on lines297-334, line355-384, line 397-417, and line
 520-551. in the revised manuscript.

**Line 458: “Taken together, our data illustrated that RIOK1 phosphorylates NPM1**
 **on Ser125 to assist the interaction of NPM1 and APEX1 resulting in cell**
 **proliferation in PC-Cc”. Again, the authors show increase proliferation by NPM1**
 **overexpression, and that NPM1 interacts with APEX1 but the rest of the data is**
 **correlative.**

**Response:**

We appreciate the reviewer for this critical suggestion, and we agree with the reviewer
 that more evidence should be provided before making conclusion. According to
 reviewer’s suggestion, in the revision, we performed further analysis and functional
 experiments to confirm our findings.

Specifically, we utilized the sarcoma cell line, RKN, for further functional experiments,
 as it originates from LMS and represents the proteomic features of PC-Cc. We
 constructed the RIOK1-overexpressed RKN cell line (RIOK1-OE-RKN) through the
 RIOK1 overexpression plasmid, pCDH-RIOK1-copGFP. Moreover, shRNA of RIOK1
 were designed and transfected into RKN cell line to knock down the expression of
 RIOK1 (RIOK1-KD-RKN). We then performed CCK8 cell proliferation assay and

evaluated the cell proliferation rates. As a result, RIOK1-OE-RKN showed most
 significantly elevated cell proliferation rates and RIOK1-KD-RKN had significantly
 decreased cell proliferation rates (**Figure RL4A**). We also treated RIOK1-OE-RKN
 cell line with RIOK1 inhibitor, and the inhibitor significantly decreased the
 proliferation of RIOK1-OE-RKN (**Figure RL4A**). These observations confirmed the
 impact of RIOK1 on promoting sarcoma tumor cell proliferation. We then performed
 comparative proteomic and phosphoproteomic analysis among RKN sarcoma cell lines
 with different treatments (RKN transfected with empty vector, RIOK1-OE-RKN,
 RIOK1-OE-RKN treated with RIOK1 inhibitor, RKN transfected with scrambled
 shRNA, RIOK1-KD-RKN). As a result, besides APEX1, the proteins participating in
 DNA base excision repair including XRCC1, XRCC4, POLB, as well as cell
 proliferation index KI67 showed elevated expression in RIOK1-OE-RKN (**Figure**
 **RL4B-C**). Intriguingly, the phosphorylation of NPM1 at Ser 125 was significantly
 increased in RIOK1-OE-RKN, implying that RIOK1 regulated the phosphorylation of
 NPM1 (**Figure RL4C**).

**Figure RL4. Functional experiments to validate the impact of RIOK1 and the interaction**
 **between NPM1 and APEX1.**

(A) Proliferation of the RKN cell line associated with different treatments (n = 4 repeats per
 group). (B) The heatmap reveals the expression patterns of DNA base excision proteins across
 the cells associated with various treatment (n = 3 repeats per group). (C) The boxplots reveal
 the abundance of APEX1, KI67 and phosphorylation of NPM1 at Ser125 in RKN cell line with

different treatments.

To further investigate the impact of NPM1 phosphorylation on cell proliferation as well
as on its interaction with APEX1, we then constructed NPM1 phosphorylation site
mutant plasmid, NPM1^{S125A}, and transfected it into RIOK1-KD-RKN cells (NPM1-
mut-OE-RIOK1-KD-RKN). The non-mutant NPM1 was also transfected into RIOK1-
KD-RKN cells (NPM1-OE-RIOK1-KD-RKN) which were utilized as controls. By
evaluating the cell proliferation rates, we observed that comparing to RIOK1-KD-RKN
cells, NPM1-OE-RIOK1-KD-RKN cells showed elevated cell proliferation rates,
whereas the cell proliferation rates of NPM1-mut-OE-RIOK1-KD-RKN showed no
significant elevation (**Figure RL4D**). Consistently, the cell proliferation index, KI67
was also observed to be elevated only in NPM1-OE-RIOK1-KD-RKN cells (**Figure**
**RL4E**). Meanwhile, comparative proteomics and phosphoproteomic data confirmed
the increased expression of APEX1 as well as the increased phosphorylation of NPM1
at Ser125 in NPM1-OE-RIOK1-KD-RKN cells (**Figure RL4E**).

These results indicated the decreased cell proliferation rates led by knocking down
RIOK1 could only be rescued by the wild type NPM1 overexpression, which further
emphasized the role of phosphorylation of NPM1 in mediating RIOK-dependent
regulation of the tumor cell proliferation.

To further illustrate whether the phosphorylation of NPM1 affected its interaction with
 APEX1, we performed IP-MS using both NPM1-mut-OE-RIOK1-KD-RKN and
 NPM1-OE-RIOK1-KD-RKN (**Figure RL4F**). As a result, 17 proteins were identified
 to interact with the wild type NPM1, but not NPM1^{S125A}. Among them, APEX1
 presented the highest abundance, proving that NPM1 Ser125 is the pivotal site for the
 interaction between NPM1 and APEX1 (**Figure RL4G-H**). The above results
 illustrated the potential mechanism that RIOK1 could impact sarcoma tumor cell
 proliferation through phosphorylating NPM1 which then interacted with APEX1 and
 promoted tumor cell proliferation accordingly.

**Figure RL4.** (D) Proliferation of the RNK cell line associated with various treatments (n = 4).
 (E) The boxplots presenting the expression of KI67, APEX1 and phosphorylation of NPM1
 among NPM1-OE-RIOK1-KD-RKN, NPM1-mut-OE-RIOK1-KD-RKN and EV-RIOK1-KD-
 RKN. (F) The schematic work flow of the IP-MS experiment for the NPM1. (G) The heatmap
 reveals the expression patterns of DNA base excision proteins across the NPM1-OE-RIOK1-
 KD-RKN, NPM1-mut-OE-RIOK1-KD-RKN (n = 3 repeats per group). (H) Diagram illustrated
 RIOK1 activates NPM1 through Ser125 and then NPM1 interacts with APEX1 to promote cell
 proliferation.

We updated the **Figure RL4** in the revised **Figure 4** and **Supplementary Figure 12**
 and added the words in the revised manuscript on lines 631-669. Meanwhile, following

reviewer's suggestion, we also toned down our statements as following "Taken together,
our data illustrated the potential mechanism underline how the axis of RIOK1-phos-
NPM1-APEX1 might promote tumor cell proliferations."

**5. The TME deconvolution analysis suggests some interesting relationships. Since**
**this type of analysis is usually done from transcriptomic data, I wonder to what**
**extent is well established for proteome analysis. Was this validated with an**
**alternative deconvolution method or by a couple of IHC markers to validate**
**enrichment of immune cell populations in some of the samples analyzed?**

**Response:**

Thanks for the reviser's suggestion. In the previous version, to evaluate the tumor
microenvironment of sarcoma, we inferred cellular compositions in the
microenvironments of sarcomas utilizing xCell deconvolution algorithm (*Genome Biol*,
PMID: 29141660) based on proteomic data. We utilized inferred cell deconvolution
data to classify the sarcomas into 3 immune subgroups with distinctive immune features.

In agreement with the reviewer, we acknowledged that the cell deconvolution analyses
were usually based on transcriptomic data. Meanwhile, previous researches have
revealed tumor microenvironment infiltration estimated by proteomic data had a high
Pearson's correlation with ones estimated by transcriptomic data (*Cell*, PMID:
31675502; *Cell*, PMID: 32649874; *J Hematol Oncol*, PMID: 35659036; *Nat. Commun*,
PMID: 36720864). Moreover, published research has indicated the potential of using
proteomic data for xCell analysis could illustrate the tumor microenvironment
infiltration. For example, in the multilevel proteomic research of diffuse-type and
intestinal-type gastric cancer (*Nat Commun*, PMID: 36788224), the immune clustering
of xCell-deconvoluted tumor microenvironment components based on proteomic data
revealed that Th1/Th2 ratio could serve as an indicator for immunotherapeutic
effectiveness, which was validated in an independent GC anti-PD1 therapeutic patient
group. In addition, a proteogenomic search of cholangiocarcinoma (*Hepatology*, PMID:
35716043) revealed that a higher level of xCell-derived CD4+ T cells based on
proteomic data was associated with the favorable prognosis, which was further

confirmed in a combined cohort. These researches showed the findings uncovered by
xCell based on proteomic data could be further validated by other independent
experiments, indicating the importance of proteomic data in the tumor
microenvironment.

In the revision, to further confirm the immune features inferred by proteomic data based
xCell analysis, we conducted the following analysis and experiments: (1) we utilized
the other two cell deconvolution analysis tools (CIBERSORT and ESTIMATE
algorithms) to infer the immune features of the sarcoma tumor microenvironments; (2)
we investigated the expression patterns of cell-type specific proteins to confirm the
distinctive cell type distribution among different immune subtypes of sarcomas; (3) We
also utilized IHC staining to verify the distinctive cell type distribution among different
immune subtypes of sarcomas.

**1. Tumor microenvironment deconvolution analysis using CIBERSORT and**
**ESTIMATE algorithms, confirmed distinctive immune features inferred by the**
**xCell algorithm.**

To confirm our cellular composition analysis by xCell algorithms, we utilized
ESTIMATE and CIBERSORT methods to infer each patient's total immune cell
infiltration scores and distinctive cell type enrichment scores. We then compared both
total immune scores and cell-type specific enrichment scores among the three immune
subtypes (clustering based on cellular deconvolution scores of the xCell algorithm).
The results confirmed the consistent conclusion inferred by the three deconvolution
methods. As for the total immune and stroma scores, both CIBERSORT and
ESTIMATE confirmed the conclusion inferred by xCell algorithms. Particularly, the
immune subtype that harbored the highest immune infiltration score was IM-S-3, and
the immune subtype that held the highest stromal scores was IM-S-1. Meanwhile, as
for cell-type specific enrichment scores among the three immune subtypes, in
concordant with the distinctive cell-type enrichments revealed by xCell analysis,
CIBERSORT also indicated that the IM-S-3 showed the highest enrichment scores of
CD8⁺ T cell, M1 macrophage and M2 macrophage, and IM-S-2 showed the highest

memory B cell enrichment scores (Figure RL5A-B). These results confirmed the
 feasibility of our proteomic-based xCell deconvolution analysis in predicting the
 distinctive cell type enrichment in sarcoma tumor microenvironments.

 **Figure RL5. The immune cell signatures and cell markers in different immune clusters**
 (A) The heatmap illustrates the immune and stromal cell types enriched in different immune
 clusters; (B) Boxplots illustrate cell signature scores inferred by xCell and CIBERSORT
 algorithm among three immune clusters.

**Figure RL5 (C)** Boxplots illustrate the immune scores and stromal scores calculated by xCell
 and ESTIMATE algorithm among the three immune clusters.

**2. The expression patterns of cell types specific markers confirmed the distinctive**
**cell type distribution among immune subtypes of sarcomas.**

To confirm the distinctive tumor microenvironment inferred by cell deconvolution
analysis, we focused on the significantly enriched cell types of each immune subtype
and evaluated the mass-spectrum-based abundance of their distinctive markers among
three immune clusters. As a result, for IM-S-1 that enriched with Keratinocyte, we
evaluated the protein expression of Keratinocyte markers, and observed dominant
expression of CD34, KRT14, KRT9 and KRT5 in IM-S-1. Meanwhile, for IM-S-2 that
enriched with endothelial cells, we evaluated the expression of endothelial cell
markers and detected MCAM showed significantly elevated expression in IM-S-2.
Moreover, for IM-S-3 that enriched with CD4⁺ T cells and macrophages, we
investigated the expression of CD4⁺ T cell markers and macrophage markers, and found
the protein expression of CD4⁺ T cell markers (CD4, CD38, and ISG20) and
macrophage markers (CD14, CD163, CSF1R, and FCGR1A) presented significantly
higher levels in IM-S-3 (Figure RL5A, 5D). These results also verified our proteomic-
based xCell deconvolution analysis.

**Figure RL5 (D)** Boxplots illustrate proteomic abundance of immune cell markers in immune
clusters.

**3. IHC staining verified the distinctive cell type distribution among different**

**immune subtypes of sarcomas**

To further verify our TME deconvolution analysis, we randomly selected several
markers for distinctive cell types of each immune subtype (KRT5 & KRT9 for
Keratinocyte, CD4 & ISG20 for CD4⁺ T cells, CD19 & IgM for B cells) and obtained
their expressions through IHC staining (**Figure RL5E**). These markers showed
consistent enrichment in immune clusters with related xCell-enriched cell types
(**Figure RL5A, 5E**). For example, CD4⁺ T cells had the highest infiltrated scores in the
IM-S-3 group. Consistently, the IHC results also presented the highest CD4 and ISG20
expressions in the IM-S-3 group. Meanwhile, IHC staining using CD19 and IgM
confirmed the elevated expressions of these two B cell markers in the IM-S-2. Moreover,
IHC staining using KRT5 and KRT9 verified the dominant expression of these
keratinocytes in IM-S-1. In sum, these IHC staining provided a convincing proof for

our TME convolution result.

**Figure RL5 (E)** IHC staining presenting expressions of immune and stromal cell markers in
three immune clusters.

In sum, both ESTIMATE and CIBERSORT algorithm confirmed xCell inferred cell
types distributions among the three immune subtypes. Further cell marker expression
analysis and IHC staining also revealed the consistence of the tumor microenvironment
features of the three distinctive immune clusters. In the revision, we have updated
**Figure RL5** in **Supplementary Figure 14**, we updated these above analyses in lines
710–722 of the “**Results**” section of the revised manuscript.

**6. The discussion is too long and has some paragraphs that are not well written.**

**See for example the paragraph starting on line 647.**

**Response:**

We thank the reviewer for this constructive suggestion and sincerely apologize for the
unclear description of the discussion section. The main points that we want to present
in the discussion section are summarized as follows:

(1) Summarizing the hypothesis and purpose of the study.

(2) Comparing and contrasting to previous studies of our main findings such as the
common and specific features of different histological subtypes of sarcomas.

(3) Discussing the potential therapeutical options of the sarcomas.

According to the reviewer's suggestion, we thoroughly revised the discussion section,
and concisely wrote the essential interpretation and main pieces of supporting evidence,
which have been described in the result section. We further added limitations of the
study, as well as potential future research. The revised discussions were presented as
follows:

**Discussion**

In this study, we establish a Chinese pan-sarcoma cohort including 272 patients and 12
usual or unusual sarcoma histologic subtypes. We performed integrate proteomic and
phosphoproteomic data to reveal the differentially overrepresented signaling pathways
in STS histologic subtypes, metastasis-related proteins, and therapeutically relevant
subgroups. Our study with this cohort would serve as a complement to the previous
genome and transcriptome studies and exhibit a range of clinic-histologic spectrums of
pan-sarcoma.

The heterogeneity and variability of sarcoma histological subtypes make it difficult to
understand the features of histological subtypes and guide clinical management.
Employing the hierarchical clustering, we could reveal the intrinsic common features
of different histological subtypes of sarcoma and define subgroups across histological

subtypes from the proteomic viewpoint. Although, at histological level, WDLPS,
MLPS, and DDLPS all belong to the category of liposarcoma, our proteomic-based
hierarchical clustering revealed the DDPLS showed the similar proteomic features with
MFS than with MLPS and WDLPS. Specifically, the cell proliferation scores were
significantly elevated in both MFS and DDLPS. These findings confirmed the previous
transcriptomic research that indicated the DDLPS showed comparatively elevated cell
proliferation features at mRNA level¹. Importantly, by performing comparative analysis,
we found the RAB signaling pathway was dominantly enriched in DDLPS, and further
illustrated that RAB2A might led to tumor cell proliferation of DDLPS by increasing
autophagy process. These results implicated that inhibiting autophagy might be a
promising therapeutical option for patients with DDLPS.

MFS was once considered a subset of UPS (“myxoid malignant fibrous histiocytoma”),
but they have been classified as distinct clinical entities based on their different
clinicopathologic features². Despite the clinical classification, the molecular diversity
of these two subtypes have not been uncovered, thus for now, the treating strategies for
UPS and MFS remain the same. Our research revealed that MFS showed enriched
transport-related pathways, whereas UPS showed enriched RNA process and
metabolism pathways. The diverse proteomic features of UPS and MFS implied the two
different histological sarcoma subtypes could be benefited from distinctive
therapeutical approaches in the feature.

AS represents a rare group of soft-tissue sarcomas and are aggressive endothelial cell
tumors of vascular or lymphatic origin^{3,4}. Angiogenesis is thought to be associated with
the pathogenesis of AS and is regarded as a potential target for treatment. However,
some clinical trials of anti-angiogenesis drugs in AS don’t have positive results or only
showed limited improved DFS, including bevacizumab (VEGF-A antibody), trebananib
(an angiopoietin-1 and -2 peptibody), and sorafenib (VEGFR and B-Raf inhibitor)^{5,6}.
By performing integrative analysis and functional experiments, our study identified
SHC1 as the key regulator, which could elevate actin cytoskeleton reorganization and

led to unfavorable outcomes of AS patients. These results implied that SHC1 might
serve as a promising therapy target for AS patients.

The diverse immune features have been reported to be associated with the prognosis of
sarcoma patients, but the majority of these researches were either down in animal
models or have one layer of omics data. For instance, Magrini and colleagues have
utilized transcriptomic data from sarcoma mice model to illustrate that the sarcoma
tumor cells could express C3 which could then recruit macrophages through C3-C3aR
axis, thus C3 deficiency-associated signatures of macrophages could lead to favorable
prognosis in sarcoma⁷. Since we have also observed elevated C3 protein expression in
tumor tissues, we then investigated the potential association among C3 protein
expression, the recruitment of macrophages and patients' prognosis. As a result, the
significant positive correlation between C3 and macrophage enrichment was observed
in our pan-sarcoma dataset. Further integrative analysis with patients' prognosis revealed
that the C3-deficiency macrophage signature based on proteomic was associated with patients'
prognosis, consistent with the result gotten from transcriptome previously. Meanwhile,
previous research conducted by Petitprez et al. have utilized transcriptomic data based
immune analysis to decipher the immune diversity in pan-sarcomas⁸. They have proved
the enrichment of B cells led to favorable out comes in several sarcoma histological
subtypes (LMS, AS, UPS and MFS). We then evaluated the prognostic relevance of B
cell enrichment, as a result, the similar clinical relevance of B cells was also observed
in the four histological subtypes in our cohort, implied the concordance in evaluating
immune features either by transcriptomic or by proteomic data. Moreover, to further
elevate the clinical applicable of utilizing B cells to prognostic index, we further
evaluated the prognostic relevance of the B cell markers' protein expression in our
sarcoma cohort and published TCGA cohort. As a result, among the 12 B cell markers
that have been detected in our dataset, 7 B cell markers showed significant association
with patients' prognosis in our pan-sarcoma cohort. 3 of these B cell markers (PTPRC,
CD9, IGLL5) showed consistent prognostic relevance at transcriptomic level in TCGA
cohort (*Cell*, PMID: 29100075). These results implying the potential clinical utilization
of these 3 B cell markers for prognostic prediction in feature.

Immune therapy has been applied to many malignancies and presents improved
clinical outcomes, such as melanoma. Some clinical studies for immune therapy in STS
have been completed and obtained positive results for advanced, metastatic, or
unresectable STS^{9, 10}. Despite the progression of immune therapy in STS, the
heterogeneity of TME components within STS histologic subtypes makes it a challenge
to distinguish patients responding to immune therapy. Intriguingly, based on TME
components, we defined a subtype of STS (IM-S-3) with enriched immune infiltration
and immune evasion markers (CD274 and CD80) which might respond to immune
therapy, especially PD-L1 inhibitors. Besides the heterogeneity in STS histologic
subtypes, the interaction between tumor biologic process and TME in STS is quite
important for the potential combination therapies for sarcoma¹¹. Our results implied
that the CTNNB1 may contribute to the transcription of CD274 in the immune-enriched
group of STS. Meanwhile, MAPK10 participates in this process by phosphorylation of
CTNNB1 Ser675. Based on our research, we provide a viewpoint that combined
blockade of MAPK10 and CD274 might be an effective strategy for STS. Meanwhile,
combined blockade of CTNNB1 and CD274 could possibly achieve the same effect.
These conclusions still require further research.

The aims of this study were to provide a proteomic and phosphoproteomic landscape
to decipher the sarcomas' heterogeneity, the prognosis-related markers, and abnormally
changed biology pathways. There are some limitations due to the sample collection and
technology as follows:

1. The sarcoma cohort in this study is single-centered from Fudan University,
Zhongshan Hospital and included only Chinese patients, so the conclusions may lead
to potential selection bias. Additional prospective studies are needed to validate our
findings in multi-center and cohort of other ethnicities.

2. We found specific subtype-enriched proteins which might be serviceable in early
diagnosis and histological subtype detection, but we couldn't exclude the possibility
that this protein could have stemmed from other affected organs or may be indirectly
induced by the effects of the tumors on their microenvironment or even systemically.

Further experiments or clinical data are necessary complement to validate the roles of
this proteins in sarcoma.

3. The proteomic data in this study was generated through bulk proteomic approach
from tumor and NAT tissues and couldn't fully reflect the heterogenous tumor regions
and the tumor-NAT boundary regions. Integrating single cell and spatial omics would
be useful to further explore the intra-tumoral heterogeneity in the future research.

4. The samples in this study were all collected from treat-naïve patients and were all
primary tumors without remote metastasis or local relapse. The information about
metastasis and local relapse come from 60-month follow up. The conclusion in this
study that SHC1 and MAPK10 promotes metastasis required further confirmatory
studies on metastatic samples. Other conclusions were also just based on localized
diseases, it will have to be determined if these conclusions are also tenable in locally
relapsed and metastatic tumors.

Please see the details in the '**Discussion**' section of revised manuscript labeled in red
text.

**Reference**

1. Hirata, M. *et al.* Integrated exome and RNA sequencing of dedifferentiated
liposarcoma. *Nat Commun* **10**, 5683 (2019).

2. Doyle, L. A. Sarcoma classification: an update based on the 2013 World Health
Organization Classification of Tumors of Soft Tissue and Bone. *Cancer* **120**, 1763–
1774 (2014).

3. Fayette, J. *et al.* Angiosarcomas, a heterogeneous group of sarcomas with specific
behavior depending on primary site: a retrospective study of 161 cases. *Annals of*
*Oncology* **18**, 2030–2036 (2007).

4. Young, R. J., Brown, N. J., Reed, M. W., Hughes, D. & Woll, P. J. Angiosarcoma.
*The Lancet Oncology* **11**, 983–991 (2010).

5. Agulnik, M. *et al.* An open-label, multicenter, phase II study of bevacizumab for
the treatment of angiosarcoma and epithelioid hemangioendotheliomas. *Ann Oncol* **24**,
257–263 (2013).

- 6. Maki, R. G. *et al.* Phase II study of sorafenib in patients with metastatic or recurrent
sarcomas. *J Clin Oncol* **27**, 3133–3140 (2009).
- 7. Magrini, E. *et al.* Complement activation promoted by the lectin pathway mediates
C3aR-dependent sarcoma progression and immunosuppression. *Nat Cancer* **2**, 218–232
(2021).
- 8. Petitprez, F. *et al.* B cells are associated with survival and immunotherapy response
in sarcoma. *Nature* **577**, 556–560 (2020).
- 9. Somaiah, N. *et al.* Durvalumab plus tremelimumab in advanced or metastatic soft
tissue and bone sarcomas: a single-centre phase 2 trial. *The Lancet Oncology* **23**, 1156–
1166 (2022).
- 10. Tawbi, H. A. *et al.* Pembrolizumab in advanced soft-tissue sarcoma and bone
sarcoma (SARC028): a multicentre, two-cohort, single-arm, open-label, phase 2 trial.
*Lancet Oncol* **18**, 1493–1501 (2017).
- 11. D’Angelo, S. P. *et al.* Combined KIT and CTLA-4 Blockade in Patients with
Refractory GIST and Other Advanced Sarcomas: A Phase Ib Study of Dasatinib plus
Ipilimumab. *Clin Cancer Res* **23**, 2972–2980 (2017).

**7. Although the last integrative analysis presented in Figure 6 brings together the**
**different aspects analyzed in this study, it is unclear what it means in the**
**perspective of heterogeneity of STS subtypes. Some subtypes are enriched in**
**specific proteomic clusters which are then enriched in different immune signatures,**
**but still, there is a lot of variability on how HC are distributed.**

**Response:**

We apologize for the incomplete description and summary in Figure 6. In concordant
with the reviewer’s comment, the aim of Figure 6 is to present the result of integrative
analysis across the histological subtypes, hierarchical cluster, proteomic subtyping, and
immune subtyping. In our previous analysis, we focused on presenting how immune
subtyping could uncover the inner heterogeneity of TME in a distinctive proteomic
subtype. Particularly, the samples of PC-Ra could be further divided into IM-S-1 and
IM-S-3, where IM-S-3 showed higher immune cell infiltrations and immune checkpoint
inhibitors (**Figure 6A**). In the revised version, we added the interaction analysis

between hierarchical clusters and proteomic clusters or immune clusters to further
 illustrate the distribution of hierarchical clusters and correlated proteomic features.

**Figure RL6. Integration analysis of clustering result from different levels**
 (A) Graphical summary showing the characteristic pathways and major molecular findings of
 different level subtypes including histologic subtypes, hierarchical clusters, unbiased consensus
 proteomic clusters, and immune clusters. The relationships of these subtypes are also displayed.
 Firstly, we portrayed Sankey plots with hierarchical clusters as the center to present the
 concordance among histological sarcoma subtyping, hierarchical clustering, proteomic
 subtyping, and immune subtyping (**Figure RL6B-C**). As a result, for the relationship
 between hierarchical clusters and proteomic subtyping, we observed more than half of
 the HC2 patients (MLPS and WDLPS) were grouped into PC-Sm (32 of 47) and HC5
 (UPS) and HC6 (LMS) were both mainly clustered into RC-Cc (HC5: 33 of 43, HC6:
 36 of 52). Besides, HC3 (MFS, OtherFS, DDLPS, and MPNST) and HC4 (RMS and
 SS) were mainly distributed into two proteomic subtypes: PC-Ra (HC3: 35 of 75; HC4:
 12 of 33) and PC-Cc (HC3: 25 of 75; HC4:17 of 33). These results revealed our
 proteomic subtyping could uncover the heterogeneity within the two HCs and also
 common proteomic features that might be shared by samples from diverse HCs (**Figure**

**RL6B-C).**

Meanwhile, as for the relationships between hierarchical clusters and immune clusters,
the immune features of samples belonging to HC2, HC4, HC5, and HC6 showed
concordance within each HC. Specifically, 35 out of 47 samples of HC2 were grouped
into IM-S-1, 20 out of 27 samples of HC4 were clustered into IM-S-2, and 28 out of 43
samples of HC5 were grouped into IM-S-3 (**Figure RL6B-C**). On the other hand, the
immune heterogeneity within the distinguished HCs was observed in HC1 and HC3.
Samples belonging to these two HCs were mainly distributed into IM-S-1 and IM-S-3
equally (HC1: 12 in IM-S-1 and 9 in IM-S-3; HC3: 30 in IM-S-1 and 30 in IM-S-3).

**Figure RL6.** (B) Sankey plot illustrating relationships between hierarchical clusters and
proteomic clusters or immune clusters; (C) Sankey plot illustrating relationships between
sarcoma histology subtypes and proteomic clusters or immune clusters.

In the revision, we performed further analysis to decipher the diverse proteomic and
immune features within one hierarchical clustering. Specifically, we focused on HC3
which showed proteomic and immune environment diversity and could be clustered
into 2 proteomic clusters (PC-Ra and PC-Cc) and 2 immune clusters (IM-S-1 and IM-
S-3). According to the distribution of HC3 in proteomic and immune clusters, we
classified HC3 into 4 subgroups: HC3-Ra-IM1, HC3-Ra-IM3, HC3-Cc-IM1, and HC3-
Cc-IM3 and performed further analysis to illustrate the potential link between
proteomic and immune features. As a result, comparing the proteomic features among
the four subgroups revealed that although comparing to HC3-Cc-IM1 and HC3-Cc-IM3,

both HC3-Ra-IM1 and HC3-Ra-IM3 showed elevated expression of MAPK10 which
 is the distinctive feature of PC-Ra, the protein expression of MAPK10 was significantly
 higher in HC3-Ra-IM3 (**Figure RL6D**). Meanwhile, comparing the immune features
 among the four subgroups, we observed that the immune scores of HC3-Cc-IM3 and
 HC3-Ra-IM3 were obviously higher than the other two subgroups (**Figure RL6E**).
 Intriguingly, the enrichment of CD4⁺ T cells and the immune checkpoint protein CD274
 was obviously higher in HC3-Ra-IM3 ((**Figure RL6F-G**).

The above observations implied the potential link between MAPK10 and elevated
 expression of CD274. Since MAPK10 is a kinase, to illustrate the mechanism
 underlying this potential link, we screened the phosphorylation level of MAPK10's
 substrates and found the phosphorylation of the CTNNB1 at Ser675 was significantly
 correlated with both the protein expression of MAPK10 and CD274. Previous
 researches have reported that the phosphorylated CTNNB could interact with
 transcription factor and promote the transcription of CD274 (*J Exp Med*, PMID:
 32860047). Thus, the elevated expression of CD274 was probably led by the MAPK10-
 mediated phosphorylation signal transduction. Our results revealed the diverse
 proteomic features and immune features within one HC, and further indicated the
 potential link between them.

D

E

F

**Figure RL6.** (D) Boxplots illustrating enrichment of protein abundance of MAPK10, the
 phosphorylation level of CTNNB1 Ser675, and the signaling pathway, “positive regulation of
 MAPK cascade” in the HC3-Ra-IM3. (E) Boxplot illustrating enrichment of CTNNB1 Ser675
 in HC3-Ra-IM3. (F) Scatter plots present positive correlations between phosphorylation level
 of CTNNB1 Ser675 and protein abundance of MAPK10, CD274, or CD80 in HC3 group.

**Figure RL6.** (G) Boxplots illustrating the enrichment of CD4⁺ T cell signature in HC3-Ra-IM3.
 Noticeably, besides the heterogeneity within one HC, the diverse immune features
 between HCs were also observed. Specifically, HC5 (UPS) and HC6 (LMS) were both
 clustered into PC-Cc and featured with fast tumor cell proliferation, which could be
 confirmed by the elevated cell proliferation index (**Figure RL6H**). Yet, the two HCs
 showed distinctive immune features. Particularly, the HC5 showed elevated CD8⁺ T
 cell infiltration (**Figure RL6I**). To illustrate the potential mechanism, we compared the
 protein expression and pathway enrichment scores of immune-related processes
 between HC5 and HC6. As a result, we observed the dominant enrichment of the TCR
 signaling pathway in HC5, and TCR-related proteins such as PTPN6, NFKBIE, IKBKG,
 BCL10, etc. were significantly elevated in HC5 (**Figure RL6I**). These observations
 suggested that even presenting the same proteomic features, the hierarchical clusters
 could have different TME features, which supported the necessity of clustering from

different levels.

**Figure RL6.** (H) The boxplot presents proliferation index in different hierarchical clusters; (I)
The heatmap presents the enrichment of CD8⁺ T cells, T cell receptor signaling pathway, and
related proteins in HC5.

In sum, we performed clustering from three aspects: histology (hierarchical clustering),
proteome, and immunology. From the hierarchical clustering, we found the similarity
of variable histological subtypes of sarcoma. From the proteomic clustering, we found
key kinases and biological pathways to distinguish sarcoma patients. From the
immunology clustering, we uncovered TME heterogeneity of sarcoma and clinically
related immune features. Integration of these three clustering systems could give a more
comprehensive definition of sarcoma subgroups and present their specific
characteristics. In the revised version, we updated our statements on **Figure 6** and
**Supplementary Figure 16** and added more details on how the three clustering systems
are associated with each other on lines 790-816 of the “**Results**” section.

**8. There are some typos and some sentences are not well constructed or are unclear.**
**This is particularly noticeable in the discussion. Some examples below:**

**Response:**

We appreciate the reviewer’s comments and revised the typos and sentences.
Specifically, our main revisions are presented as follows:

**Line 73. “A potential explanation is that these mechanisms could not reflect the**

**functional effects, as they reside many regulatory layers away from the protein.”**

**Response:**

Thanks for the comments, we have rewritten the sentence as follows: “A potential
explanation for this phenomenon is that previous researches focus on genomic or
transcriptomic data, which could not panoramically reflect the molecular features of
STS.”

**Line 106. “It is necessarily required for immune therapy that more detailed
information about the characteristics of immune infiltration and the effective
immune components.”**

**Response:**

We appreciate the suggestion and we have revised the sentence to the following
sentence: “To enhance the efficiency of immune therapy, it is important to characterize
the diverse immune cell infiltration signatures of STS and to uncover the heterogeneity
of TME in STS.”

**Line 594. “When considering targeting the molecular in the TGFβ signaling
pathway (such as SHC1), ES might have a similar response with AS.”**

**Response:**

Thanks for the comment, we have revised the sentence as the following sentence: “Our
data revealed that ES and AS patients might benefit from SHC1 targeting therapy.”

Besides the above, we have also carefully revised lines 49-119 in the introduction, and
lines 819-924 in the discussion section, please see the revised manuscript for details.

**Minor comments:**

**1. Line 257: “We found the activity level of two pathways enriched in HC1, actin
cytoskeleton reorganization (Pearson’s correlation, $r = 0.21$ p-value = 0.0049) and
epithelial cells migration (Pearson’s correlation, $r = 0.22$ p-value = 0.0027),
changed *tightly followed the abundance variation of SHC1 (Figure 2F).**

I would not say there is a *tight correlation between SHC1 abundance and
Epithelial cell migration. It is just a correlation.

Response:

We appreciate the reviewer for this helpful suggestion and we have changed the
sentence to the following sentence: “We found there’re significantly positive
correlations between protein abundance of SHC1 and two HC1-enriched biological
pathways, actin cytoskeleton reorganization (Pearson’s correlation, $r = 0.21$ p-value =
0.0049) and epithelial cells migration (Pearson’s correlation, $r = 0.22$ p-value = 0.0027).”

**2. Fig. 2K it would be better to show in the plot the IC50 of all cell lines individually
and they are only 6.**

Response:

We thank the reviewer for the comment and we have labeled in the plot the IC50 of all
cell lines individually in Figure 2K (Figure RL7A-C).

**Figure RL7.** (A) Dose-response curves (left panel) and IC50 values (right panel) of
carbamoylcholine (the SHC1 inhibitor) in AS (blue), ES (purple), and WDLPS (brown) cell
lines. (B-C) The bar plots indicated the IC50 of six cell lines to SHC1 inhibitor (B, the
comparison among 6 distinct cell lines, C, the comparison between HC1 and HC2 cluster).

**3. The literature references are not always correct. For example, reference for
CellX should be 62 (not 63 as mentioned in the text).**

Response:

We thank the reviewer for the comment and we have carefully checked the reference
and revised the citations accordingly. Furthermore, we have revised all the citations
through the manuscript.

**Reviewer #2 (Remarks to the Author): Expert in tumour immunology and**
**immune landscapes in sarcoma**

**The present study undertakes a comprehensive proteomic profiling of 272 STS**
**patients representing 12 major subtypes. The authors identify six subtypes on the**
**base of hierarchical classification, three subtypes based on proteomic analysis and**
**three subtypes based on immune signatures. For some clusters they identified**
**some mechanism/s relevant for patient prognosis. Interestingly, some of the main**
**mechanisms identified with bioinformatics approaches are verified by wet**
**laboratory experiments.**

**The study provides a valuable proteomic resource for the scientists working on**
**sarcomas. The study is correctly written, although the logic of the analyzes carried**
**out is not always fluent and sometimes it is difficult to follow.**

**Moreover, there are some concerns:**

**1) The three main clustering analysis should be performed also taking into account**
**the anatomical site distribution and the therapies applied to the patients, in order**
**to verify if the clustering may be influenced by the location of the tumor or by the**
**therapy.**

**Response:**

**We thank the reviewer for the suggestions. We agree with the reviewer that anatomical**
**site distribution as well as the therapies applied to the patients should be like into**
**account. In the revision, in order to explore whether the proteomic clustering was**
**influenced by anatomical site distribution etc., comparative analyses were made among**
**the 3 proteomic clusters, respectively.**

**1. Correlation of proteomic clustering and anatomical site distribution**

**For the samples collected in our cohort, their anatomical sites could be classified into 5**
**different locations: extremity (E), head and neck (H), Intraabdominal /pelvis**
**/retroperitoneum/visceral (IB), Intrathoracic/mediastinal (IT), and Trunk. To assess the**
**intersection of our proteomic clusters with anatomical sites, we performed a correlation**
**analysis between proteomic clusters with anatomical sites. As a result, there was no**

significant difference in anatomical sites among the proteomic clusters (p-value = 0.381,
Chi-square test).

2. Correlation of proteomic clusters and drug treatment

All the 272 samples collected in our study were from treatment-naïve patients. All the
patients received primary resection for sarcomas without any anti-cancer treatments
prior to surgery. Postoperative surveillance and treatment were conducted consistently
according to Zhongshan Hospital’s guidelines. Specifically, 64 patients received
chemotherapies, and 27 patients received target therapies after sugary. We compared
the overall survival between patients with and without postoperative treatments, and
observed no significant difference (Log-rank test, p-value > 0.1). We further performed
a correlation analysis between postoperative treatment and our proteomic clusters to
assess. As a result, there was no significant difference in the distribution of postoperative
treatment among the proteomic clusters (p-value = 0.633 (target therapy) & 0.077
(chemotherapy), Chi-square test, **Table RL3**).

In addition, statistical analysis uncovered that there’s no significant difference of age
and gender among proteomic clusters (p-value = 0.264 (age) & 0.916 (gender), Chi-
square test, **TableRL3**). These results indicated that the proteomic clustering is an
independent risk factor of the prognosis, which could be better to predict the survival
time.

**Table RL3.** The baseline characteristics of patients belonging to different proteomic
clusters.

	Level	PC1	PC2	PC3	p	test
	n	86	122	64		
age (median [IQR])		55.00 [43.50, 64.00]	56.00 [47.00, 63.75]	58.50 [49.75, 64.25]	0.264	nonnorm
gender (%)	female	44 (51.2)	66 (54.1)	34 (53.1)	0.916	
	male	42 (48.8)	56 (45.9)	30 (46.9)		
target therapy (%)	no	42 (82.4)	69 (85.2)	48 (88.9)	0.633	
	yes	9 (17.6)	12 (14.8)	6 (11.1)		
chemotherapy (%)	no	30 (58.8)	50 (61.7)	42 (77.8)	0.077	
	yes	21 (41.2)	31 (38.3)	12 (22.2)		
location (%)	E	23 (26.7)	32 (26.2)	24 (37.5)	0.381	
	H	5 (5.8)	2 (1.6)	2 (3.1)		
	IB	35 (40.7)	59 (48.4)	21 (32.8)		
	IT	11 (12.8)	10 (8.2)	6 (9.4)		
	Trunk	12 (14.0)	19 (15.6)	11 (17.2)		

In sum, there was no significant difference in the distribution of the anatomical site
distribution or the therapies applied to the patients among the proteomic clusters. In the

revision, we have updated these comparative analyses in the “**Result**” section on lines
439-444 in the revised manuscript.

**2) in figure 5A complement and coagulation cascade pathways are enriched in the**
**IM-S-1 cluster corresponding to the stroma-enriched subtype and B cells in the**
**IM-S-2. Results already published on the role of complement activation and B cells**
**in sarcomas (doi: 10.1038/s43018-021-00173-0 and doi: 10.1038/s41586-019-1906-**
**8) should be mentioned and discussed. Are the main findings of these two papers**
**true by proteomic point of view? For example, is the C3aR or complement soluble**
**proteins/receptors expression associated with M2-like macrophages and/or UPS**
**patient survival? Are B cell markers associated with increased overall survival?**
**Do they correlate with metastasis?**

**Response:**

We thank the reviewer for the instructive suggestion. As commented by the reviewer,
in our research, by performing immune cell deconvolution and immune features-based
clustering, we classified our pan-sarcoma dataset into 3 immune subtypes with
distinctive immune characteristics. Specifically, we found the IM-S-1 featured with
complement and coagulation cascade, and IM-S-2 featured with B cell enrichment.
Following the reviewer’s suggestion, we summarized the main findings of the two
transcriptomic papers and added further analysis to investigate whether their findings
on sarcoma tumor microenvironment could be validated at the proteomic level. The
detailed analyses for each paper were presented as follows:

**1. About the impact of complement activation on sarcoma progression.**

The first paper conducted by Magrini and colleagues performed a systematic
assessment of complement activation and effector pathways in sarcomas. Their main
findings were: (1) they utilized a mice model and found that C3 and its receptor C3aR
promoted 3-MCA-induced sarcoma genesis; (2) they found that C3 and C3aR
participated in macrophage recruitment; (3) they used TCGA data to confirm C3
deficiency-associated signatures of macrophages related to favorable prognosis. We
then investigated their main conclusions in our data. Particularly:

**1.1 The expression of C3 is elevated in the tumor tissues of our sarcoma cohort.**
Since the Magrini, et.al. reported the role of C3-C3aR in sarcoma genesis, especially in
UPS, we then evaluated the expression of C3 and C3aR in the tumors and NATs. As a
result, we observed significantly elevated expression of C3 in our pan-sarcoma cohort
(**Figure RL8A**). We further evaluated the expression of C3 in the 12 histological
subtypes and observed the protein expression of C3 was significantly elevated in tumors
of DDLPS, MLPS, MFS, et al (**Figure RL8A**).

**Figure RL8. Impacts of C3-C3aR axis and B cell markers from the proteomic viewpoint.**

(A) Boxplots illustrate the proteomic expression of C3 in NATs and tumors.

As for C3aR, its expression was detected in 3 samples, thus it was excluded for further
analysis. This might be caused by the fact that the C3aR is a membranal protein and is
enriched in macrophages. Since Magrini et al. have reported that C3aR promotes
sarcoma progression through lectin pathway, we further evaluated the enrichment of
lectin signaling pathway between tumors and NATs, and found that the enrichment
scores of lectin signaling pathway were significantly higher in tumors of our pan-
sarcoma cohort, and in tumors of histological subtypes, like UPS, DDLPS, MFS, et al.
(**Figure RL8B**). These results revealed the elevation of C3 and lectin signaling
pathways in sarcoma tumor tissues, especially in UPS at the proteomic level, and
confirmed its role in sarcoma genesis.

**Figure RL8.** (B) Boxplots present the enrichment scores of the lectin signaling pathway in
 NATs and tumors.

**1.2 The protein abundance of C3 presents a positive correlation with macrophage**
 **signature.**

Another major finding reported by Magrini, et al. is the role of C3 in macrophage
 recruitment. To verify this conclusion in our pan-sarcoma proteomic cohort, we
 performed a correlation analysis between the protein expression of C3 and macrophage
 signature. As a result, the protein expression of C3 was observed to be positively
 correlated with the enrichment of macrophages in our pan-sarcoma cohort and in
 histological subtypes LMS, SS, WDLPS, and AS (**Figure RL8C, D**). These results
 indicated the role of C3 in recruiting macrophages in sarcomas.

**Figure RL8.** (C) The scatter plot presents a positive correlation between C3 protein abundance
 and macrophage signatures in pan-sarcoma; (D) Scatter plots presents a positive correlation
 between C3 protein abundance and macrophage signatures in LMS, SS, WDLPS, and AS.

**1.3 The C3 deficiency-associated signatures of macrophages is related to patients'**
**favorable prognosis.**

We first evaluated the prognostic relevance of the C3's protein expression. As a result,
the protein expression of C3 showed no significant correlation with patients' prognosis
(**Figure RL8E**). Consistently, Magrini and colleagues evaluated the correlation
between mRNA expression of C3 and prognosis utilizing transcriptomic data in TCGA
SARC and they also didn't get access to a positive result. Then to further investigate
the prognostic role of C3, Magrini, et al. focused on C3-recruited macrophages and
established a signature to represent sarcoma-infiltrated macrophages with the C3-
deficiency phenotype and evaluated its prognostic relevance. As a result, they found
high C3-deficiency macrophage signature was associated with increased overall
survival times in TCGA SARC cohort. To estimate whether this finding could be
confirmed at the proteomic level, we calculated the C3-deficiency macrophage
signature utilizing the proteomic data in our cohort following the same method as
Magrini, et al. described. As a result, we observed a significantly positive association
between the C3-deficiency macrophage signature and patients' favorable outcomes in
our pan-sarcoma cohort (**Figure RL8F**). These results confirmed the C3 deficiency-
associated signature of macrophages is related to patients' favorable prognosis at the
proteomic level and implies the potential of using C3 deficiency-associated signatures
of macrophages as the prognostic index for sarcoma in the future.

**Figure RL8.** (E) Kaplan-Meier curve for OS stratified by C3 proteomic abundance in pan-
sarcoma; (F) Kaplan-Meier curves for OS and DFS stratified by levels of C3-deficiency
macrophage signatures in pan-sarcoma;

**2. About the B cells are associated with survival in sarcoma.**

The second paper conducted by Petitprez, et al. presented an immune classification of
soft tissue sarcomas and identified B cells as a prognostic factor for sarcomas. We then
evaluated the prognostic relevance of B cells in our dataset.

**2.1 The enrichment of B cells is associated with the prognosis in specific sarcoma**
**histological subtypes.**

We first estimated the enrichment of B cells using ssGSEA algorithm based on B cell
signatures from xCell (*Genome Biol*, PMID: 29141660) and investigated its association
with patients' prognosis. As a result, although we didn't observe a significant
association between B cell enrichment and patients' prognosis in our whole pan-
sarcoma cohort, we observed that LMS, UPS, MFS, and AS patients with high B cell
signatures trended to have longer overall survival times (**Figure RL8G**).

**Figure RL8.** (G) Kaplan-Meier curve for OS stratified by B cell signatures in LMS, UPS, MFS,
and AS.

**2.2 Some B cell markers are relevant to prognosis at the proteomic level.**

Based on Petitprez et al.'s finding, which indicated the prognostic relevance of B cells
in sarcomas, we further evaluated the clinical applicability of utilizing specific B cell
markers as a prognostic index for sarcomas. We investigated the association between
the protein expression of B cell markers (referring to the human cell marker database

[http://xteam.xbio.top/CellMarker/]) and patients' prognosis. As a result, among the 12
 B cell markers that have been detected in our dataset, 7 B cell markers showed
 significant association with patients' prognosis in our pan-sarcoma cohort. We further
 verified the prognostic relevance of these 7 B cell markers in TCGA dataset and found
 3 out of 7 B cell markers (PTPRC, CD9, IGLL5) showed consistent prognostic
 relevance at the transcriptomic level in TCGA SARC cohort (*Cell*, PMID: 29100075)
 (**Figure RL8H**). These results imply the potential clinical utilization of these B cell
 markers for prognostic prediction in future.

**Figure RL8 (H)** Kaplan-Meier curve for OS stratified by B cell markers. top: our cohort,
 stratified by proteomic abundance; bottom: TCGA SARC cohort, stratified by mRNA
 expression.

 We thank the reviewer for suggestions about the comparative analysis between our
 study and these two researches. In the revision, we investigated the major findings of
 those two papers in our proteomic data. As a result, we verified the high expression of
 C3 in sarcoma tissues and the positive correlation between C3 protein abundance and
 macrophage signature. Utilizing proteomic data to establish the C3-deficiency
 macrophage signature, we further proved the availability of this signature in predicting
 prognosis. Meanwhile, we found high B cell signature is correlated with increased
 overall survival times in specific sarcoma histological subtypes, especially in MFS.
 Specific B cell markers, including PTPRC, CD9, and IGLL5, have prognostic relevance

at both proteomic and transcriptomic levels. We have updated the above findings in the
“**Discussion**” sections of our revised manuscript on lines 858-885 and **Supplementary**
**Figure 17**.

**3) The authors should discuss some limitations of the study, such as:**

- **the requirement of future validation in independent cohorts.**

- **considering the extensive intra-tumoural heterogeneity, the inability of bulk**
**proteomic approach to dissect the contribution of distinct heterogenous tumour**
**regions.**

- **the study is based on localised disease, thus it will have to be determined if these**
**findings will be true also for locally relapsed and metastatic tumours.**

**Response:**

Thank the reviewer for the comment. We have added Limitations in the discussion
section of the revised manuscript as following:

***Limitations***

The aims of this study were to provide a proteomic and phosphoproteomic landscape
to decipher the molecular heterogeneity of sarcomas, the prognosis-related markers,
and abnormally changed biology pathways. There are some limitations due to the
sample collection and technology as follows:

(1) The sarcoma cohort in this study is single-centered from Fudan University,
Zhongshan Hospital and included only Chinese patients, so the conclusions may
lead to potential selection bias. Additional prospective studies are needed to
validate our findings in multi-center and cohort of other ethnicities.

(2) We found specific subtype-enriched proteins which might be serviceable in early
diagnosis and histological subtype detection, but we couldn't exclude the
possibility that this protein could have stemmed from other affected organs or may
be indirectly induced by the effects of the tumors on their microenvironment or
even systemically. Further experiments or clinical data are necessary complement
to validate the roles of this proteins in sarcoma.

(3) The proteomic data in this study was generated through bulk proteomic approach

from tumor and NAT tissues and couldn't fully reflect the heterogenous tumor
regions and the tumor-NAT boundary regions. Integrating single cell and spatial
omics would be useful to further explore the intra-tumoral heterogeneity in the
future research.

(4) The samples in this study were all collected from treat-naïve patients and were all
primary tumors without remote metastasis or local relapse. The information about
metastasis and local relapse come from 60-month follow up. The conclusion in this
study that SHC1 and MAPK10 promotes metastasis required further confirmatory
studies on metastatic samples. Other conclusions were also just based on localised
diseases, it will have to be determined if these conclusions are also tenable in
locally relapsed and metastatic tumors.

Please see the details on the end of the '**Discussion**' section in the revised manuscript.

**Reviewer #3 (Remarks to the Author): Expert in MS-based cancer proteomics**

**Comments on "Proteomic characterization identifies clinically relevant subgroups**
**of soft tissue sarcoma" by Tang et al.**

**The authors present proteome data from 272 soft tissue sarcoma tissues and 91**
**matched tumor-adjacent tissues (total of 363 samples). In addition,**
**phosphoproteome data were generated from 138 sarcoma and 24 tumor-adjacent**
**tissues. Data analysis is based on clustering the data, extract functional predictions**
**from the clusters, and follow-up with some cell line experiment to understand the**
**role of top-scoring proteins in the specific functional categories. The authors are -**
**in general – overstating the evidence from the molecular mechanisms they are**
**interrogating (see comments). Overall, I did not find the study to be very exciting.**
**I think that Nature Communications is a good place for resource-style papers like**
**this, and proteomics studies on soft tissue sarcoma have the potential to help us**
**better understand the diseases and to identify new treatment strategies. Also, 361**
**sample is a quite large number. What I am missing is evidence that proteomics is**
**adding crucial information beyond what we know about the disease. I also think**

**that the follow-up experiments need more depth. I am on the fence regarding**
**recommending to consider a publication after major revisions, but I am happy to**
**look the manuscript after the below comments have been addressed.**

**(1) The authors state that 15,552 proteins were identified across all samples with**
**an average of 5,593 proteins being quantified per sample on average. It is very**
**unlikely that 5.5 k proteins per sample using unfractionated sample leads to a total**
**of > 15 k proteins across 363 samples. I wonder if the false-discovery filtering at**
**the protein level was done for each individual sample but not for all datasets**
**combined. It is the latter, that should have been done. Merely filtering for each**
**individual run will greatly inflate the protein FDR for the entire dataset (as false**
**assignments will be different for each run). It is also not clear if a parsimony**
**filtering was used on the identified proteins. This should also be done the combined**
**dataset. The same question applies to the phosphoproteomics analysis: was the**
**filtering done on the combined dataset (which it should have been) or only on each**
**individual dataset?**

**Response:**

We sincerely thank the reviewer for the comment and apologize for the unclear
description of protein identification methods in our previous manuscript. In our
research for each experiment, we employed “Firmiana” a one-stop proteomic cloud
platform (*Nat Biotechnol*, PMID: 28486446) for protein quantification. To optimize
the number of proteins identified, we applied a very stringent filter with 1% FDR at the
peptide level and 1% FDR at the protein level. The same cutoff strategies of FDR at
protein/peptide level based on label-free quantification have been widely used in recent
researches (*Nature*, PMID: 30814741; *Cell*, PMID: 32649877; *Nat Commun*, PMID:
28429721; *Nat Commun*, PMID: 29520031). As a result, an average of 5,593 proteins
was quantified per sample. To count the total identified proteins, we combined all the
experiments and 15,552 proteins were observed, the combined number of identified
proteins was only utilized for presenting the detected protein numbers, but not utilized
for further analysis.

In concordant with the reviewer’s suggestion, for all the analyses including hierarchical
 cluster, proteomic subtyping, tumor microenvironment analysis, etc. we utilized a
 protein matrix that applied 1% FDR filtering at the protein level for all datasets, which
 contained 10,118 proteins in total. We further referred to recently published proteomic
 cohort researches of different cancer types and compared cohort sample size, the
 average number and the total number of identified proteins between these researches
 and our study. As a result, both the average and total identified protein numbers were
 comparable with no significant differences between our study and previously reported
 samples (**Ref1, Ref2, Ref3, Ref4, Ref5 in Table RL4**).

**Table RL4. The total and average protein numbers of recent proteomic studies and our**
 **study**

Reference	Journal	Quantification method	Cancer & size	Average proteins	Total proteins	PMID
Our study		LFQ	Sarcoma (272 tumors)	5,593	10,118	
Ref1	Nature	LFQ	Hepatocellular carcinoma (110 tumors)	5,953	9,142	PMID: 30814741
Ref2	Blood	TMT	Acute myeloid leukemia (44 tumors)	5,664	10,651	PMID: 35895896
Ref3	Cell	LFQ	Lung adenocarcinoma (103 tumors)	6,682	11,091	PMID: 32649877
Ref4	Cell	SILAC	Melanoma (116 tumors)	4,500	10,376	PMID: 31495571
Ref5	Cancer Cell	TMT	Intrahepatic cholangiocarcinoma (262 tumors)	5,690	10,529	PMID: 34971568

Moreover, for the phosphoproteomic analysis, a label-free based quantification analysis
 was performed using Proteome Discover (version 2.3) (*Cell*, 2020).
 Phosphorylation sites were localized with ptmRS module (*J Proteome Res*, 2011).
 Peptide spectrum matches (PSMs) were filtered with 75% localization probability for
 all phosphorylation sites were included for further analysis. For global
 phosphoproteomic analysis, the FDR at the peptide level and the protein level were also
 set as 1%. In total, 37,842 phosphosites belonging to 6,483 phosphoproteins were
 identified (an average of 7,912 phosphosites belonging to 3,120 phosphoproteins for
 each individual experiment). We also compared our results with previous published
 researches. As a result, the number of average and total number of identified
 phosphosites and phosphoproteins were also comparable with those published
 researches (**Ref1, Ref2, Ref6 in Table RL5**).

**Table RL5. The phosphosite and phosphoprotein numbers of recent proteomic studies**
 **and our study.**

Ref	Journal	Quantification method	Cancer & size	Average phosphosites	Total phosphosites	Average phospho-proteins	Total phospho-proteins	PMID
Our study		LFQ	Sarcoma (272 tumors)	7,912	37,842	3,120	6,483	
Ref1	Nature	LFQ	Hepatocellular carcinoma (110 tumors)	8,941	22,564	1,485	5,277	PMID: 30814741
Ref2	Blood	TMT	Acute myeloid leukemia (44 tumors)	11,817	29,201	3,609	5,407	PMID: 35895896
Ref6	Cell Reports Medicine	TMT	Ovarian HGNC (83 tumors)	11,331	38,194	4,006	7,080	PMID: 32529193

 In sum, the combined proteome was only used for presenting the detected protein
 numbers. As for downstream bioinformatic analysis, the proteomic and
 phosphoproteomic matrix that have applied FDR filtering for all dataset were utilized,
 thus our main findings remained unchanged. In the revision, we have added the number
 of proteins that have applied FDR filtering for all datasets and utilized for analysis:
 “Proteomic analysis identified 15,552 proteins in total, with 5,593 proteins per sample
 on average. We then applied FDR filtering for all datasets, and 10,118 proteins were
 utilized for further analysis.” We have also added the description of the protein and
 phosphoprotein identification in the “Methods” section, as follows: “For conducting
 bioinformatic analysis, the proteomic/phosphoproteomic datasets, that have applied
 FDR filtering for all datasets were utilized.” Please see 153-155 lines in the “**Result**”
 section and 1052-1055 lines in the “**Methods**” section of the revised manuscript for
 details.

 **(2) Peptides/proteins were quantified using a label-free approach (iBAQ).**
 **Reproducibility is shown in Supp Fig 1 A. I would like to see the median CV across**
 **all the HEK standard samples as well as the CV in dependence to the signal-to-**
 **noise ratio.**

**Response:**
 We sincerely appreciate the reviewer’s comment. In our previous manuscript, for the
 quality control of MS performance, the HEK293T cell lysate was measured every three
 1342 days as the quality control standard. A pairwise Pearson’s correlation coefficient was
 1343 calculated for all quality control runs, and the results showed the median correlation

coefficients of proteome standards were 0.9 (0.85-0.95). We also referred to previously
published works, and the correlation coefficient of the standards was comparable to our
results (**Ref 1, Ref 2, Ref3, Ref4, Ref5, Ref6, and Ref7**) (**Table RL6**).

To comprehensively respond to the reviewer's comment, we divided the responses into
two parts:

**1. About the coefficient variations (CVs) across all the HEK standards.**

In the revision, we have calculated the coefficient of variation (CVs) across the 15 HEK
standards. As a result, the median CVs of HEK293 standards were 0.23 and the mean
CVs were 0.29 (**Figure RL9A-B**). We also referred to previously published researches,
and the CVs across replicates of previously published researches were also comparable
to our results (**Table RL6**)

Figure RL9. CVs and signal-to-noise of the proteomic data

(A) The scatter plot illustrates the CV of each protein (using iBAQ) across all HEK293 stand
samples. (B) Cumulative distribution curve illustrating the distribution of CVs.

**Table RL6. The standards' correlations and CVs of our work and recently published**
**studies**

Reference	Journal	Quantification method	CVs	The average correlations of standards	PMID
Our study		LFQ	median:0.23; mean:0.29 (based on iBAQ) median:0.23; mean:0.30 (based on S/N ratio)	0.9	
Ref1	Cell	LFQ	median:0.31; mean:0.36	0.87	PMID: 31585088
Ref2	Nature	LFQ	median:0.35; mean:0.44	0.93	PMID: 30814741
Ref3	Cell	LFQ	median:1.02; mean:1.28	0.91	PMID: 32649877
Ref4	Nature	LFQ	median:0.73; mean:1.18	0.95	PMID: 25043054
Ref5	Cell Reports	TMT	median:0.85; mean:1.02	0.85	PMID: 33086064
Ref6	Cell	TMT	median:0.99; mean:1.52	0.91	PMID: 33212010
Ref7	Nature	iTRAQ	median:1.29; mean:1.69	0.88	PMID: 27251275

**2. About the CVs independence to signal-to-noise (S/N) ratios.**

In the revision, following the reviewer’s comments, we calculated the CVs across the
 HEK293 standards based on signal-to-noise (S/N) ratios. As a result, the median CVs
 was 0.23 and the mean CVs was 0.30 (**Figure RL9C-D**), which were similar to the
 median and mean CVs calculated based on iBAQs. The correlation between the iBAQ
 based CVs and S/N ratio based on CVs was around 0.95 (Spearman correlation $p <$
 0.05). These results confirmed the reproducibility for repeat experiments, and
 demonstrated the consistent stability of our MS platform.

**Figure RL9.** (C) The scatter plot illustrates the CV of each protein (using S/N ratio) across all
 HEK293 stand samples. (D) Cumulative distribution curve illustrating the distribution of
 CVs.

In sum, we have added the results of CVs in the supplementary figure X of the revised
 manuscript, and added the methodologies of CV analysis on lines 161-164 of the
 “**Methods**” section and 161-164 lines of the ‘**Result**’ section.

**(3) Supp Fig 2 A. The PCA plot shows quite an overlap of NAs and tumor samples.**
**It would be great to see an unsupervised clustering of NAs and tumor sample and**
**some cluster purity measurement to evaluate the separation of tumor and normal**
**samples.**

**Response:**

Thanks for the constructive comment. To systematically respond to the reviewer's
comments, we divided the response into 3 parts:

**1. The criteria for sample collection and assessments**

In this study, for tumor samples, 272 formalin-fixed, paraffin-embedded (FFPE)
sarcoma tumor tissues and 91 paired tumor-adjacent tissues were acquired from
Zhongshan Hospital, Fudan University from 2010 to 2019. One 4 μm thick slide from
each FFPE block was sectioned and stained by hematoxylin and eosin (H&E) for
histological evaluation. Specifically, each tumor/ tumor adjacent sample was checked
by three expert pathologists to confirm the sample quality according to the following
criteria:

For tumor samples: (1) pathologists evaluated and defined tumor area on the slices of
FFPE specimens with tumor cell rate (tumor purity) > 90%; (2) the histological
subtypes of sarcoma were diagonalized by pathologists according to WHO
classification of Soft Tissue & Bone Tumor (*Adv Anat Pathol*, PMID: 32960834). As
for tumor-adjacent samples: (1) pathologists evaluated and defined the tumor-adjacent
areas on the slices of FFPE specimens with no tumor cell rate; (2) NATs were chosen
based on tumor locations and the original lineages of tumors for different histological
sarcoma subtypes, according to WHO classification of Soft Tissue & Bone Tumor (*Adv*
*Anat Pathol*, PMID: 32960834).

**2. Unsupervised clustering of NATs and tumor samples.**

In agreement with the reviewer, an unsupervised clustering of tumors and NATs could
help to illustrate the separation of tumor samples and NATs. Thus, in the revision, we
conduct unsupervised consensus clustering of NAT and tumor samples with the
ConsensusClusterPlus R package (*Bioinformatics*, PMID: 204275). The following

detail settings were used: number of repetitions = 1,000 bootstraps; pItem = 0.8
 (resampling 80% of any sample); pFeature = 1 (resampling 100% of any protein);
 clusterAlg = “K-means”; and distance = “Euclidean”. As a result, 2 clusters were
 determined based on the average pairwise consensus matrix within consensus clusters,
 the delta plot of the relative change in the area under the cumulative distribution
 function (CDF) curve, and the average silhouette distance for consensus clusters.

We then calculated specificity and purity to evaluate the distribution of tumors and
 NATs and tumors in these 2 clusters (cluster1: NAT-distance and cluster2: NAT-similar)
 (**Figure RL10A**). Specifically, for sample’s specificity, the following formula was
 utilized: $\text{specificity} = \max \{N_{c1}/N_{\text{total}}, N_{c2}/N_{\text{total}}\}$. N_{total} means the whole number of
 tumors or NAT samples. N_{c1} and N_{c2} mean the samples belonging to cluster1 or cluster2
 in N_{total} . As for cluster purity, the following formula was utilized: $\text{purity} = \max \{C_N/C_{\text{total}},$
 $C_T/C_{\text{total}}\}$. C_{total} means the whole number of cluster1 or cluster2. C_N and C_T means the
 numbers of tumors or NATs in C_{total} . As a result, in concordant with the PCA analysis,
 around 89% of the NATs were grouped into cluster1, and 56% of the tumors were
 grouped into cluster2. Forty-four percent of tumors were grouped with NATs, implying
 that these tumors might not show significantly diverse proteomic features compared to
 NATs (**Figure RL10B**). The unsupervised clustering confirmed the results of PCA
 analysis, we then tried to illustrate the potential reasons under this phenomenon. Since
 our cohort contained 12 histological types of sarcomas and NATs paired with them also
 included various tissue types, we then hypothetically assumed that the overlap between
 tumors and NATs might be caused by the diverse tumor heterogeneity of different
 histological subtypes of sarcoma.

Figure RL10. Unsupervised clustering of NATs and tumor samples

(A) The table about unsupervised clustering results of NATs and tumor samples;

(B) The PCA result of NATs and tumor samples.

3. The overlap between tumors and NATs in PCA analysis might be caused by the tumor heterogeneity of different histological subtypes of sarcomas

To illustrate whether the overlap between tumors and NATs in PCA analysis was associated with different histological subtypes of sarcomas, we conducted PCA analysis for each histological type of sarcomas, separately. As a result, the tumors were perfectly separated with NATs in each histological type of sarcomas. The representative PCAs are shown in **Figure RL10C**.

Figure RL10. (C) PCA plots illustrate separation levels between NAT and tumor samples in

histological subtypes.

These results confirmed our assumption that the overlap between tumors and NATs was caused by the tumor heterogeneity of diverse histological sarcomas, further revealed the value of research in deciphering the tumor heterogeneity of different histological sarcomas. In the revision, we have added the histological type-based PCA analysis for tumors and NATs in **Supplementary Figure 3**. Meanwhile, we added the above analysis on lines 169-187 of the **‘Result’** section.

(4) What criteria were used to define the clusters (HC1-6)? This is not clear based on the dendrogram alone. The dendrogram implies that there was very clean clustering histological subtypes. I am missing a plot showing how well the subtypes

**were separated from each other using unsupervised clustering (see also comment**
**3).**

**Response:**

We thank the reviewer for the critical comment. We apologize for the unclear
presentation of the clustering cutoffs and details in our previous manuscript. To
systematically response to the comment, we will address this comment from 3 aspects:

- 1. The process to create the dendrogram;
- 2. The criteria to determine the cluster number;
- 3. Biological insights based on hierarchical clusters.

**2. The process to create the dendrogram**

To investigate the intrinsic common features of STS histological subtypes, we
employed hierarchical clustering on the 12 STS histologic subtypes. R (version 4.2.0)
and the R package “factoextra” (version 1.0.7) were utilized for data process and
visualization.

Firstly, we performed ANOVA analysis to filter proteins with high variable values
among different histology subtypes. The protein expression matrix had been processed
as described in the “**Method**” section of the manuscript. 2536 proteins were finally
filtered out with less than 0.001 p-values. Then, we calculated the mean values of these
filtered proteins for each sarcoma histology subtype. The “Pearson” distances between
each two subtypes were calculated utilizing these mean values (**Supplementary Table**
**2**). Next, based on the “Pearson” distances, we created the dendrogram with “hclust”
and “fviz_dend” functions in R using default parameters (**Figure RL11A**).

Figure RL11. Process and details of hierarchical clustering

(A) The cluster dendrogram of 12 histological subtypes of sarcoma

2. The criteria to determine the cluster number

The cluster number of hierarchical clustering is determined by the height where the
 cluster dendrogram is cut. To find the appropriate cluster number (k), we cut the cluster
 dendrogram at different heights to get the cluster numbers from 2 to 10 (**Figure RL11B**).

Referring to previous research, we utilized the silhouette coefficient to estimate the
 similarity of samples in one cluster and the difference of samples among different
 clusters. The silhouette coefficients reached the peak when the cluster number was 5 or
 6 (**Figure RL11C**).

To further investigate the clinical availability of our hierarchal cluster, we evaluated the
 association between hierarchal clustering with patients' prognosis. As a result, when the
 cluster number is 6, patients belonging to different clusters presented distinguished
 overall survival time (log-rank test, $p < 0.03$) (**Figure RL11D**), suggesting its potential
 clinical utilization. Therefore, we cut the dendrogram at 0.95 and clustered the 12
 histological subtypes of sarcoma into 6 subgroups: HC1 (AS and ES), HC2 (MLPS and
 WDLPS), HC3 (MFS, DDLPS, and otherFS), HC4 (RMS and SS), HC5 (UPS), and
 HC6 (LMS) (**Figure RL11E**).

**Figure RL11.** (B) The circled cluster dendrograms of sarcoma histological subtypes with
 cluster numbers from 2 to 10. (C) The scaled mean values of silhouette coefficients for different
 cluster numbers. (D) Kaplan-Meier curves for overall survival times when cluster number is 5
 or 6.

**Figure RL11.** (E) Cluster dendrogram for hierarchical clustering when cluster number is 6

**3. Biological insights based on hierarchical clusters**

Besides clinical availability, our HC clustering showed strong biological relevance,
 each subgroup showed distinctive biological features, helping to uncover the intrinsic
 common features of different histological subtypes belonging to the same hierarchical
 cluster. Particularly, in our previous version, we found that HC1 contains AS and ES,
 both of which could be distinguished from other clusters with elevated expression of
 SHC1-TGF β signaling pathways.

In the revision, we conducted further analysis to investigate how hierarchical clusters
 could decipher the common features and heterogeneity among 12 histological subtypes
 of sarcoma. As a result, we found that our hierarchical clustering divided the lipid
 sarcoma (WDLPS, MLPS, and DDLPS) into two clusters. Particularly, DDLPS were
 clustered together with fibrosarcomas (MFS and otherFS) and MPNST in HC3.
 WDLPS and MLPS were clustered into another cluster (HC2). Considering different
 differentiation levels of WDLPS, MLPS, and DDLPS, these findings revealed the
 difference of tumor differentiation within lipid sarcomas might lead to the diverse
 molecular features between DDLPS and WDLPS, further implying that the degree of
 tumor differentiation might serve as an important factor in determining the molecular

features of sarcomas within lipid sarcomas. Because DDLPS is more metastatic and
 proliferative than WDLPS (*Adv Anat Pathol*, PMID: 32960834), we compared the ratio
 of KI67-positive tumor cells in WDLPS and DDLPS. DDLPS showed an obviously
 higher ratio of KI67-positive tumor cells than WDLPS (**Figure RL11F**). Consistently,
 HC3 also presented the higher ratio of KI67-positive tumor cells than HC2, implying
 that HC3 featured fast cell proliferation characteristics (**Figure RL11F**).

**Figure RL11.** (F) Boxplots illustrating the ratio of KI67-positive tumor cells in HC2 and
 HC3 (left) and histological subtypes belonging to HC2/HC3 (right).

 GSEA analysis revealed that DDLPS (HC3) could be distinguished from WDLPS and
 MLPS (HC2) by elevated enrichments of Rab pathway (**Figure RL11G-H**). The
 elevated protein expression of Rab GTPases including RAB14, RAB5A, RAB2A, etc.
 in HC3 confirmed the increased Rab pathway in HC3 (**Figure RL11I**).

**Figure RL11.** (G) The heatmap of specifically enriched pathways in hierarchical clusters; (H)
 Boxplots showing GSEA scores of Rab regulation of trafficking and Rab pathway in
 histological subtypes belonging to HC2/HC3.

Moreover, among the Rab GTPases that showed elevated expression in HC3, we
observed that the protein abundance of RAB2A and RAB14 were significantly
correlated with patients' prognosis (**Figure RL11J**).

**Figure RL11.** (I) The heatmap presenting Rab GTPases enriched in HC3; (J) The forest plot
showing the hazard ratios of Rab GTPases enriched in HC3.

Previous researches have reported that Rab GTPases participated in cell autophagy
(*Cell Death Differ*, PMID: 24440914; *Cell Biosci*, PMID: 33557950). RAB2A has
been proved to regulate the formation of autophagosome and autolysosome (*Autophagy*,
PMID: 30957628). Researches have indicated that the elevated autophagy might be
associated with tumor proliferation (*Clin Cancer Res*, PMID: 26567363), we then
hypothetically assumed that the elevated autophagy might lead to significantly fast
tumor cell proliferation and cell proliferation index in HC3.

Aim to confirm this assumption, we compared the autophagy pathway between HC2
and HC3, and found that both the autophagy pathway enrichment scores as well as
autophagy markers (ATG5, ATG7, MTOR, WIPI1) showed elevation in HC3 than HC2
(**Figure RL11K-M**). Moreover, proliferation index of sarcoma is both correlated with
protein expression of RAB2A and autophagy pathway GSVA scores (**Figure RL11N**).
These findings illustrated that comparing to WDLPS and MLPS which belong to HC3,
DDLPS, which belongs to HC2, showed fast tumor cell proliferation features, which
might be caused by the RAB2A-associated autophagy process.

**Figure RL11.** (K) The scatter plot presenting the positive correlation between RAB2A and
 autophagy pathway; (L) Boxplots presenting the enrichment scores of autophagy in different
 clusters; (M) Boxplots presenting the abundances of autophagy markers in different clusters;
 (N) The scatter plot presenting the positive correlation between proliferation index and
 autophagy pathway (left) or abundance of RAB2A (right).

In sum, our hierarchical clustering showed clinical relevance and could help to illustrate
 the common features among different histological sarcomas and could further decipher
 the distinctive biological features of lipid sarcomas varies with degrees of
 differentiation. In the revised manuscript, we have updated the methods for hierarchical
 clustering in the “**Methods**” section and updated our analysis on the HC2 and HC3 in
 the “**Result**” section (line297-330). Also, we updated **Figure RL2** in the revised
 **Figure2, Supplementary Figure 8&9.**

**(5) line 250. A correlation between TGFbeta proteins and SHC1 does not**
 **necessarily mean that SHC1 plays a key role in TGFbeta signaling. It may suggest**
 **that it plays a role, but this needs more evidence. This should be re-worded.**

**Response:**

We appreciate the reviewer for this helpful suggestion. We apologize for the unclear
 description of the relationship between SHC1, TGFbeta protein, and the elevated cell
 migration features of HC1.

In our previous version, we grouped the 12 histological types of sarcomas into 6
hierarchical clusters (HC), among which HC1 containing both AS and ES showed the
worst prognosis. Differential expression analysis combined with GO pathway analysis
revealed HC1 featured with enrichment of the TGF β signaling pathway. To further
elucidate the mechanism underlying the poor prognosis of HC1 patients, we focused on
the HC1 specifically elevated proteins that enriched in the TGF β signaling pathway,
and identified SHC1 as the top-ranked HC1 elevated protein that associated with
patients' poor prognosis. As an adaptor protein, SHC1 has been reported to interact with
various ligands and activate downstream processes, including TGFbeta signaling
pathway (*EMBO J*, PMID: 17673906). We then performed correlation analysis and
observed positive correlation between SHC1 with both the expression of TGFB3 and
the GSVA scores of both TFGbeta signaling pathway and epithelial cell migration
pathway (Spearman's correlation, p-value < 0.05). For this reason, we then
hypothetically assumed that SHC1 might play an important role in leading the poor
prognosis of HC1 sarcoma, through cooperating with TGFB3 and promoting tumor cell
migration. In agreement with the reviewer's comment, more evidence could help to
elucidate the relationship among SHC1, TGFbeta and elevated tumor cell migrations of
HC1 cluster. In the revision, to illustrate the above relationships, we utilized ASM cell
line, the cell line of AS, to represent the HC1 cluster. We constructed the SHC1-
overexpressed vector and transfected it into the ASM cell line (SHC1-OE-ASM).
Meanwhile, we also utilized shRNA to knock down SHC1 (SHC1-KD-ASM). RT-PCR
analysis was utilized to verify the expression of SHC1 in SHC1-OE-ASM and SHC1-
KD-ASM. The results confirmed the significantly elevated expression of SHC1 in
SHC1-OE-ASM and significantly decreased expression of SHC1 in SHC1-KD-ASM
(**Figure RL12A**). We then evaluated the cell migration rates using transwell assay. As
a result, SHC1-OE-ASM showed increased cell migration ability, whereas SHC1-KD-
ASM exhibited decreased cell migration ability (**Figure RL12B**).

**Figure RL12. Functional experiments to validate the role of SHC1 in the TGFbeta**
 **signaling pathway in sarcoma cell lines**

(A) the expression of SHC1 in SHC1-OE-ASM, SHC1-KD-ASM and controlled cells by RT-
 PCR. (B) The Effects of SHC1 on the migration of ASM cells were confirmed by transwell
 assay. The bar plots indicated the migrated cell counts of ASM cells under different treatments.

We then treated SHC1-OE-ASM and OE-Ctrl-ASM with TGFB3 and evaluated the
 tumor cell migration rates. As a result, SHC1-OE-ASM treated with TGFB3 showed
 significantly elevated tumor cell migration rates, whereas OE-Ctrl-ASM showed no
 significantly changes in tumor cell migration rates by treating with TGFB3 (**Figure**
 **RL12C**). These results confirmed the role of TGFB3 in activating SHC1-mediated
 tumor cell migrations.

**Figure RL12. (C) The effects of TGFB3 on the migration of ASM cells were confirmed by**
 **transwell assay. The bar plots indicated the migrated cell counts of ASM cells under different**

treatments.

In sum, our data illustrated the TGFB3 might participate in promoting tumor cell
migration through cooperating with SHC1. According to reviewer's comments, we also
toned down our statements as follows: "Consistently, we found a significantly positive
correlation between the protein abundance of SHC1 and the TGF β signaling pathway
enrichment score (Pearson's correlation, $r = 0.15$, p -value = 0.028), suggesting that
SHC1 might participate in the TGF β signaling in sarcoma (Figure 2E). Among the
TGF β families, TGFB3 showed a statistically positive correlation with SHC1
(Pearson's correlation, $r = 0.25$, p -value = 0.026), suggesting the potential association
between TGFB3 and SHC1, and implying they might cooperate to impact downstream
signaling pathways (Figure 2E)". Besides above updates, we also added the results of
the functional experiments on lines 378-384.

**(6) Line 283: In sum, None of that is shown with enough evidence. The language**
**should be toned down. Higher kinase expression does not necessarily mean higher**
**kinase activity. Did ADD2 S2 phosphorylation level drop with inhibition of SHC1?**
**How specific is the inhibitor. What is the kinase phosphorylating ADD S2?**

**Response:**

Thanks again for the constructive suggestions. We apologized for the unclear
description on the relationship among SHC1, phosphorylation of ADD2 and tumor cell
migrations. In the revision, to decipher this relationship, we performed the following
analysis and functional experiments:

**1. Comparative and correlation analysis revealed PTK2 as the core kinase that**
**linked SHC1 and the phosphorylation of ADD2.**

Published researches have indicated that SHC1 participated in various biological
process, and might regulate downstream pathways through phosphorylation (*Nature*,
PMID: 23846654; *Nat Commun*, PMID: 28276425; *Front Cell Dev Biol*, PMID:
33693003). Therefore, in our previous version, to further illustrate how SHC1 led to
cell migration, we performed correlation analysis and observed that the
phosphorylation of ADD2 (functions in cytoskeleton reorganization and epithelial

migration) at Ser2 showed the most significantly correlation with SHC1. Combined
with clinical information, we found the phosphorylation of ADD2 at Ser2 was
significantly associated with patients' poor prognosis.

Functionally, SHC1 is an adapter protein that could interact with different kinases and
participated in signal transduction pathways (*Nature*, PMID: 23846654). In the revision,
to elucidate the kinase that related to SHC1 and might regulate the phosphorylation of
ADD2 at Ser2 in HC1, we referred to the public database (PhosphoSite [[https://](https://www.phosphosite.org/homeAction.action)
www.phosphosite.org/homeAction.action], Phos-pho.ELM [[http://](http://phospho.elm.eu.org/dataset.html)
phospho.elm.eu.org/dataset.html], and PhosphoPOINT [[http://](http://kinase.bioinformatics.tw/)
kinase.bioinformatics.tw/]) and conducted correlation analysis. As a result, among the
kinases reported to regulate phosphorylation of ADD2, PTK2 was identified as the
kinase that showed most significantly correlation with SHC1 and comparatively higher
expression in HC1 cluster (**FigurRL13A-B**).

Figure RL13. SHC1 recruits PTK2 to phosphorylate ADD S2

(A) The Spearman-rank correlation of the expression of PTK2 with SHC1 expression
(Spearman's correlation). (B) The violin plot indicated the PTK protein expression among HC
clusters.

**1.2. Inhibiting PTK2 could impact the increased cell migration leading by SHC1.**

To further investigate the role of PTK2 in impacting cell migration, SHC1-OE-ASM
and OE-Ctrl-ASM cell lines were used and were treated with PTK2 inhibitors. We then
evaluated the cell migration by transwell assay. As a result, inhibiting PTK2 could
significantly decreased the cell migration rates increased by SHC1 (**Figure RL13C-D**).

Moreover, overexpression of PTK2 in SHC1-KD-ASM significantly increased cell
 migration which was inhibited by knocking down SHC1(**Figure RL13D**). These results
 implied that the kinase, PTK2, participated in cell migration driven by SHC1.

**Figure RL13.** (C-D) The effects of SHC1-PTK2 axis on the migration of ASM cells were
 confirmed by transwell assay. The bar plots indicated the migrated cell counts of ASM cells
 under different treatments.

 We further performed phosphoproteomic analysis between SHC1-OE-ASM treated
 with or without PTK2 inhibitor. As a result, the phosphorylation of proteins such as
 ADD2 Ser2, FGD4 Ser702 and EPB41 Ser542, which participate in actin cytoskeleton
 reorganization and epithelial cell migration, showed significantly elevation in SHC1-
 OE-ASM and significantly decreasing in SHC1-OE-ASM treated with PTK2 inhibitor
 (**Figure RL13E**). These observations confirmed the role of PTK2 in phosphorylating
 ADD2 at Ser2 and elevating actin cytoskeleton reorganization pathways.

 **Figure RL13.** (E) The boxplots indicating the phosphorylation intensity of ADD2 S2 and other
 phosphosites participating in actin cytoskeleton reorganization under different treatments.

In sum, our data illustrated the mechanism that by interacting with PTK2 and
phosphorylating ADD2 at Ser2, SHC1 will enhance the cell migration, and lead to poor
prognosis of HC1 patients. According to reviewer's comments, we also updated our
statements as following: "In sum, the upregulation of SHC1 might interact with kinase
PTK2, phosphorylating ADD2 at Ser2, enhanced cell migration. This phosphorylation
cascade might associate with the poor prognosis with HC1 patients (AS or ES)."

In the revision, we have updated **Figure RL3** in the revised **Supplementary Figure**
**10&11** and the "**Result**" section on lines297-334, line355-384, line 397-417, and line
520-551. in the revised manuscript.

**(7) Fig 3 and Supp Fig 6: Is the inhibition of SHC1 and MAPK10 affecting the**
**phosphorylation levels at CTNNB1Ser552 and Ser675?**

**Response:**

We appreciate the reviewer's comment and apologize for the not clearly illustrating the
mechanism how SHC1 and MAPK10 affect the phosphorylation levels at CTNNB1
Ser552 and Ser675. In the revision, to elucidate the mechanism, we conducted the
following analysis and functional experiments:

**1. Comparative and correlation analysis revealed CSNK1G1 as the core kinase**
**that linked SHC1 and the phosphorylation of CTNNB1 at Ser552.**

As an adaptor protein, SHC1 has been reported to participate in various signaling
pathways. To illustrate the kinase that related to SHC1 and might regulate the
phosphorylation of CTNNB1 at Ser552 in Pc-Ra, we also referred to the public database
(PhosphoSite [[https:// www.phosphosite.org/homeAction.action](https://www.phosphosite.org/homeAction.action)], Phos-pho.ELM
[[http:// phospho.elm.eu.org/dataset.html](http://phospho.elm.eu.org/dataset.html)], and PhosphoPOINT [[http://](http://kinase.bioinformatics.tw/)
kinase.bioinformatics.tw/]) and conducted correlation analysis. As a result, the among
the public reported kinases of CTNNB1, CSNK1G1 showed the significantly positive
correlation with both SHC1 and the phosphorylation of CTNNB1 at Ser55 (**Figure**
**RL14A-B**). Consistently, the phosphorylation of CSNK1G1 also showed elevated
expression level in PC-Ra (**Figure RL14C**).

**Figure RL14. phosphorylation levels of CTNNB1Ser552 and Ser675 are impacted by**
 **SHC1 and MAPK10 inhibitors.**

(A)The scatter plot illustrates the positive correlation between CSNK1G1 and SHC1
 (Spearman’s correlation). (B) The scatter plot illustrates the positive correlation between
 CSNK1G1 and the phosphorylation level of CTNNB1 Ser552 (Spearman’s correlation).(C)
 The boxplot presents the expression of CSNK1G1 in different proteomic clusters.

**2. Phosphoproteomic analysis using SHC1-overexpressed cell line confirmed the**
 **role of CSNK1G1 in phosphorylating CTNNB1 at Ser552.**

To further confirm the role of CSNK1G1 in phosphorylating CTNNB1, we constructed
 the SHC1-overexpressed vector and transfected it into the ISOHAS cell line (the cell
 line of AS) which showed similar expression patterns with PC-Ra-HC1. We then treated
 SHC1-OE-ISOHAS with or without the CSNK1G1 inhibitor and performed
 phosphoproteomic analysis. As a result, the phosphosites of proteins participating in
 angiogenesis, especially CTNNB1 Ser552, significantly decreased in SHC1-OE-
 ISOHAS treated with CSNK1G1 inhibitor (**Figure RL14D**). These observations
 confirmed the role of CSNK1G1 in phosphorylating CTNNB1 at Ser552. The above
 results confirmed our assumption that SHC1 could lead to PC-Ra-HC1 tumor migration
 through phosphorylating CTNNB1 mediated by CSNK1G1.

**Figure RL14.** (D) The boxplots indicated the phosphorylation levels of CTNNB1 Ser552 and
 other phosphosites participating in angiogenesis under different treatments.

**3. Phosphoproteomic analysis using MAPK10-overexpressed cell line confirmed**
 **the role of MAPK10 in phosphorylating CTNNB1 at Ser675.**

As for the impact of MAPK10 on the phosphorylation of CTNNB1 at Ser675. We
 constructed the MAPK10 overexpressed vector and transfected it into SW872 cell line
 (MAPK10-OE-SW872) which showed similar expression patterns with PC-Ra-oHCs.
 We then treated MAPK10-OE-SW872 cells and treated with or without MAPK10
 inhibitor. We also conducted phosphoproteomic analysis, and observed the
 phosphorylation of proteins such MAPK13, CTNNB1 and MAPK14 which participate
 in MAPK signaling pathway, showed significantly elevated expression in MAPK10
 overexpressed cells and downregulated in MAPK10 inhibitor treated cell lines (**Figure**
 **RL14E**). The above results confirmed our assumption that MAPK10 could lead to PC-
 Ra-oHCs tumor migration through phosphorylating CTNNB1 at Ser675.

**Figure RL14.** (E) The boxplots indicated the phosphorylation levels of CTNNB1 Ser675 and
 other phosphosites participating in MAPK signaling cascade under different treatments.

In the revision, we have updated the relationship among SHC1-PTK2-phosphorylated
 CTNNB1 at Ser552, and the relationship among MAPK10-phosphorylated CTNNB1

at Ser675 on lines 520-551 of the “**Result**” section. We also updated the **Figure RL14**
into **Supplementary Figure 11** of the revised manuscripts, respectively.

**(8) Fig 7P and line 457. There is lots of evidence missing for RIOK1**
**phosphorylating NPM1 and thereby regulating the interaction of APEX1 and**
**NPM1. Does inhibition/KD of the kinase affect the phosphorylation level**
**(phosphoproteomics, WB)? Does the inhibition affect the interaction of the 2**
**proteins (IP-MS, WB)? Does it affect the co-regulation of the two proteins**
**(proteomics)?**

**Response:**

We appreciate the reviewer for this critical suggestion and agree with that more
evidence should be provided to verify our findings on the RIOK1-phosphorylated-
NPM1-APEX1 axis in promoting tumor cell proliferations. According to the reviewer’s
suggestion, in the revision, we performed further analysis and functional experiments
to confirm our findings.

Specifically, we utilize the sarcoma cell line, RKN, for further functional experiments,
as it originates from LMS and represents the proteomic features of PC-Cc. We
constructed the RIOK1-overexpressed RKN cell line (RIOK1-OE-RKN) through the
RIOK1 overexpression plasmid, pCDH-RIOK1-copGFP. Moreover, shRNA of RIOK1
were designed and transfected into RKN cell line to knock down the expression of
RIOK1 (RIOK1-KD-RKN). We then performed CCK8 cell proliferation assay and
evaluated the cell proliferation rates. As a result, RIOK1-OE-RKN showed most
significantly elevated cell proliferation rates and RIOK1-KD-RKN had significantly
decreased cell proliferation rates (**Figure RL15A**). We also treated RIOK1-OE-RKN
cell line with RIOK1 inhibitor, and the inhibitor significantly decreased the
proliferation of RIOK1-OE-RKN (**Figure RL15A**). These observations confirmed the
impact of RIOK1 on promoting sarcoma tumor cell proliferation. We then performed
comparative proteomic and phosphoproteomic analysis among RKN sarcoma cell lines
with different treatments (RKN transfected with empty vector, RIOK1-OE-RKN,
RIOK1-OE-RKN treated with RIOK1 inhibitor, RKN transfected with scrambled

shRNA, RIOK1-KD-RKN). As a result, besides APEX1, the proteins participating in
 DNA base excision repair including XRCC1, XRCC4, POLB, as well as cell
 proliferation index KI67 showed elevated expression in RIOK1-OE-RKN (**Figure**
 **RL15B-C**). Intriguingly, the phosphorylation of NPM1 at Ser 125 was significantly
 increased in RIOK1-OE-RKN, implying that RIOK1 regulated the phosphorylation of
 NPM1 (**Figure RL15C**).

**Figure RL15. Functional experiments to validate the role of RIOK1 in phosphorylating**
 **NPM1 and interaction of NPM1 and APEX1**

(A) Proliferation of the RKN cell line associated with different treatments (n = 4 repeats per
 group). (B) The heatmap reveals the expression patterns of DNA base excision proteins across
 the cells associated with various treatment (n = 3 repeats per group). (C) The boxplots reveal
 the abundance of APEX1, KI67 and phosphorylation of NPM1 at Ser125 in RKN cell line with
 different treatments.

To further investigate the impact of NPM1 phosphorylation on cell proliferation as well
 as on its interaction with APEX1, we then constructed NPM1 phosphorylation site
 mutant plasmid, NPM1^{S125A}, and transfected it into RIOK1-KD-RKN cells (NPM1-
 mut-OE-RIOK1-KD-RKN). The non-mutant NPM1 was also transfected into RIOK1-
 KD-RKN cells (NPM1-OE-RIOK1-KD-RKN) which were utilized as controls. By
 evaluating the cell proliferation rates, we observed that comparing to RIOK1-KD-RKN
 cells, NPM1-OE-RIOK1-KD-RKN cells should elevated cell proliferation rates,

whereas the cell proliferation rates of NPM1-mut-OE-RIOK1-KD-RKN showed no
 significant elevation (**Figure RL15D**). These results indicated the decreased cell
 proliferation rates led by knocking down RIOK1 could only be rescued by the wild type
 NPM1 overexpression, which further emphasized the role of phosphorylation of NPM1
 in mediating RIOK-dependent regulation of the tumor cell proliferation.

To further illustrate whether the phosphorylation of NPM1 affected its interaction with
 APEX1, we performed IP-MS using both NPM1-mut-OE-RIOK1-KD-RKN and
 NPM1-OE-RIOK1-KD-RKN (**Figure RL15F**). As a result, 17 proteins were identified
 to interact with the wild type NPM1, but not NPM1^{S125A}. Among them, NPM1
 presented the highest abundance, proving that NPM1 Ser125 is the pivotal site for the
 interaction between NPM1 and APEX1 (**Figure RL15G-H**). The above results
 illustrated the potential mechanism that RIOK1 could impact sarcoma tumor cell
 proliferation through phosphorylating NPM1 which then interacted with APEX1 and
 promoted tumor cell proliferation accordingly.

**Figure RL15.** (D) Proliferation of the RKN cell line associated with various treatments (n = 4).
 (E) The boxplots present the expression of KI67, APEX1 and phosphorylation of NPM1 among
 NPM1-OE-RIOK1-KD-RKN, NPM1-mut-OE-RIOK1-KD-RKN, and EV-RIOK1-KD-RKN.
 (F) The schematic work flow of the IP-MS experiment for the NPM1. (G) The diagram
 illustrates the mechanism underlying cell proliferation of PC-Cc driven by NPM1 and APEX1.

(H) The heatmap reveals the expression patterns of DNA base excision proteins across the
NPM1-OE-RIOK1-KD-RKN, NPM1-mut-OE-RIOK1-KD-RKN (n = 3 repeats per group).

In the revision, we have added **Figure RL15** in **Supplementary Figure 12**, and
updated our description on the role of RIOK1-phosphorylated-NPM1-APEX1 axis in
promoting tumor cell proliferations in PC-Cc, with more evidence (both from
functional experiments and bioinformatic analysis). Please see the lines 631-669 in the
“**Results**” section of the revised manuscript.

**(9) Fig 7O and line 527: Evidence is missing. Does inhibition/KD of MAPK10 affect**
**the CTNNB1 Ser657 phosphorylation level. Does the inhibition of MAPK10 in**
**cells derived from the according strain affect immune infiltration (xenograft**
**model)?**

**Response:**

We sincerely thank the reviewer for the comment. To comprehensive respond to the
comment, we divided the response into two parts.

**1. The impact of MAPK10 on phosphorylation of CTNNB1 at Ser657**

As we responded to Q7-part3, indeed, by both knocking down the expression of
MAPK10 or inhibiting its kinase activity could significantly decrease the
phosphorylation of CTNNB1 at Ser657. Please see the response for Q7 for details.

**2. The impact of MAPK10 on tumor immune infiltrations (xenograft model).**

According to the reviewer’s suggestion, we further validated the impact of MAPK10
on tumor immune infiltration using C57/BL6J mice, which usually used as the model
for immune microenvironment analysis (*Nature Reviews Cancer*, PMID: 27687979;
*Cell Reports*, PMID: 35732118; *Clin Cancer Res*, PMID: 15709162). We constructed
xenograft mice models using SW872 cells in which MAPK10 were stably
overexpressed or knocked down. Twenty C57/BL6J mice were randomized into four
groups (n = 5 each), and separately injected MAPK10 overexpressed and MAPK10
knocked down SW872 cell lines (OE-MAPK10 and sh-MAPK10) and control cell lines

(OE-Ctrl and sh-Ctrl) to form subcutaneous tumors. Tumor size and weight were
 measured throughout the tumor growth process and tumor volume was calculated. After
 4 weeks, mice were sacrificed and tumors were collected for further proteomic and IHC
 staining analysis. As a result, tumors from mice transplanted with OE-MAPK10-
 SW872 showed significantly increased immune cell infiltrations, which were evidenced
 by elevated expression of T cell and macrophage markers (CD4, CD8 and CD163).
 Moreover, the immune checkpoint proteins such as CD274 (PD-L1) and CD80 were
 also observed to be elevated in OE-MAPK10-SW872 mice (**Figure RL16A**). On the
 contrary, mice which were transplanted with sh-MAPK10-SW872 showed obviously
 decreased immune cell infiltrations, with decreased expression of both immune cell
 markers as well as immune checkpoint proteins (**Figure RL16A**).

**Figure RL16. The impact of MAPK10 on immune infiltration in mouse xenograft model.**

(A) Boxplots illustrate the expressions of immune cell markers, including CD274, CD80, CD4,
 and CD8 in differently treated mouse xenograft models.

IHC staining further confirmed the increased immune cell infiltrations in OE-
 MAPK10-SW872 mice and decreased immune cell infiltrations in sh-MAPK10-
 SW872 mice (**Figure RL16B**).

We updated the above results about the impact of MAPK10 on immune infiltration from
 line 753 to line 772 of the ‘**Result**’ section.

**Figure RL16.** (B) IHC images illustrate the expression of CD8, CD163, and CD274 in
 subcutaneous tumors of the C57/BL6J mice transplanted with SW872 sarcoma cell lines.
 Positive cell percentage is presented on the right.

**(10) As the control samples are matched tumor-adjacent tissue, the authors may**
 **consider comparing sarcoma and control tissue in a patient-specific manner to**
 **better understand tumor/normal differences (does it matter if I normalize the**
 **sarcoma proteome by the adjacent tissue proteome for each patient, rather than**
 **compare all control samples with all sarcoma samples?).**

**Response:**

We appreciate the reviewer's constructive comments. In our previous version, to
 present the features of tumors and NATs, we performed comparative analysis between
 all tumors and all NATs. The results illustrated that proteins elevated in tumor tissues
 majorly enriched in biological pathways such as cell growth, RNA splicing, and antigen
 processing and presentation. On the other hand, proteins dominantly expressed in NATs
 were enriched in ATP metabolic process, glycolytic process, and muscle system process.

To address the reviewer's comments, in the revision, we conducted further tumor and
 NAT comparative analysis, by normalizing the sarcoma proteome using the adjacent
 tissue proteome for each patient. As a result, the GO features of tumors and NATs
 basically remained unchanged comparing to our previously portrayed molecular

features of all tumors and all NATs. Specifically, in concordant with our previous results,
 the GO enrichment analysis revealed that sarcoma tumors were also featured with
 biological pathways such as cell cycle, synthesis of DNA, MYC targets up, signaling
 by interleukins, and antigen processing and presentation (**Figure RL17A**). Meanwhile,
 the muscle system process, actin filament organization, and TCA cycle we observed to
 be enriched by proteins elevated in the NATs (**Figure RL17A**). These results illustrated
 that the distinctive biology pathways between tumors and NATs is stable and largely
 unaffected by comparison methods.

**Figure RL17. Pairwise comparison between NAT and tumor of sarcoma**

(A) The heatmap presents the significant difference of enriched pathways between tumors and
 NATs through the pairwise comparison.

 Moreover, we also compared the difference of biology pathways among histological
 subtypes of sarcoma utilizing the tumors' proteome which was normalized by paired
 NAT samples. Compared with our previous result, the histological specific features of
 sarcomas basically remained unchanged (**Figure RL17B-C**). For instance, TGF β
 signaling and p53 pathway were dominantly enriched in AS, myogenesis were observed
 to be elevated in LMS, and MYC target pathway was significantly enriched in SS&UPS
 (**Figure RL17B**), etc. The above results confirmed that the distinctive biological
 features of diverse histological subtypes of sarcoma remain the same despite whether
 being normalized by paired NAT samples.

**Figure RL17.** (B) Heatmaps illustrates enriched cancer hallmarks in STS histologic subtypes
 through non-pairwise (top) and pairwise (down) methods. (C) Boxplots presents the enriched
 pathways in specific histological subtypes processed through non-pairwise or pairwise method.
 Based on the above results, in the revision, following reviewer's suggestion, besides
 our original results about the comparison of NATs and tumors, we also added the results
 of paired comparison between tumor and NAT in **Supplementary Figure 4**. Moreover,
 we also added sarcomas' histological specific features that were also normalized by
 their paired NATs in the **Supplementary Figure 5**. Please see lines 197-199 and lines
 241-244 in the revised manuscript.

REVIEWERS' COMMENTS

Reviewer #1 (Remarks to the Author):

The authors have addressed all my questions and provided a detailed explanation on how each question was addressed.

Reviewer #2 (Remarks to the Author):

The authors appropriately discussed or answered to comments raised during the review process.

Reviewer #3 (Remarks to the Author):

The authors carefully addressed all my suggestions and comments and did several follow-up experiments to confirm hypotheses they have stated in the original experiment. I think the paper is suitable for publications if the below comments/suggestions are addressed.

(i) I think it is wrong to state that 15 k plus proteins were identified across all datasets based on the results of filtering each run individually and then combine the protein lists. As the 10 k protein received from the combined dataset protein filtering shows, the FDR of the identified 15 k proteins is about 33 %. I suggest just reporting the 10 k proteins (or mentioning the estimated FDR for the 15 k proteins).

(ii) The authors made follow-up experiments on all my comments and were able to confirm all hypotheses stated in the original manuscript. This is quite some work and impressive. The hypotheses included identifying the phosphorylation at ADD2 Ser2, the effect of SHC1 and MAPK10 inhibition on CTNNB1 Ser552 and Ser 675, and the regulation of NPM1/APEX1 binding through RIOK1 catalyzed phosphorylation of NPM1. The authors should provide the entire proteome and phospho datasets from this experiments as supplemental tables and upload the RAW files to the repository.

Comments not affecting my recommendation on publishing:

(a) a median 30 % CV is quite high for technical replicates. The authors only show S/N to CV correlation at the protein level. What I recommend is to look at this relationship at the peptide level, as this may

readily allow to filter out low S/N peptides with poor CVs without affecting the protein counts too much. I don't think the message in this manuscript will be hugely affected when applying this filter (and I am not requesting this), but in general, I would recommend trying to improve CV by some S/N filtering.

(b) How were the numbers in Table RL6 generated? I did not look at every paper, but a median CV of 0.99 for a TMT dataset produced by Steve Carr seems completely off (PMID: 33212010). What we should consider is the median CVs of technical replicates (not across all samples analyzed in a study, this would rather be sample-dependent than method-dependent). For TMT this should be in the 0.05-0.1 range. I could not find any data in the Carr paper. This is not relevant to the reviewed manuscript, but the table seems to be off.

Reviewer #3 (Remarks to the Author):

The authors carefully addressed all my suggestions and comments and did several follow-up experiments to confirm the hypotheses they stated in the original experiment. I think the paper is suitable for publication if the below comments/suggestions are addressed.

Response:

We sincerely appreciate the constructive comments that the reviewer has provided, which truly help us in improving our work. We have revised the manuscript and provided specific point-to-point responses as follows:

Q1. I think it is wrong to state that 15 k plus proteins were identified across all datasets based on the results of filtering each run individually and then combining the protein lists. As the 10 k protein received from the combined dataset protein filtering shows, the FDR of the identified 15 k proteins is about 33 %. I suggest just reporting the 10 k proteins (or mentioning the estimated FDR for the 15 k proteins).

Response:

We are grateful for the constructive comment that the reviewer has provided. According to the reviewer's comment, we have removed the statement that 15k plus proteins were identified across all datasets, and revised the description as follows: "Quality control was applied on both peptide and protein level with less than 1%FDR. As a result, 10,118 proteins and 37,842 phosphosites were identified, with 5,593 proteins and 6,483 phosphosites per sample on average." Please see lines 154-157 in the result section of the revised manuscript.

Q2. The authors made follow-up experiments on all my comments and were able to confirm all hypotheses stated in the original manuscript. This is quite some work and impressive. The hypotheses included identifying the phosphorylation at ADD2 Ser2, the effect of SHC1 and MAPK10 inhibition on CTNNB1 Ser552 and Ser 675, and the regulation of NPM1/APEX1 binding through RIOK1 catalyzed phosphorylation of

NPM1. The authors should provide the entire proteome and phosphor datasets from these experiments as supplemental tables and upload the RAW files to the repository.

Response:

We appreciate the reviewer's comments. We have now deposited all the RAW files of the entire proteome and phosphor datasets to the iProX (<https://www.iprox.org/>). Specifically, all the proteome and phosphoproteome datasets for the cohort study can be accessed with the ProteomeXchange ID: PXD047297. For functional studies, all the raw data can be accessed with the iProX accession: IPX000764500, with the following url: <https://www.iprox.cn/page/PSV023.html?url=1701407935333pK2j>, and password: BbZF. The entire proteome and phosphoproteome datasets from these experiments were uploaded to OMIX and can be accessed with the accession no OMIX005327. Moreover, we provide source data for all data presented in graphs within the Figures. For details, please see the Data Availability section in the revised manuscript and the Source data.

Comments not affecting my recommendation on publishing:

Q1. A median of 30 % CV is quite high for technical replicates. The authors only show S/N to CV correlation at the protein level. What I recommend is to look at this relationship at the peptide level, as this may readily allow to filter out low S/N peptides with poor CVs without affecting the protein counts too much. I don't think the message in this manuscript will be hugely affected when applying this filter (and I am not requesting this), but in general, I would recommend trying to improve the CV by some S/N filtering.

Response:

We thank the reviewer for the constructive suggestion. Taking the reviewer's suggestion, we have calculated the S/N to CV correlation at the peptide level. As a result, the median CV calculated based on S/N ratios at the peptide level was 0.25 and the mean CV was 0.32. The result is comparable with the CV at the protein level (**Figure RL1A-B**). In agreement with the reviewer, the median CV is a bit high for technical replicates. Thus, to improve the CV for technical replicates, we took the reviewer's suggestion and filtered out low S/N peptides with poor CVs (peptides with

CV > 0.30). As a result, the median CV at peptide level was decreased to 0.18 (**Figure RL1C**). Moreover, after filtering out peptides with low CV, we then evaluated the number of proteins and calculated the CV at the protein level. As a result, the number of proteins was 7,229 (7,564 before peptide filtering) which was not affected too much by the filtering process (**Figure RL1D**). Meanwhile, the median CV, at the protein level, was significantly decreased to 0.14 (0.30 before peptide filtering) (**Figure RL1E**). These results indicated the peptide-filtering process significantly decreased the variability across the technical repeats at both peptide and protein levels, without affecting protein counts. In the revision, we revised our statement about the CVs across the technique repeats as follows: “The correlations of these control samples were 0.83-0.95 and the median coefficient of variation (CV) was 0.14 (**Methods, Supplementary Figure 1A-C**), which is comparable to previously published papers (*Nature Medicine*, **PMID: 35654907**), presenting the stability of the mass spectrometry across quality controls.”, and added description about the S/N filtering process in the **Methods** section. Please see lines 164–167 in the result section, and lines 1122–1127 in the **Methods** section for details.

Figure RL1. The CVs and S/N of quality control samples' peptides and proteins

A-C and E. The left scatter plots illustrate the CV and S/N ratio of proteins/peptides identified in all HEK293 stand samples. The right cumulative distribution curves illustrate the distribution of CVs. (A) CV and S/N ratio at the protein level (before peptide filtering); (B) CV and S/N ratio at the peptide level (before peptide filtering); (C) CV and S/N ratio at the peptide level (after peptide filtering); (E) CV and S/N ratio at the protein level (after peptide filtering). (D) The bar plot presents the identified protein numbers before and after filtering the peptides.

(b) How were the numbers in Table RL6 generated? I did not look at every paper, but a median CV of 0.99 for a TMT dataset produced by Steve Carr seems completely off (PMID: 33212010). What we should consider is the median CVs of technical replicates (not across all samples analyzed in a study, this would rather be sample-dependent than method-dependent). For TMT this should be in the 0.05-0.1 range. I could not find any data in the Carr paper. This is not relevant to the reviewed manuscript, but the table seems to be off.

Response:

We thank the reviewer for the instructive suggestion. We apologize for the unclear presentation of the Table RL6. We agree with the reviewer that we should calculate the CVs among technical replicates, not across all samples. In the revision, we screened out the published papers, and since very few papers provided data for technical replicates, we only found the paper by Matthias Mann's group (*Nature Medicine*, PMID: 35654907) that provided quality control data. They utilized DIA methods, and the median CVs across quality assessment samples were 0.12 - 0.19, comparable to our results (our result median CV is 0.14). In the revision, we have updated the citation and revised our description on the quality control. Please see lines 164–167 in the result section, and lines 1122–1127 in the **Material and Methods** section for details.